# Early-life gut microbiome associates with positive vaccine take and shedding in neonatal schedule of the human neonatal rotavirus vaccine RV3-BB

Rotavirus vaccines are less effective in high mortality regions. A rotavirus vaccine administered at birth may overcome challenges to vaccine uptake posed by a complex gut microbiome. We investigated the association between the microbiome and vaccine responses following RV3-BB vaccine (G3P[6]) administered in a neonatal schedule (dose 1: 0-5 days), or infant schedule (dose 1: 6-8 weeks) in Indonesia (Phase 2b efficacy study) (*n* = 478 samples/193 infants) (ACTRN12612001282875) and in Malawi (Immunigenicity study) (*n* = 355 samples/186 infants) (NCT03483116). Vaccine responses assessed using anti-rotavirus IgA seroconversion (IgA), stool shedding of vaccine virus and vaccine take (IgA seroconversion and/or shedding). Here we report, high alpha diversity, beta diversity differences and high abundance of *Bacteroides* is associated with positive vaccine take and shedding following RV3-BB administered in the neonatal schedule, but not with IgA seroconversion, or in the infant schedule. Higher alpha diversity was associated with shedding after three doses of RV3-BB in the neonatal schedule compared to non-shedders, or the placebo group. High abundance of *Streptococcus* and *Staphylococcus* is associated with no shedding in the neonatal schedule group. RV3-BB vaccine administered in a neonatal schedule modulates the early microbiome environment and presents a window of opportunity to optimise protection from rotavirus disease.

Rotavirus vaccines have resulted in a significant decline in global rotavirus-associated deaths and hospitalisations[1]. As of August 2024, 127 countries have introduced rotavirus vaccines into national or sub-national immunisation programmes, including 42 GAVI (The Vaccine Alliance) eligible countries[2]. Despite this major achievement, rotavirus remains the most common cause of severe dehydrating diarrhoea in children less than 5 years of age, causing over 230,000 deaths globally each year[3]. Challenges ensuring timely administration of a full 2 or 3-dose schedule from 6 weeks of age, cost and ongoing concerns regarding safety and rotavirus strain replacement remain barriers to

the success of rotavirus vaccines[4]. Rotavirus vaccines have also been shown to be 23% to 47% less effective in high child mortality regions compared to low child mortality regions[5–12]. The reasons for this disparity are not fully understood but presumed to be multifactorial, including interference by co-administered oral poliovirus vaccine, histoblood group antigen status and enteric pathogen load reflective of the living environment[13–15]. An association between the gut microbiome and serum immunoglobin A (IgA) response following administration of a rotavirus vaccine (Rotarix®; GlaxoSmithKline, Belgium and RotaTeq®; Merck, USA) has been reported in infants in Ghana[16],

e-mail: josef.wagner@mcri.edu.au; jebines@unimelb.edu.au

Pakistan[17], Zimbabwe[18], Nicaragua[19] and Spain[20,21] with inconsistent outcomes described. A negative correlation between alpha diversity of the gut microbiome and IgA seroconversion and post-vaccination anti-rotavirus serum IgA concentration was reported after administration of the Rotarix® vaccine in infants in India and Malawi[22].

Rotavirus vaccines are orally administered with the first dose after 6 weeks of age, and subsequent doses at four-week intervals. It has been postulated that administration of the first dose of a rotavirus vaccine at birth may limit challenges to vaccine uptake and replication observed in older infants. The newborn gut is developmentally immature, characterised by a thin mucus layer, increased epithelial permeability and immaturity of the mucosal system, which may contribute to the increased susceptibility to gastrointestinal infection observed in newborns[3,23–25]. The newborn gut microbiome is also less complex, initially reflecting maternal flora, but develops over the first months of life in response to feeding, antibiotic and environmental exposure and, in turn, is shaped, by the developing infant immune system[26]. Regional differences in the composition of the gut microbiome are reported in infants (0 to 1 year) and toddlers (1 to 3 years) between countries and between continents[27–29]. Early colonisation of the gut microbiome with opportunistic pathogens (i.e., *Klebsiella, Streptococcus, Staphylococcus* and *Enterococcus)*, has been linked to late onset neonatal sepsis, particularly in low- and middle-income countries (LMICs)[30].

The RV3-BB vaccine is based on the asymptomatic human neonatal rotavirus strain RV3 (G3P[6]) developed with the aim of administering the first dose at birth to provide early protection from severe rotavirus disease[31,32]. Neonatal P[6] asymptomatic rotavirus strains differ from disease-causing rotavirus strains in both structure and function[33]. Neonatal P[6] strains, such as RV3, are naturally adapted to the newborn gut and replicate well in the presence of maternal and breast milk antibodies, irrespective of histoblood group antigen status[34–37]. In the RV3-BB efficacy trial in Indonesia, the neonatal schedule (with the first dose administered at 0 to 5 days of age) was associated with 75% protective efficacy against severe rotavirus gastroenteritis at 18 months of age compared with 51% when administered in the infant schedule (with the first dose from 8 weeks of age)[32]. Subsequent modelling suggests that there could be less waning of protection when a rotavirus vaccine is administered in the neonatal schedule[38].

We hypothesised that vaccine response (vaccine take, IgA seroconversion and/or stool vaccine virus shedding) following administration of RV3-BB in a neonatal schedule (first dose administered at 0 to 5 days of age) would be associated with characteristics of the early gut microbiome (alpha diversity, beta diversity and bacterial taxa profile), but that these changes would not be observed when vaccine was administered in the infant schedule (first dose administered at 6 to 8 weeks of age). To explore this association, we analysed the gut microbiome in stool samples collected during clinical trials of the RV3-BB vaccine in Indonesia[32] and Malawi[31], focusing on three key study time points, week 1 when the neonatal schedule group had received their first dose of vaccine compared to a placebo dose as the first dose in the infant schedule group; week 6 when the neonatal schedule group had received two doses of vaccine compared to one placebo and one vaccine dose in the infant schedule group and after three doses of vaccine in either the neonatal or infant schedule (week 14–20) and analysed this in relation to measures of vaccine response (vaccine take, IgA seroconversion, vaccine virus stool shedding). To examine if the administration of a live rotavirus vaccine (RV3-BB) administered soon after birth influenced the development of the gut bacterial microbiome, we compared the gut microbiome (alpha diversity, beta diversity and bacterial taxa profile) in vaccine recipients and placebo recipients over time. To determine if our findings were vaccine-specific (RV3-BB:G3P[6]) or administration schedule-specific (neonatal schedule versus infant schedule), we then compared the abundant bacterial taxa identified at key study timepoints in the RV3-BB clinical trials conducted in Indonesia and Malawi with data obtained from clinical trials of the Rotarix® vaccine (G1P[8]) administered in the routine infant schedule conducted in India and Malawi[22].

## Results

### Sample framework and bacterial 16S rRNA analysis

Participants from the Indonesia RV3-BB trial and Malawi RV3-BB trial were eligible for inclusion into the RV3-BB Microbiome study cohort if they had adequate stool volume available for microbiome analysis at key study time points (Malawi: baseline, week 1, week 6, week 14; Indonesia: week 1, week 14–16, week 18–20) in the per-protocol population (Indonesia $n = 193$; Malawi $n = 186$). The faecal microbiome was characterised from 478 samples from 193 children enrolled in the RV3-BB clinical trial conducted in Indonesia (http://www.anzctr.org.au/Trial/Registration/TrialReview.aspx?ACTRN=12612001282875)[32] and from 355 samples from 186 children in the RV3-BB clinical trial conducted in Malawi. (clinicaltrials.gov/ct2/show/NCT03483116)[31] (Fig. 1).

The participant characteristics were similar in the per-protocol population of the RV3-BB clinical trial and in the microbiome study in both Malawi and Indonesia cohorts (Supplementary Table 1). The participant characteristics were also similar across the Indonesia and Malawi microbiome cohorts except for exclusive breastfeeding to 18 weeks (Malawi 98.4% compared to Indonesia 26.0%) and mode of delivery (vaginal delivery: Malawi 86% compared to Indonesia 95.9%) (Table 1). In the Indonesia microbiome cohort, 54 of 193 (28%) participants reported at least one episode of acute gastroenteritis during the 18-week study, whereas, in the Malawi cohort, only 26 of 186 (14%) participants reported an episode during the study period (Table 1). Exposure to antibiotics in the first 18 weeks of life was similar in the Malawi and Indonesia cohorts (Indonesia cohort: 45 of 193 [23.3%]; Malawi cohort: 55 of 186 [29.6%]). However, in the Malawi cohort, 18 of 186 participants (9.6%) received an antibiotic within 7 days of the stool collection compared to only one of 193 participants (0.5%) in the Indonesia cohort.

After administration of the investigational product (vaccine or placebo), stool samples were collected on days 1 to 7 in Indonesia and days 3–5 in Malawi (Fig. 1). The extracted bacterial 16S RNA was subjected to bacterial 16S variable region 3 and 4 sequencing. From the 65 million Malawi and 55 million Indonesia bacterial 16S reads, 66% and 67%, respectively, were assessed as high-quality reads using the MOTHUR pipeline. The oligotyping pipeline generated 4600 and 5563 oligotypes (OTPs), respectively, for the Malawi and Indonesia cohorts and yielded 108 and 96 bacterial taxa, respectively (Supplementary Table 2 and Supplementary Data 1).

### Vaccine response in association with alpha and beta diversity

Higher alpha diversity and differences in beta diversity of the early gut microbiome was observed in participants who had positive vaccine take and/or vaccine virus stool shedding compared to no vaccine take and/or shedding after receiving RV3-BB in the neonatal schedule, with greater differences observed in Malawi compared to Indonesia. These differences were not observed in the infant schedule group in Indonesia or Malawi (except for a measure of beta diversity and shedding at week 14 in Malawi; Table 2). No significant differences in alpha and beta diversity were observed in association with IgA seroconversion in the neonatal or infant schedule groups in the Indonesia or Malawi cohorts (Table 2 and Supplementary Table 3). Further description below focuses on analyses with significant diversity differences observed in association with vaccine take and stool shedding in the neonatal schedule group.

In the Malawi cohort, we found no differences in alpha or beta diversity observed at baseline (pre-vaccine) and vaccine response (vaccine take, IgA seroconversion or stool shedding 3 to 5 days) after

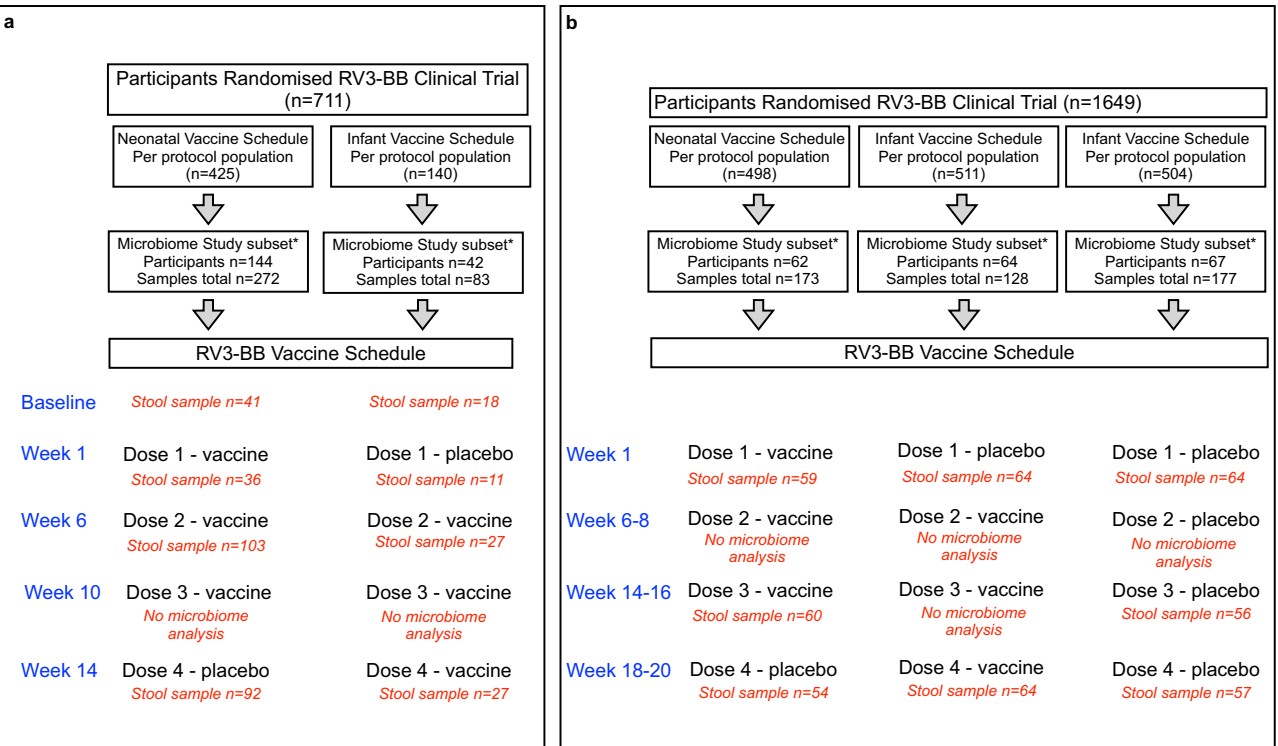

**Fig. 1 | RV3-BB Microbiome Study Participant Flow and Vaccine Schedule Flow.**
**a** The Malawi neonatal and infant vaccine schedule groups and the number and timing of sample collection for the vaccine and placebo doses. **b** The Indonesia neonatal and infant vaccine schedule groups and placebo group and the number and timing of sample collection for the different vaccine and placebo doses.

administration of dose 1 of RV3-BB vaccine (Table 2). Fisher's alpha index and the Richness measure were consistently higher in participants in the neonatal schedule group with a positive vaccine take and/or stool shedding (Table 2 and Fig. 2). At week 1, high alpha diversity (Richness measure) was associated with a positive vaccine take and stool shedding after administration of vaccine doses 1 and 2 in the neonatal schedule group (Table 2 and Fig. 2b). This association was even stronger at week 6 when high alpha diversity was also associated with positive vaccine take and stool shedding after vaccine dose 2 and dose 4* (d4* refers to three doses of vaccine and one dose of placebo), (Table 2 and Fig. 2c: Fisher's alpha index: dose 2 vaccine take $p = 0.0117$ and shedding 0.0039; dose 4* vaccine take $p = 0.0345$ and shedding $p = 0.0006$; Richness measure: dose 2 vaccine take $p = 0.0006$ and shedding $p = 0.0003$; dose 4* vaccine take $p = 0.0048$ and shedding $p = 0.0003$). Similarly, in Indonesia in the neonatal schedule group, high alpha diversity and differences in beta diversity was observed in participants with vaccine virus shedding at week 14 (Table 2 and Fig. 3b). Additional data and alpha diversity box plot analyses for the Indonesia study cohort are shown in Supplementary Fig. 1 and Supplementary Table 4.

As we only observed differences in alpha diversity and beta diversity in association with vaccine response in the neonatal schedule group and not the infant schedule group in both Malawi and Indonesia, we conducted a cross-analysis of alpha diversity detected at week 14 and 18 between participants in the neonatal schedule group and placebo group following three doses of vaccine from the Indonesia study (Fig. 3d, e and Supplementary Table 5). At week 14, the Fisher's alpha index and Observed richness measure were significantly higher in the neonatal schedule group with positive vaccine virus shedding compared to the placebo group at vaccine dose 3 (Fisher's index $p = 0.0027$, Observed Richness $p = 0.0016$) and at dose 4* (Fisher's alpha index $p = 0.0025$, Observed Richness measure $p = 0.0015$). No significant differences

in alpha diversity were observed at week 18, or for differences in beta diversity at week 14 and 18.

As most differences in beta diversity were observed in the neonatal schedule group, this was further explored using the Principal Coordinate Analysis (PCoA) (Fig. 4a, b for Malawi and Fig. 4c for Indonesia). In Malawi, distinct bacterial clusters were identified at week 6 between participants with positive or negative vaccine take (Fig. 4a) and between participants with positive or negative stool shedding (Fig. 4b) following administration of the vaccine at dose 2 and at dose 4*. In Indonesia, separation of bacterial clusters was observed between participants with positive or negative stool shedding at dose 3 in the neonatal schedule group (Fig. 4c). There was no significant association observed for positive or negative IgA seroconversion and beta diversity at week 6 or week 14 (Table 2).

## Age, breast feeding, mode of delivery, birth weight and antibiotic exposure
Alpha diversity increased with increasing age in both the Malawi and Indonesia cohorts using the Fisher's alpha index and Richness measure, but not the Simpson index. In Malawi, the Fisher's alpha index increased from baseline to week 1 (p = 0.0239), baseline to week 6 ($p < 0.0001$), and baseline to week 14 ($p = 0.0002$), and in Indonesia, from week 1 to week 14 ($p < 0.0001$) and week 1 to week 18 ($p < 0.0001$) (Fig. 5a, d). The Richness measure also increased with increasing age for Malawi, both baseline to week 6 ($p = 0.0002$) and baseline to week 14 ($p = 0.0002$); and Indonesia, both week 1 to week 14 [$p < 0.0001$] and week 1 to week 18 ($p < 0.0001$) (Fig. 5c, f). In Malawi, no significant difference in the change in alpha diversity with increasing age was observed between the infant and neonatal schedule group when analysed using Fisher's alpha index or Richness measure by timepoints, independent of vaccine response (Fig. 5g, i). The Simpson index identified a difference at week 6 with a higher alpha diversity ($p = 0.012$) in the Malawi neonatal schedule group, when participants

**Table 1 | Participant Characteristics of the RV3-BB Microbiome Study Cohorts**

| Participant variables | Malawi Microbiome Study | Indonesia Microbiome Study |
|---|---|---|
| Number of participants | 186 | 193 |
| Number of microbiome samples | 355 | 478 |
| Gender | | |
| Male: n (%) | 96 (51.6%) | 101 (52.3%) |
| Gestational age | | |
| Mean weeks (SD) | 37.5 (1.121)[a] | 39.53 (1.052) |
| Birth weight | | |
| Mean grams (SD) | 3069 (338.5) | 3091 (339.85) |
| Mode of delivery | | |
| Vaginal birth: n (%) | 160 (86%) | 185 (95.9%) |
| Age at first dose of vaccine or placebo | | |
| Mean number of days (SD) | 1.34 (1.359) | 3.141 (3) |
| Exclusive breastfeeding to 18 weeks | | |
| n (%) | 183 (98.4%) | 50 (25.95%) |
| Antibiotic exposure | | |
| Within 18 weeks of life: n (%) | 55 (29.6%) | 45 (23.3%) |
| Within 7 days of stool collection: n (%) | 18 (9.6%) | 1 (0.5%) |
| Episode of acute gastroenteritis | | |
| Within 18 weeks of life: n (%) | 26 (14%) | 54 (28%) |
| Within 7 days of stool collection: n (%) | 3 (1.6%) | 8 (4.1%) |

[a]Data available for $n = 184/186$ participants.

had received 2 doses of vaccine compared to one vaccine dose in the infant schedule group (Fig. 5h). As the Indonesia study design included a placebo group, we were able to compare the diversity in participants who received the RV3-BB with those who received placebo. We found no significant difference in alpha diversity between the neonatal schedule group (independent of vaccine response) and the placebo group at the key study time points (weeks 1,14, and 18) (Fig. 5j–l). In the infant schedule group, we observed a higher alpha diversity independent of vaccine response when compared to the placebo group at week 1 (Fisher's alpha index: $p = 0.0007$; Richness measure $p = 0.0108$) and at week 18 (Fisher's alpha index: $p = 0.0194$) Fig. 5j, l).

No significant differences in beta diversity were identified between the neonatal schedule group and infant schedule group at the key study time points in the Malawi study cohort (Supplementary Table 6 and Supplementary Fig. 2); or in the Indonesia study cohort between the neonatal schedule group, infant schedule group and placebo group, at the study timepoints (Supplementary Table 7). When analysed to account for important potential co-factors, differences in beta diversity were identified at week 1 and week 4 between exclusively breastfed infants and formula-fed infants in the placebo group in Indonesia ($p = 0.0421$ and $p = 0.0161$ respectively) but not in the vaccine groups (Supplementary Table 7). In the neonatal schedule group, differences in beta diversity were detected between stool collected at baseline and week 1 in relation to the mode of delivery (Malawi: caesarean section versus vaginal delivery [$p = 0.0004$], week 1 [$p = 0.0003$]; Indonesia: week 1 [$p = 0.0311$]), and for birth weight (Malawi: week 1 [$p = 00248$]; Indonesia: week 1 [$p = 0.0046$], and week 6 [$p = 0.0203$]) (Supplementary Tables 6 and 7). No significant differences were observed in association with gender or antibiotic use anytime during the study period in Malawi or Indonesia (Supplementary Fig. 3).

Beta diversity assessed using Principal Component Analysis (PCA) stratified by study time points identified two distinct age clusters in Malawi (a baseline/week 1 cluster and a week 6/14 cluster) (Fig. 6a) and similar two distinct clusters in Indonesia (week 1 cluster and a week 14/18 cluster) (Fig. 6b). In the Malawi study, the primary taxa responsible for driving age-related cluster separation along PC axis 1 was *Bifidobacterium* (loading value 0.852), while *Escherichia* was a driver of age-related cluster separation along the PC axis 2 (loading value 0.815). In the Indonesia study PCA analysis, *Bifidobacterium* was the primary taxa responsible for the age-related separation along PC axis 1 (loading value − 0.88287). The PC loading values are shown in Fig. 6 and additional PCA plots with overlayed Biplots visualising species driving age-related separations are shown in Supplementary Fig. 4.

**Bacterial taxa associated with RV3-BB vaccine response**

We identified bacterial taxa (Malawi: $n = 108$; Indonesia $n = 96$) with a minimum abundance of 0.01% that were then tested for an association with vaccine response (vaccine take, IgA seroconversion and stool shedding) using univariate and multivariate analysis by the MaAsLin2 statistical R package. Significant taxa defined using MaAsLin2 default significant Q-value of 0.25 are presented in Supplementary Data 2. At baseline, we observed no significant differences in bacterial taxa between vaccine response groups (positive or negative vaccine take; or positive or negative stool shedding; positive or negative IgA seroconversion) in Indonesia and in Malawi study cohort, except for *Enterococcus,* which was positively associated with vaccine virus shedding at week 14 in the neonatal schedule group in Malawi (coefficient = 15.6). All taxa with a minimum abundance of 1% positively or negatively associated with a vaccine variable with a coefficient value within the range of ± 1.5 for the RV3-BB and Rotarix® cohort are shown in Fig. 7.

*Bacteroides* was consistently associated with a positive vaccine take and stool shedding in the Malawi RV3-BB and in the Indonesia RV3-BB cohort (Fig. 7a, b). Further, in Malawi, a high abundance of *Bacteroides* was associated with a positive vaccine take and stool shedding following administration of the first vaccine dose, irrespective of whether the first dose was administered in the neonatal schedule (IP dose 1) or infant schedule (IP dose 2) (Fig. 7a). *Bacteroides* was also associated with positive shedding at later timepoints in the neonatal schedule group in Malawi (week 6 and shedding post-vaccine dose 2) and in Indonesia (week 14 and shedding post-vaccine dose 3) (Fig. 7a, b). Other bacteria identified in high abundance in Malawi RV3-BB samples in association with a positive vaccine response include *Parabacteroides* in the neonatal schedule group (week 1 and vaccine take and stool shedding post-vaccine dose 1) and *Bifidobacterium* in the infant schedule group (week 6 and shedding post-vaccine dose 1/IP dose 2). High abundance taxa associated with negative vaccine take included *Staphylococcus, Klebsiella, Prevotella* and *Escherichia*. *Staphylococcus* detected in high abundance at week 1 and week 6 was negatively associated with stool shedding in the neonatal schedule group in Malawi (week 1 and shedding post-vaccine dose 1; week 6 and shedding post-vaccine-dose 2). A high abundance of *Streptococcus* at week 6 was also negatively associated with vaccine take and stool shedding in the neonatal schedule group (post-vaccine dose 2) in Malawi, and at week 14 with stool shedding in the neonatal schedule group (post-vaccine dose 3) in Indonesia (Fig. 7a, b).

**Bacterial taxa associated with India and Malawi Rotarix® study vaccine response**

From the India Rotarix® study[22] we re-analysed data from 1145 samples (307 participants) yielding 140 bacterial taxa with a minimum abundance of 0.01%, and from the Malawi Rotarix® study[22] we re-analysed data from 283 samples (107 participants) yielding 137 bacterial taxa with a minimum abundance of 0.01% (Supplementary Table 2 and Supplementary Data 1). Participant characteristics for the Rotarix

**Table 2 | Alpha and Beta diversity at key study time points in association with IgA seroconversion, stool shedding, and vaccine take after administration of RV3-BB vaccine or placebo according to vaccine schedule groups in Malawi and Indonesia**

**Malawi RV3-BB study - Alpha and Beta diversity outcome for week1, week 6, and week 14**

| Week 1 microbiome analysis | Neonatal schedule group | | | Infant schedule group | | |
|---|---|---|---|---|---|---|
| | **Vaccine dose 1** | | | **Placebo dose 1** | | |
| | sIgA conversion | Vaccine take | Stool Shedding | sIgA conversion | Vaccine take | Stool Shedding |
| Participants with positive IgA seroconversion, vaccine take or stool shedding | Y 7 N 29 | Y 12 N 24 | Y 7 N 29 | Y 0 N 11 | Y 6 N 11 | Y 10 N 11 |
| Alpha diversity (P value) | | | | | | |
| Fisher's alpha index | NS | NS | 0.0348* | NA | NS | NS |
| Simpson index | NS | NS | NS | NA | NS | NS |
| Richness measure | NS | **0.0186*** | **0.0017*** | NA | NS | NS |
| Beta diversity: | | | | | | |
| R2 value | 0.019 | 0.063 | 0.101 | NA | 0.04 | 0.015 |
| *P*-value | NS | **0.0397*** | **0.0021*** | NA | NS | NS |
| **Week 6 microbiome analysis** | **Vaccine dose 2** | | | **Vaccine dose 1** | | |
| | sIgA conversion | Vaccine take | Stool Shedding | sIgA conversion | Vaccine take | Stool Shedding |
| Participants with positive IgA seroconversion, vaccine take or stool shedding | Y 29 N 74 | Y 71 N 32 | Y 57 N 46 | Y 4 N 23 | Y 16 N 11 | Y 13 N 14 |
| Alpha diversity (P value) | | | | | | |
| Fisher's alpha index | NS | **0.0117*** | **0.0039*** | NS | NS | NS |
| Simpson index | NS | NS | NS | NS | NS | NS |
| Richness measure | NS | **0.0006*** | **0.0003*** | NS | NS | NS |
| Beta diversity: | | | | | | |
| R2 value | 0.007 | 0.03 | 0.046 | 0.02 | 0.048 | 0.044 |
| *P*-value | NS | **0.0165*** | **0.0009*** | NS | NS | NS |
| Week 10 | Vaccine dose 3 | | | Vaccine dose 2 | | |
| | no microbiome analysis | | | no microbiome analysis | | |
| **Week 14 microbiome analysis** | **Placebo dose 4** | | | **Vaccine dose 3** | | |
| | sIgA conversion | Vaccine take | Stool Shedding | sIgA conversion | Vaccine take | Stool Shedding |
| Participants with positive IgA seroconversion, vaccine take or stool shedding | Y 48 N 44 | Y 73 N 19 | Y 52 N 40 | Y 13 N 14 | Y 25 N 2 | Y 21 N 6 |
| Alpha diversity (P value) | | | | | | |
| Fisher's alpha index | NS | NS | NS | NS | NS | NS |
| Simpson index | NS | NS | NS | NS | NS | NS |
| Richness measure | NS | NS | NS | NS | NS | NS |
| Beta diversity: | | | | | | |
| R2 value | 0.013 | 0.025 | 0.012 | 0.04 | 0.0245 | 0.135 |
| *P*-value | NS | **0.0397*** | NS | NS | NS | **0.0164*** |

**Indonesia RV3-BB study - Alpha and Beta diversity outcome for week 1, week 14–16, and week 18 -20**

| Week 1 microbiome analysis | Neonatal schedule group | | | Infant schedule group | | | Placebo group | | |
|---|---|---|---|---|---|---|---|---|---|
| | **Vaccine dose 1** | | | **Placebo dose 1** | | | **Placebo dose 1** | | |
| | sIgA conversion | Vaccine take | Stool Shedding | sIgA conversion | Vaccine take | Stool Shedding | sIgA conversion | Vaccine take | Stool Shedding |
| Participants with positive IgA seroconversion, vaccine take or stool shedding | Y 13 N 49 | Y 14 N 45 | Y 2 N 57 | Y 0 N 64 | Y 1 N 63 | Y 1 N 63 | Y 13 N 49 | Y 13 N 51 | Y 2 N 61 |
| Alpha diversity (P value) | | | | | | | | | |
| Fisher's alpha index | NS | NS | NS | NA | NA | NA | NS | NS | NS |
| Simpson index | NS | NS | NS | NA | NA | NA | NS | NS | NS |
| Richness measure | NS | NS | NS | NA | NA | NA | NS | NS | NS |
| Beta diversity: | | | | | | | | | |
| R2 value | 0.013 | 0.011 | 0.0173 | NA | 0.016 | 0.016 | 0.015 | 0.016 | 0.046 |
| *P*-value | NS | NS | NS | NA | NS | NS | NS | NS | NS |

**Table 2 (continued) | Alpha and Beta diversity at key study time points in association with IgA seroconversion, stool shedding, and vaccine take after administration of RV3-BB vaccine or placebo according to vaccine schedule groups in Malawi and Indonesia**

| Indonesia RV3-BB study - Alpha and Beta diversity outcome for week 1, week 14–16, and week 18 -20 | | | | | | | | | | | |
|---|---|---|---|---|---|---|---|---|---|---|---|
| **Week 1 microbiome analysis** | **Neonatal schedule group** | | | **Infant schedule group** | | | **Placebo group** | | | | |
| | **Vaccine dose 1** | | | **Placebo dose 1** | | | **Placebo dose 1** | | | | |
| | sIgA conversion | Vaccine take | Stool Shedding | sIgA conversion | Vaccine take | Stool Shedding | sIgA conversion | Vaccine take | Stool Shedding | | |
| Week 6-8 | Vaccine dose 2 no microbiome analysis | | | Vaccine dose 1 no microbiome analysis | | | Placebo dose 2 no microbiome analysis | | | | |
| **Week 14–16 microbiome analysis** | **Vaccine dose 3** | | | **Vaccine dose 2** | | | **Placebo dose 3** | | | | |
| | sIgA conversion | Vaccine take | Stool Shedding | **No microbiome analysis** | | | sIgA conversion | Vaccine take | Stool Shedding | | |
| Participants with positive IgA seroconversion, vaccine take or stool shedding | Y N | Y N | Y N | | | | Y N | Y N | Y N | | |
| | 44 16 | 57 3 | 41 19 | | | | 24 30 | 25 31 | 4 52 | | |
| Alpha diversity (P value) | | | | | | | | | | | |
| Fisher's alpha index | NS | NS | **0.0024*** | | | | NS | NS | NS | | |
| Simpson index | NS | NS | NS | | | | NS | NS | NS | | |
| Richness measure | NS | NS | **0.0018*** | | | | NS | NS | NS | | |
| Beta diversity: | | | | | | | | | | | |
| R2 value | 0.015 | 0.024 | 0.103 | | | | 0.021 | 0.009 | 0.022 | | |
| *P*-value | NS | NS | **0.0003*** | | | | NS | NS | NS | | |
| **Week 18–20 microbiome analysis** | **Placebo dose 4** | | | **Vaccine dose 3** | | | **Placebo dose 4** | | | | |
| | sIgA conversion | Vaccine take | Stool Shedding | sIgA conversion | Vaccine take | Stool Shedding | sIgA conversion | Vaccine take | Stool Shedding | | |
| Participants with positive IgA seroconversion, vaccine take or stool shedding | Y N | Y N | Y N | Y N | Y N | Y N | Y N | Y N | Y N | | |
| | 42 11 | 52 2 | 35 19 | 54 9 | 64 0 | 15 49 | 34 22 | 35 23 | 2 56 | | |
| Alpha diversity (P value) | | | | | | | | | | | |
| Fisher's alpha index | NS | NS | NS | NS | NA | NS | NS | NS | NS | | |
| Simpson index | NS | NS | NS | NS | NA | NS | NS | NS | NS | | |
| Richness measure | NS | NS | NS | NS | NA | NS | NS | NS | NS | | |
| Beta diversity: | | | | | | | | | | | |
| R2 value | 0.017 | 0.037 | 0.031 | 0.01 | 0.005 | 0.036 | 0.01 | 0.012 | 0.006 | | |
| *P*-value | NS | NS | NS | NS | NS | NS | NS | NS | NS | | |

Differences in alpha and beta diversity between positive (YES = Y) and negative (NO = N) vaccine responders (IgA seroconversion, vaccine take and stool shedding) were determined for three alpha diversity indices (Fisher's alpha index, Simpson index and Observed richness measure) and one beta diversity measure (permuted multivariate analysis of variance (PERMANOVA) test with Bray-Curtis measure). For alpha diversity, the p-value is given. For beta diversity, the R² and p values are reported.

*Benjamini-Hochberg adjusted significant p-values within 95% confidence intervals are shown in bold. Non-significant p-values are indicated as NS. NA = no statistical tests were performed for the following reasons: either no data were available for one of the groups, or there were not at least two values for one of the groups. Data were tested for Gaussian distribution using the Shapiro-Wilk and Kolmogorov-Smirnov tests. For normally distributed data, the two-tailed unpaired parametric $t$ test was used for statistical analysis. For non-normally distributed data, the two-tailed unpaired non-parametric Mann-Whitney test was used. Additional data on alpha diversity and beta diversity analysis at key study time points and vaccine/placebo doses are presented in Supplementary Table 4. sIgA = serum IgA antibody.

samples are shown in Supplementary Table 8. Significantly associated taxa from the MaAsLin2 analysis from the India and Malawi Rotarix® cohorts are shown in Supplementary Table 9. All taxa with a minimum abundance of 1% positively or negatively associated with a vaccine variable with a coefficient value within the range of ± 1.5 for are shown in Fig. 7c for the India cohort and in Fig. 7d for the Malawi cohort. In the India Rotarix® cohort, prior to administration of the first dose of vaccine (week 1 or 4), highly abundant taxa, *Escherichia*, was associated with positive stool shedding following Rotarix® dose 2 at week 4 and week 10. At week 6, *Klebsiella* was associated with positive stool shedding following Rotarix® dose 1. *Bifidobacteria* was negatively associated with shedding after dose 2. In the Malawi Rotarix® cohort, highly abundant taxa observed prior to vaccine administration at week 4 included *Collinsella*, which was associated with positive stool shedding after dose 1. After the first dose of Rotarix® at week 6, *Clostridium sensu stricto* and *Klebsiella* were associated with positive stool shedding.

## Bacterial taxa comparisons over time between the RV3-BB and Rotarix® study cohorts

We compared the prominent bacterial taxa development identified in samples obtained from the RV3-BB and Rotarix® study cohorts[22,31]. The study cohorts were compared using the best available matched time-points. For timepoint analysis of bacterial taxa, the data from the Malawi RV3-BB study was presented as a combined neonatal and infant schedule group as there were no significant differences in taxa between these groups by PERMANOVA and PCA plot analysis (Supplementary Fig. 2 and Supplementary Table 6). Specifically, Rotarix® study week 1 (pre-vaccine) was compared to RV3-BB study baseline (pre-vaccine), Rotarix® study week 6 was compared to RV3-BB study week 6, and Rotarix® study week 10 was compared to RV3-BB study week 14–18.

The comparison between data from Malawi RV3-BB and the Malawi Rotarix® cohorts was of particular interest as participants for both studies were recruited from similar sites and similar populations

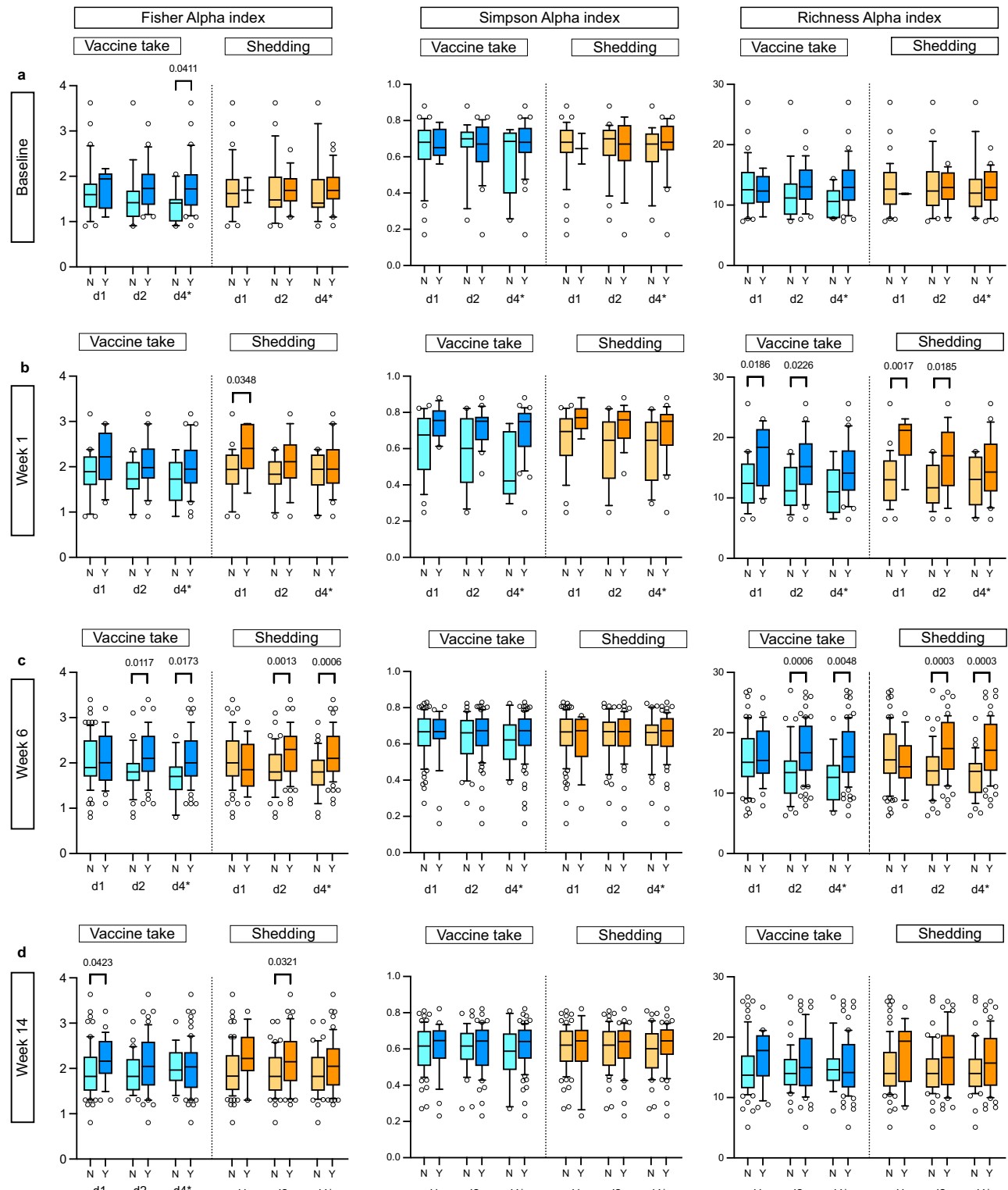

**Fig. 2 | Alpha diversity analysis for the RV3-BB Malawi participants in the Neonatal vaccine schedule group.** The alpha diversity was analysed between the negative vaccine response group (N, light blue colour for vaccine take and light orange for shedding) and the positive vaccine response group (Y, dark blue colour for vaccine take and dark orange for shedding) for the vaccine variables "Vaccine take" and "Shedding". The data are presented for the baseline group in (**a**), for the week 1 group in (**b**), for the week group 6 in (**c**), and for the week 14 group in (**d**). The analysis included three distinct alpha diversity indexes: Fisher's index, Simpson's index, and observed richness measure. The analysis was conducted on three vaccine doses: dose 1 (d1), dose 2 (d2), and dose 4 (marked d4*as this time point is after three doses of vaccine and one dose of placebo). The data were tested for normal distribution using the Shapiro-Wilk and Kolmogorov-Smirnov tests. For normally distributed data, a two-tailed parametric unpaired *t* test was employed, whereas, for non-normally distributed data, a two-tailed non-parametric unpaired Mann-Whitney test was used. Data are presented in a box and whisker plot. The box extends from the 25th to the 75th percentile, and the line in the middle is plotted at the median. The whiskers represent the 10–90 percentiles. All data points outside the 10–90 percentile are shown All statistical tests and graph generation were conducted in GraphPad Prism 10 for macOS (v 10.3.0). All the individual numbers used for box plot generation are presented in Table 2 and Supplementary Table 4.

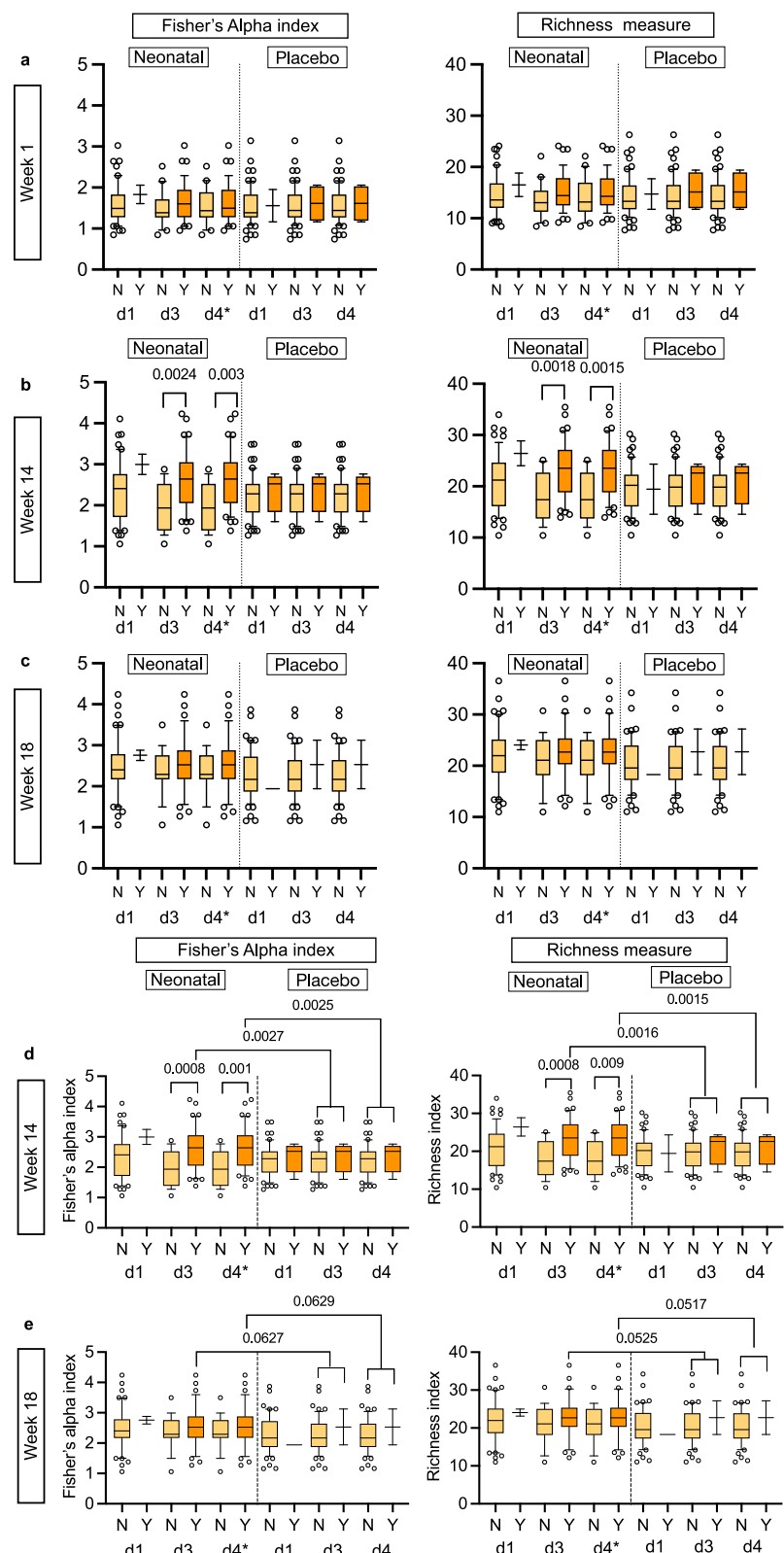

in Blantyre, although recruitment occurred over a consecutive time period (Rotarix® study: December 2015 to April 2018; RV3-BB study: September 2018 to January 2020). We explored the differences in bacterial taxa at the best-matched time points in the Malawi RV3-BB and Malawi Rotarix® studies and observed some similarities but also some differences in the abundance of prominent bacterial taxa. The abundance of *Bacteroides* decreased with increasing age in the RV3-BB

study cohort and was lower at week 14 when compared to the Rotarix® study cohort (*p* = 0.0065) (Fig. 8). *Bifidobacterium* abundance increased with increasing age in both the RV3-BB and Rotarix® study cohorts. A higher abundance of *Bifidobacterium* in the RV3-BB study cohort was observed at week 6 and week 14 compared to the Rotarix® study cohort (*p* = 0.0046 and *p* = 0.0006 respectively) however, there was a longer duration of exclusive breastfeeding in the RV3-BB study

**Fig. 3 | Alpha diversity analysis for the RV3-BB Indonesia participants in the Neonatal and Placebo vaccine schedule group.** Alpha diversity was analysed between the negative vaccine response group (N, light orange colour) and the positive vaccine response group (Y, dark orange colour) for the vaccine variables 'stool shedding' across three timepoints, (**a**) for week 1, (**b**) for week 14, and (**c**) for week 18. The analysis included three different alpha diversity indices: Fisher's index, Simpson's index and Richness measure. Fisher's index and observed richness measure are shown here. None of the Simpson's alpha indices were statistically significant between the vaccine groups and are not shown here. The results of Simpson's alpha index are shown in Table 2. The analysis was conducted on three vaccine doses: dose 1 (d1), dose 2 (d2), and dose 4* (d4* dose 4 is after three doses of vaccine and one dose of placebo). Alpha diversity – cross-analysis between the neonatal and placebo groups for shedding at week 14 shown in (**d**) and week 18 shown in (**e**) for the Indonesian cohort. The objective was to test for differences between the positive vaccine response for shedding in the neonatal vaccinated

group and the combined placebo-matched dosing group. Only those groups that exhibited a statistically significant difference between the vaccine response group and the non-vaccine response group with regard to shedding were subjected to further analysis through cross-validation. Data were tested for normal distribution using the Shapiro-Wilk and Kolmogorov-Smirnov tests. A two-tailed parametric unpaired $t$ test was used for normally distributed data, and a two-tailed non-parametric unpaired Mann-Whitney test was used for non-normally distributed data. The $P$-value across the three different time points was corrected using the Benjamin-Hochberg correction method. The uncorrected $P$-values are shown in Supplementary Table 4. Data are presented in a box and whisker plot. The box extends from the 25th to the 75th percentile, and the line in the middle is plotted at the median. The whiskers represent the 10–90 percentiles. All data points outside the 10–90 percentile are shown. All the individual numbers used for box plot generation are presented in Table 2 and Supplementary Table 4. All statistical tests and graph generation were performed in GraphPad Prism 10 for MacOS (v 10.3.0).

cohort (Fig. 8b). *Streptococcus* abundance also increased over time in the RV3-BB study cohort and was higher at weeks 6 and 14 when compared to the Rotarix® study cohort ($p = 0.0002$ both) (Fig. 8c). In the Rotarix® study cohort the abundance of *Veillonella* was higher at baseline, week 6 and week 10 compared to similar timepoints for the RV3-BB study cohort ($p = 0.0011$, $p = 0.0021$ and $p = 0.0185$ respectively) (Fig. 8d). *Staphylococcus* and *Escherichia* abundance decreased over time in both study cohorts with no significant differences in the pattern of decrease observed between the study cohorts (Fig. 8e, f). *Enterobacter*, non-further classified *Enterobacteriaceae*, and *Enterococcus* had higher abundances detected at baseline in the RV3-BB cohort, with some differences persisting at later timepoints (Fig. 8g–i).

## Discussion

Evaluating the role of the gut microbiome in explaining the difference in the effectiveness of oral rotavirus vaccines between high- and low-child mortality regions has been difficult to date. No significant association between gut microbiome alpha diversity and IgA seroconversion was reported in rotavirus vaccine studies in Ghana, Zimbabwe, Spain or Nicaragua, whereas a higher alpha diversity was observed in participants without IgA seroconversion was reported in studies in India and Malawi[16–22]. The newborn gut differs structurally and functionally from the mature gut[23–25]. The first weeks after birth are also acknowledged as a critical period in the development of the gut microbiome as the newborn responds to the ex-utero environment. Birth may provide a window of opportunity to target the first dose of a live, oral rotavirus vaccine, with the aim to improve vaccine uptake and provide early protection from severe rotavirus disease. Here we report that vaccine take and/or stool shedding was associated with a high alpha diversity, differences in beta diversity and bacterial taxa in the early gut microbiome when the human neonatal rotavirus vaccine (RV3-BB) was administered in the neonatal schedule in Malawi and Indonesia, but not in the infant schedule (Table 2).

Our results suggest the timing of the first dose of RV3-BB is important to optimise vaccine response. Although we found no differences in alpha diversity observed at baseline (pre-vaccine) and vaccine response (vaccine take, IgA seroconversion or stool shedding 3 to 5 days) after a birth dose of RV3-BB, higher alpha diversity observed at early time points (baseline and week 1) was associated with positive vaccine take and/or shedding at later timepoints (post-vaccine dose 2 and post-vaccine dose 3/placebo dose [IP dose 4*]) in Malawi. Positive vaccine take and shedding was also associated with higher alpha diversity in stool collected 3 to 5 days after the time-matched dose of RV3-BB (vaccine dose 1 and microbiome week 1; and vaccine dose 2 and microbiome week 6). However, this association was only observed in the neonatal schedule group. Our results when RV3-BB was administered in the infant schedule group are consistent with studies using the Rotarix® and RotaTeq® vaccines, also administered in the routine infant schedule[16–21] (Supplementary Data 3). We measured vaccine

response using a combination of vaccine take, IgA seroconversion and vaccine virus shedding in the stool[31,32]. This acknowledges that serum IgA is not a mechanistic correlate of protection for rotavirus infection. As maternal serum IgA is not transferred across the placenta, newborns are relatively IgA deficient at birth. For this reason, IgA seroconversion or serum IgA titre may not be a reliable marker to assess the vaccine response of a rotavirus vaccine administered at birth. Although not validated as a correlate of protection, vaccine-virus shedding in the stool provides a valuable insight, at the gut level, of the physiological response to a live virus, orally administered vaccine and the potential impact of the gut microbiome on the rotavirus vaccine response.

As expected, alpha diversity increased with increasing age across all treatment allocation groups in the Malawi and Indonesia RV3-BB cohorts. Consistent with previous studies, we also observed a significant impact on beta diversity by the mode of delivery, birth weight and duration of breastfeeding, although these differences were only observed at early study time points (baseline, week 1 and week 6) and were not sustained at 14 or 18 weeks of age[39,40]. Antibiotics administered prior to delivery of a rotavirus vaccine in adult volunteers increased vaccine response provided proof of principle that the gut microbiome may impact vaccine response[41]. Exposure to antibiotics in early life was associated with increased IgA seropositivity at 7 months of age in Brazil, Peru and South Africa[42]. However, we did not observe a significant difference in beta diversity of the gut microbiome in infants who had been exposed to antibiotics during the 14-to-18-week study period compared to those who had not received antibiotics.

Compared to the placebo group, participants receiving three doses of RV3-BB vaccine in the neonatal schedule have higher alpha diversity, although beta diversity and abundant bacterial taxa are similar. But, we observed a significant association, positive and negative, between high abundance bacterial taxa and vaccine response to the RV3-BB vaccine. *Bacteroides* was consistently associated with a positive vaccine response to the RV3-BB vaccine. *Bacteroides* species are considered beneficial taxa contributing to the metabolism of polysaccharides and oligosaccharides and facilitating the provision of nutrients to the host and luminal bacteria[40]. In Ghana, the abundance of *Bacteroides* was reported in participants with no evidence of a vaccine response following Rotarix® vaccine administration whereas, in Zimbabwe, *Bacteroides thetaiomicron* was associated with serum IgA titer but not with IgA seroconversion[16,18]. In Nicaragua, a difference in the abundance of *Bacteroides* was observed between RotaTeq® vaccine sero-responders and non-responders[19]. A high abundance of the facultative anaerobes, *Streptococcus, Staphylococcus, Klebsiella* and *Enterbacteriaceae*, were negatively associated with vaccine response in our RV3-BB vaccine study in Malawi and Indonesia. This contrasts with the study in Ghana, where a positive association between the abundance of *Streptococcus bovis* and Rotarix® vaccine IgA seroconversion was reported[16]. The decrease in *Staphylococcus* abundance observed

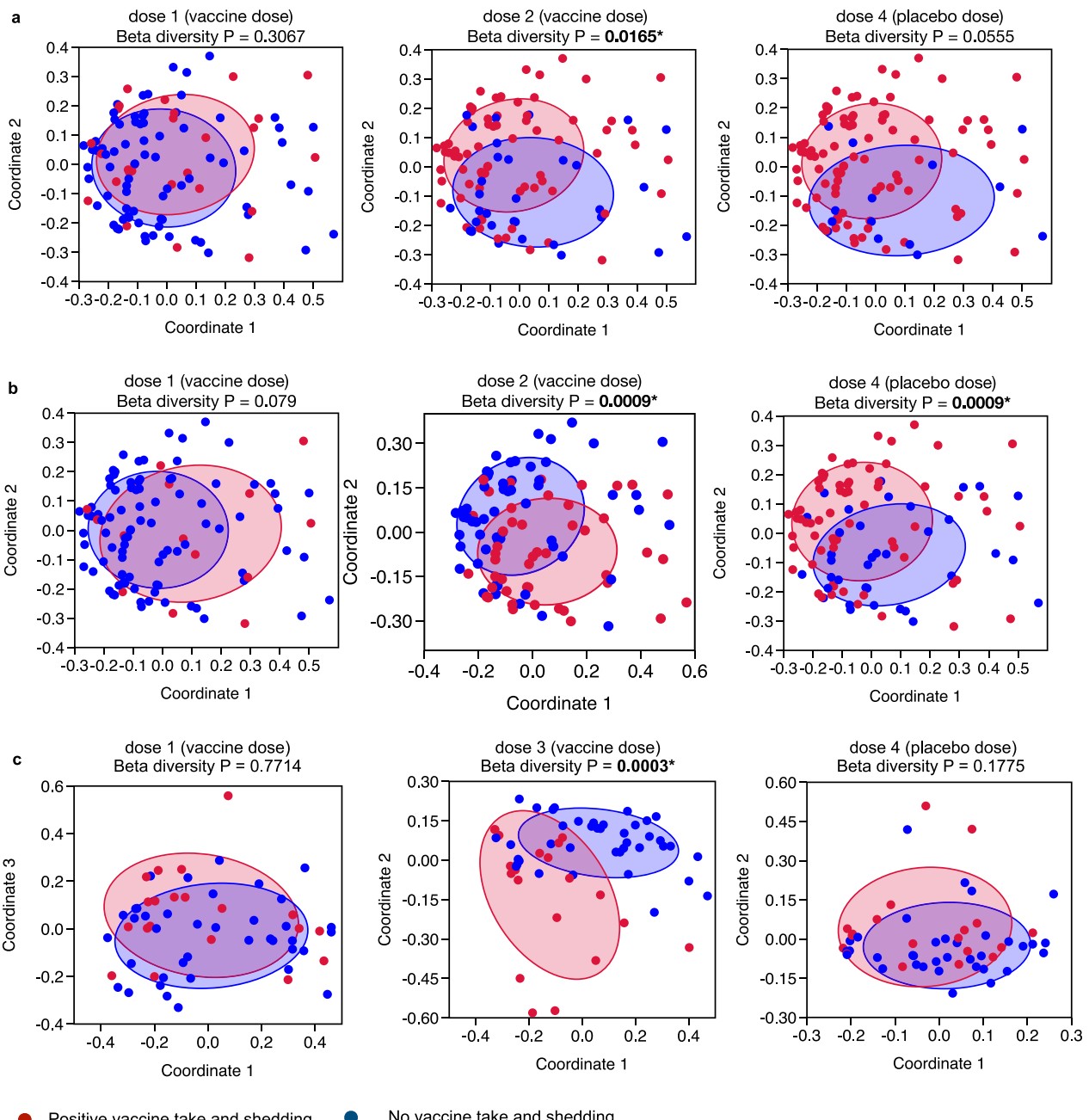

● Positive vaccine take and shedding     ● No vaccine take and shedding

**Fig. 4 | Principal co-ordinate analysis (PCoA) between the positive and negative vaccine take group and stool shedding group in the neonatal vaccine schedule group in RV3-BB Malawi and Indonesia study cohorts.** PCoA based on the Bray-Curtis distance matrix performed in the Malawi study neonatal vaccine schedule group at week 6 showed a distinct bacterial cluster between participants with positive and negative vaccine take (**a**) and positive or negative stool shedding (**b**) following administration of vaccine at dose 2 and at dose 4 (following three doses of vaccine and one dose of placebo [Fig. 1]). **c** In the Indonesia neonatal vaccine schedule group, the most pronounced bacterial clusters were observed for stool shedding at dose 3 and, to a lesser extent, at dose 4. The PCoA was performed using the Palaeontological Statistic software package for training and data analysis (v PAST 4.04) with 9999 permutations on Total Sum Scaling (TSS)-transformed data. The confidence ellipse was plotted at 50%. The FDR corrected significant *P*-value from the one-sided beta-diversity Permutational Multivariate Analysis of Variance test has been added to each PCoA plot. Since the test only evaluates whether the observed pseudo-F is greater than expected under the null, it is inherently one-sided. Adonis PERMANOVA test, commonly implemented in R's vegan package, is a one-sided test by default.

from week 1 to week 14 in our RV3-BB vaccine study in Malawi is consistent with patterns of colonisation in early life[43]. *E. coli* abundance was similar in the Malawi and Indonesia RV3-BB vaccine study cohorts, although an association with the lack of a vaccine response was only observed in the Indonesian cohort. This finding conflicts with a study from Pakistan where increased *E. coli* abundance was associated with a positive vaccine response to Rotarix® vaccine[17]. Although *Streptococcus, Staphylococcus, Enterococcus, E. coli* and *Klebsiella* often

colonise the neonatal microbiome, these bacteria are also associated with neonatal mortality, particularly in infants in LMICs[43]. Whether a luminal environment that promotes an abundance of obligatory anaerobes (*Bacteroides* and *Bifidobacterium*) reflects an environment that is more likely to be associated with vaccine uptake and replication, or whether a high abundance of facultative anaerobes (*Streptococcus, Staphylococcus, Klebsiella* and *Enterbacteriaceae*) challenges vaccine uptake and replication is of worthy of further exploration.

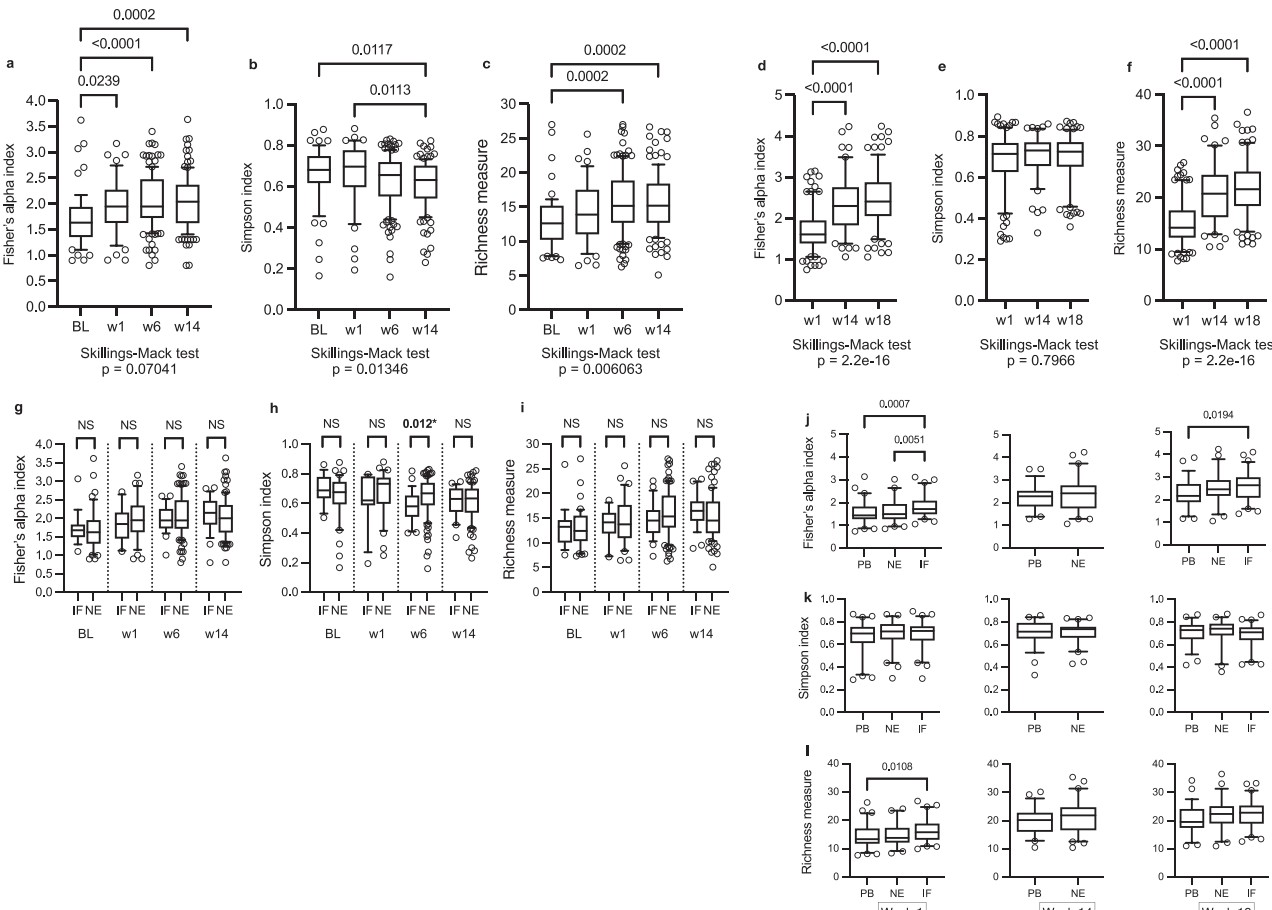

**Fig. 5 | Increased alpha diversity over time in the RV3-BB Malawi and Indonesia study cohorts.** **a**–**c** Alpha diversity using the Fisher's alpha index, Simpson's index and Observed Richness were compared between time points for the Malawi, and the Indonesia cohort (**d**–**f**), independent of vaccine response. **g**–**i** Fisher's alpha index, Simpson's index and Observed Richness analysis for the Malawi cohort separated by time points. **j**–**l** Fisher's alpha index, Simpson's index and Observed Richness analysis for the Indonesia cohort separated by time points. Data were tested for Gaussian distribution using the Anderson-Darling and Shapiro-Wilk tests. Data were not normally distributed, and therefore, the two-sided non-parametric Skillings-Mack test for incompletely matched data with Dunn's multiple comparisons test was used for analysis. Only *p*-values < 0.05 are shown for this time-point

analysis. For statistical analysis in (**g**–**l**), the two-tailed non-parametric Mann-Whitney test was applied. Data are presented in a box and whisker plot. The box extends from the 25th to the 75th percentile, and the line in the middle is plotted at the median. The whiskers represent the 10–90 percentiles. All data points outside the 10–90 percentile are shown. Statistical analysis and plotting were performed in Prism 10 for MacOS. The Skillings-Mack test was performed in R using the PMCMR plus package (v1.9.10). All the individual numbers used for box plot generation are presented in Table 2 and Supplementary Table 4. Abbreviations: BL = baseline; w = week; IF = infant vaccine schedule group; NE = neonatal vaccine schedule group; PB = placebo group.

Understanding this dynamic is important not only to understand the impact of the microbiome on rotavirus vaccine response but also for efforts to address the high mortality and morbidity associated with late-onset neonatal sepsis. Late-onset neonatal sepsis has been associated with an altered microbiome and reduced alpha diversity, and the causative bacteria frequently colonise the gut prior to emerging as a pathogenic infection[30,44].

A shared limitation of this study and previous studies is that the gut microbiome is influenced by multiple factors, including environmental, social and genetic factors[22,28]. Study methodology, age at sample collection, limitation in sample numbers within subgroups, intrinsic differences between vaccines and vaccine schedules, variability in definitions of vaccine response and vaccine virus shedding makes it also challenging to compare data between studies and avoid the need for multiple testing[16–22]. We addressed this by re-analysing the data from the Rotarix® study from Malawi and India, noting that the location and some aspects of the location methodology were similar, although the definition of stool shedding differed between studies[22,31,32]. This comparison revealed striking differences in the association between highly abundant taxa with vaccine virus

shedding[22]. Whether this reflects the intrinsic differences between the vaccines (RV3-BB: G3P[6] vs. Rotarix®: G1P[8]), age at administration of the first dose (day 0-5 vs. week 6), geography or the definition of vaccine shedding could not be determined in this analysis. Ultimately the success of a rotavirus vaccine is measured by its ability to protect individuals from severe rotavirus disease, and serum immune response and stool shedding of vaccine virus are markers of the vaccine response and not a direct predictor of the level of protection provided by the vaccine. We acknowledge that the gut microbiome is complex and consists of not only bacteria but also viruses and other eukaryotes. It is hoped that future advances will enable a more comprehensive investigation of the interactions between the gut microbiome and oral vaccine responses. These advances should also overcome the intrinsic limitations using the 16S marker gene analysis compared to metagenomic sequencing from which functional gene differences between vaccine responders and non-vaccine responders could be explored.

Our study shows high alpha diversity, differences in beta diversity, and a high abundance of *Bacteroides* in the gut microbiome is associated with positive vaccine take and stool shedding following

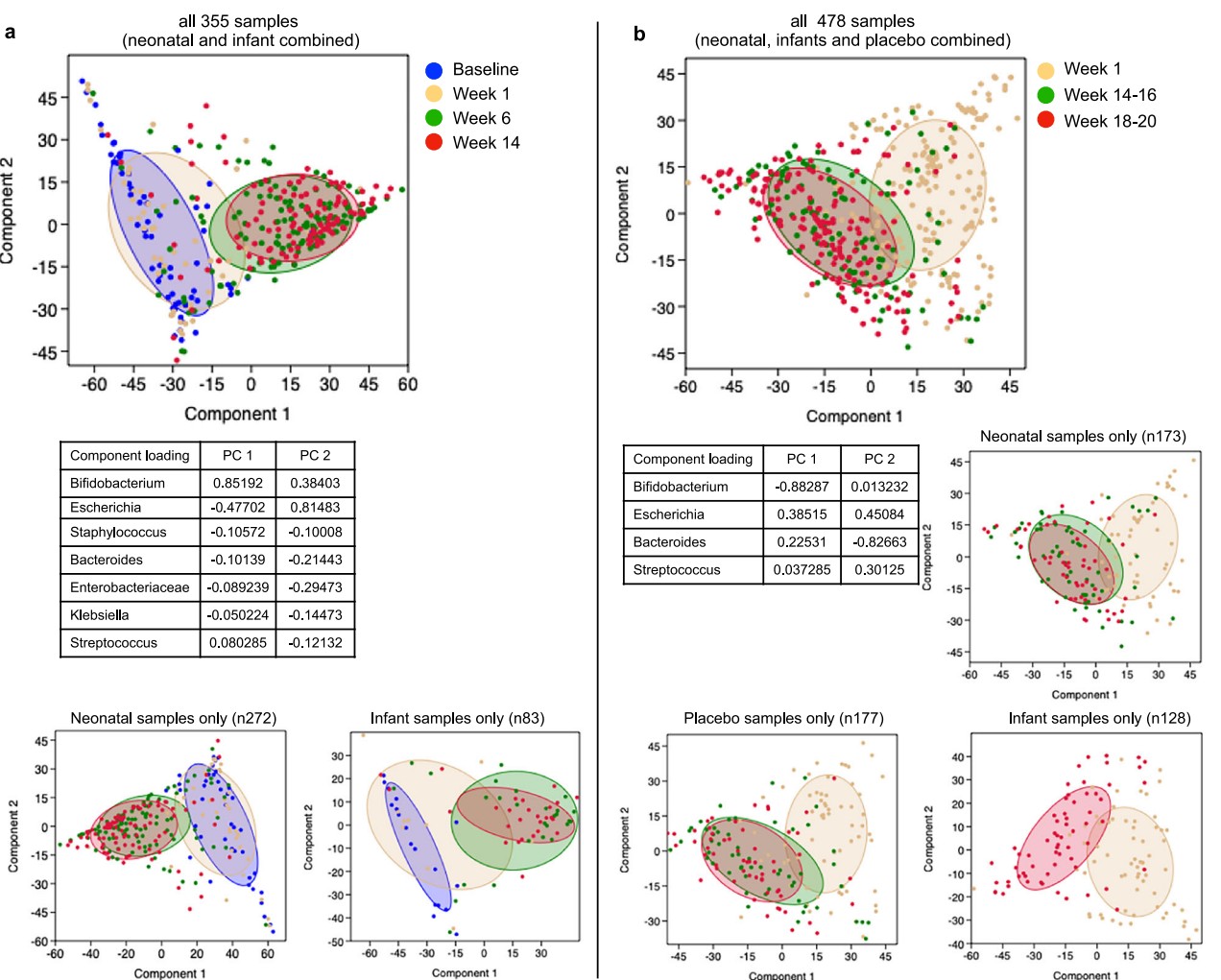

**Fig. 6 | Principal Component Analysis (PCA) at study timepoints (age groups) in the RV3-BB Malawi and Indonesia study cohorts. a** Describes the PCA of all 355 Malawi samples in the combined treatment group and further separated by neonatal and infant groups. **b** Describes the PCA of all 478 Indonesia samples in the combined treatment group and further separated by neonatal, placebo, and infant groups. Supplementary Fig. 4 presents the same data with a biplot overlay to illustrate the key taxa responsible for the observed cluster separation. The PCA was performed using the Palaeontological Statistic software package for training and data analysis (v PAST 4.04) with 9999 permutations on Total Sum Scaling (TSS)-transformed data. The confidence ellipse was plotted at 50%. The table inserted in Fig. 6 presents the PCA loading values for all taxa from the all-sample cohort analysis, with a minimum positive value of 0.1 and a minimum negative value of −0.1. A PCA loading value is equivalent to the coefficients of the taxa and provides information regarding the taxa that contribute the most to the components and, hence, the separation of the individual clusters. Loadings range from −1 to 1. A high absolute value (towards 1 or −1) indicates that the variable strongly influences the component, in our case age separation.

administration of RV3-BB in the neonatal schedule but not in the infant schedule or placebo groups. These data suggest that the early gut microbiome provides a gut environment that optimises the potential for a positive vaccine response after a dose of RV3-BB administered at that time point and at later doses. Participants who did not shed the vaccine virus were more likely to have a high abundance of *Streptococcus, Staphylococcus, Klebsiella* and *Enterococcus*, suggesting that facultative anaerobic bacteria colonising the gut can interfere with vaccine response. The gut microbial environment during the first days and week after birth provides a window of opportunity to target a birth dose of a live, oral rotavirus vaccine (RV3-BB) with the aim to optimise vaccine response and address the suboptimal protection currently observed in rotavirus vaccine programmes in LMICs.

## Methods
### Statistical analysis
All data are plotted in GraphPad Prism 9 for macOS and using PAST, The Palaeontological Statistics Software Package for education and data analysis (version 4.11)[45] for the construction of PCA and PCoA plots. Most of the data were plotted on a box-and-whisker plot with the plot extending from the 25th to the 75th percentile and the line in the middle of the box plotted at the median. The whiskers indicate the 10th to 90th percentile. All data points that fall outside the 10th to 90th percentile are displayed. Most statistical analyses are conducted using GraphPad Prism 9 for macOS, encompassing both two-group and multiple-group investigations as detailed in the respective figure legends. Most two-group analyses are conducted using the non-parametric, two-tailed Mann-Whitney test. In instances where the data were normally distributed, the two-tailed parametric unpaired $t$ test was employed. The data were subjected to a test for Gaussian distribution, employing the Anderson-Darling and the Shapiro-Wilk tests. If both tests failed to confirm a Gaussian distribution in the entirety of the sample set utilised for any given analysis, a non-parametric test was subsequently employed. All three and more group analyses were conducted using the non-parametric Kruskal-Wallis test with Dunn's multiple comparisons test. The data were found to be statistically significant at the 95% confidence level. For *P*-value correction for three

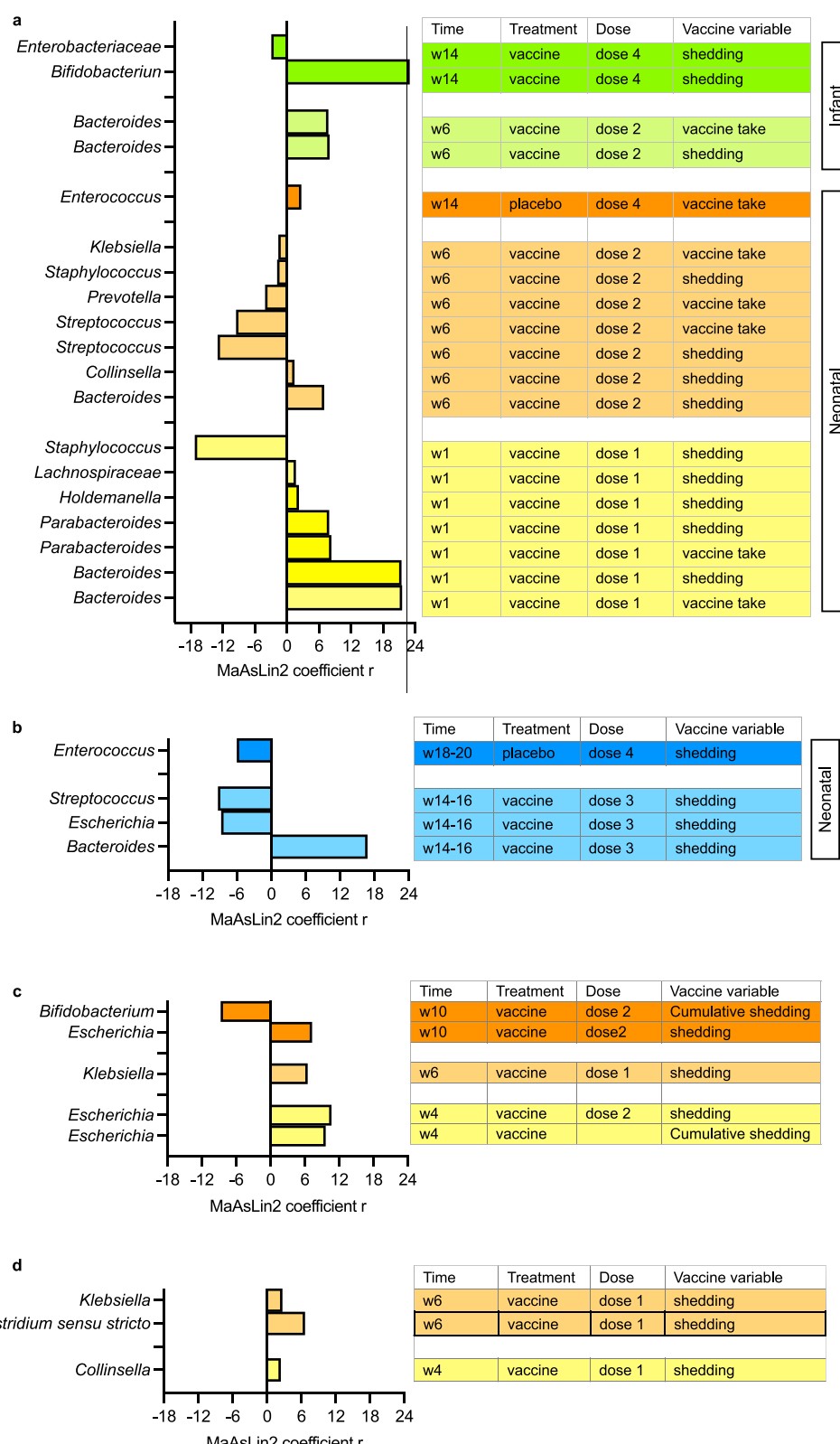

**Fig. 7 | Most abundant bacterial taxa significantly associated with the RV3-BB vaccine outcome (Malawi and Indonesia) and Rotarix vaccine outcome (Malawi and India).** The MaAsLin2-generated coefficient values from the most abundant bacterial taxa associated with vaccine outcome in the RV3-BB Malawi and Indonesia cohorts are shown in (**a** and **b**), respectively. The coefficient values from the most abundant bacterial taxa associated with vaccine outcome in the re-analysed Rotarix India and Malawi study cohorts are shown in (**c** and **d**), respectively. Figure 7 depicts all taxa with a minimum abundance of 1% and a coefficient value within the range of ± 1.5. The figure was generated using GraphPad Prism 10 (v10.2.1) and MAC Keynote (v13.1).

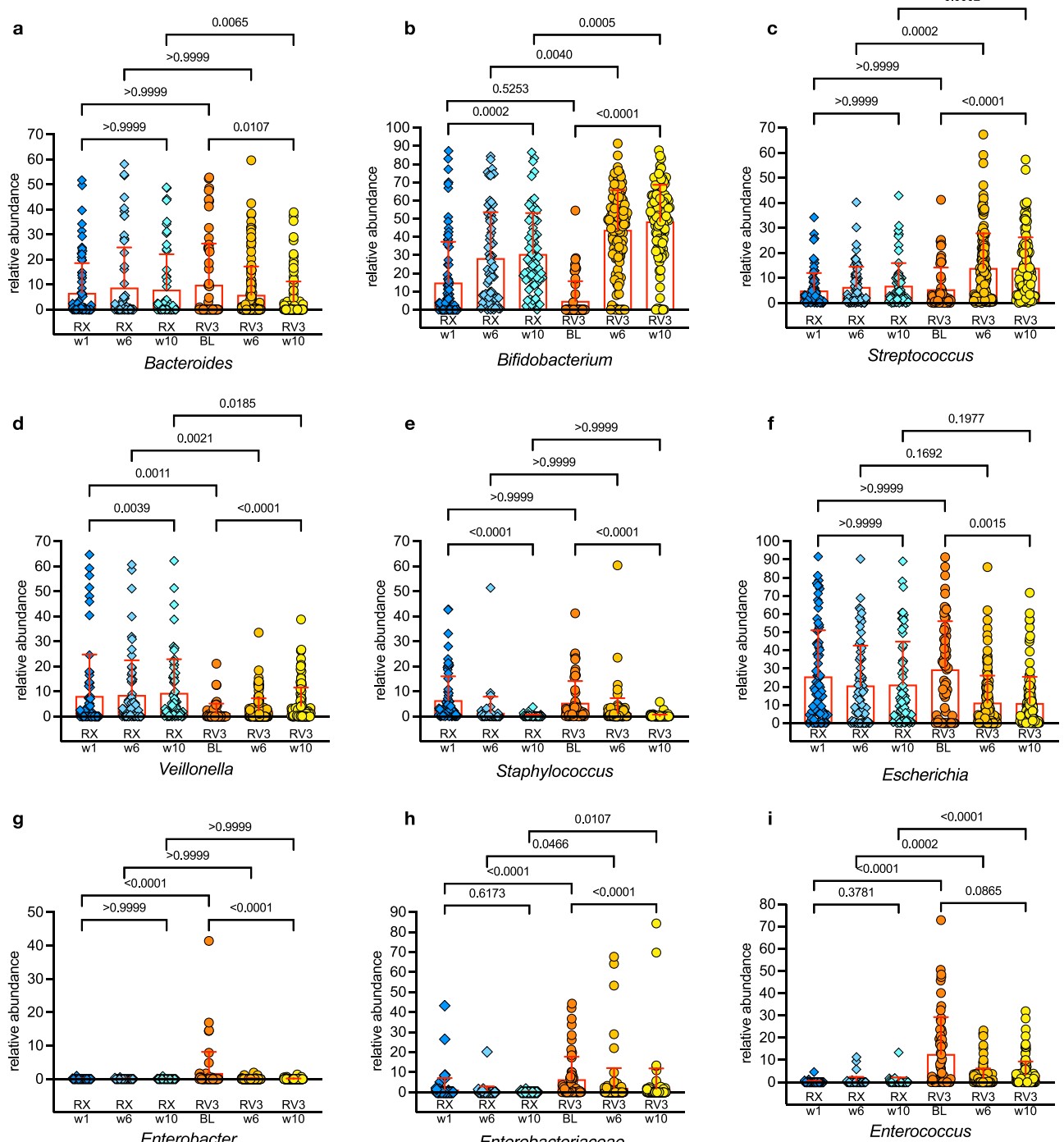

**Fig. 8 | Bacterial taxa differences between Rotarix and RV3-BB Studies conducted in Malawi presented according to key study timepoints.** Timepoint comparisons: Rotarix (RX) week 1 (w1) was compared to RV3-BB (RV3) baseline (BL), Rotarix week 6 (w6: dose 1) was compared to RV3-BB week 6 (w6: dose 2), and Rotarix week 10 (w10: dose 2) was compared to RV3-BB week 10 (w10: dose 4*: 3 doses of RV3-BB + 1 dose of placebo). In addition, the adjusted *P*-values for the following comparisons of interest between the first and last time points for each cohort (i.e., Rotarix week 1 versus week 10 and RV3-BB baseline versus week 10) are also shown. This analysis shows the similarities and differences for the nine most

abundant and/or implicated taxa (**a** *Bacteroides*, **b** *Bifidobacterium*, **c** *Strepotococcus*, **d** *Veillonella*, **e** *Staphylococcus*, **f** *Escherichia*, **g** *Enterobacter*, **h** *Enterobacteriaceae*, **i** *Enterococcus*). Statistical analysis was conducted using the Kruskal-Wallis test to determine significant differences in mean rank, along with Dunn's multiple comparisons test. The error bars show the mean with SD. The plotting and statistical analysis were performed using GraphPad Prism 10 for macOS (v10.2.0). The number of samples in the different time groups were: RX week 1 *n* = 81, RX week 6 *n* = 66, RX week 10 *n* = 61, RV3 baseline *n* = 59, RV3 week 6 *n* = 130, RV3 week 10 *n* = 119.

or more group comparisons we used the Prism default FDR method of Benjamini and Hochberg[46]. PERMANOVA tests were conducted in R using the adonis function from the R package VEGAN (v2.6-2) and in PAST 4.11 for multiple group comparison. Additionally, a multiple logistic regression model was constructed in GraphPad Prism 9 for

macOS to identify factors associated with non-vaccine responses. We employed multiple logistic regression modelling between all independent non-vaccine clinical/participant variables and a single dependent (outcome) vaccine variable (vaccine take, IgA seroconversion, stool shedding), in both univariate and multivariate analyses. The

multiple logistic regression modelling was limited to groups with the most significant PERMANOVA *P*-values.

The following software with version number has been used for analysis:

R - version 4.2.2 (2022-10-31)

RStudio - 2021.09.1 Build 372

Online web portal Calypso (v8.84)

PAST: Palaeontological Statistics software package for education and data analysis (v4.11)

MOTHUR (v1.45.2)

For bacterial 16S alignment, we used the Silva bacterial database "silva.nr_v138"

For chimera removal, we used "vsearch" (v2.16.0)

For oligotyping we used the "Minimum Entropy Decomposition" pipeline (v0.96)

For data sorting and table generation, we used Microsoft Excel for Mac (v16.71)

For data plotting, we used GraphPad Prism – version 9.5.1 and Mac Keynote (v12.2.1)

All statistically analysis was done in GraphPad Prism and using the R package MaAsLin2 (v1.5.1)

PERMANOVA was done in R using the "adonis" function from the R package VEGAN (v2.5.7

## Study design and sample size

The RV3-BB Microbiome study is an exploratory study of the RV3-BB rotavirus vaccine clinical trials, conducted in Indonesia (http://www.anzctr.org.au/Trial/Registration/TrialReview.aspx?ACTRN=12612001282875)[32] and in Malawi (ClinicalTrials.gov: NCT 03483116)[31]. The studies were conducted in accordance with the International Council for Harmonisation Good Clinical Practice Guidelines. The protocols were approved by the Ethics Committees of the Royal Children's Hospital, Melbourne, and the relevant Ethics Committees and Regulatory Boards in Indonesia (Universitas Gadjah Mada and the National Agency for Drug and Food Control, Republic of Indonesia) and Malawi (University of Liverpool, National Health Science Research Committee and the Pharmacy and Medicines and Poisons Board). A two-stage consent process was followed. Pregnant women provided consent for the collection of a pre-birth maternal blood and stool sample and an after-birth infant cord blood and stool sample. Participants were recruited irrespective to sex or gender. Written informed study consent was obtained after birth from parents or guardians, prior to confirming eligibility for enrolment.

The Indonesian RV3-BB phase IIb trial was a double-blind, randomised, parallel-group trial involving 1649 healthy newborns. It was conducted from January 2013 through July 2016 in primary health centres and hospitals in Central Java and Yogyakarta. The infants were randomly assigned in a 1:1:1 ratio to one of three groups: the neonatal schedule group, the infant schedule group, and a placebo group. The participants received four oral one-millilitre doses of the investigational product (IP), with doses administered at 0 to 5 days of age (IP dose 1), 8 to 10 weeks of age (IP dose 2), 14 to 16 weeks of age (IP dose 3) and 18 to 20 weeks of age (IP dose 4) (Fig. 1b). Each participant received three oral doses of RV3-BB and one dose of placebo. In the neonatal schedule group, doses 1, 2 and 3 were RV3-BB vaccine, while dose 4 was a placebo dose. In the infant schedule group, the first dose was a placebo dose, and the second, third, and fourth doses were the RV3-BB vaccine. The clinical trial lots of the RV3-BB vaccine were developed from the human neonatal strain RV3 (G3P[6]) to a titer of 8.3 to $8.7 \times 10^6$ focus forming units per millilitre (FFU/mL) in a serum-free medium that was supplemented with 10% sucrose. Placebo consisted of the same medium with 10% sucrose and was visually indistinguishable from RV3-BB. The primary objective of this Phase IIb trial was to evaluate the efficacy of three doses of RV3-BB vaccine in preventing severe rotavirus gastroenteritis with secondary objectives to assess the safety and immunogenicity of the RV3-BB vaccine[32]. More details on the Indonesia clinical trial can be found at http://www.ncbi.nlm.nih.gov/pmc/articles/pmc5774175/ (DOI: 10.1056/NEJMoa1706804). In brief, for the primary efficacy endpoint, the sample size was calculated on the basis of local surveillance data, the assumption of 3% of the participants in the placebo group would have an episode of severe rotavirus gastroenteritis during the trial, and we calculated that an enrolment target of 549 participants in each of the three trial groups would provide the trial with 80% power to reject the null hypothesis of no difference between the combined vaccine group and the placebo group if the true efficacy of the vaccine was 60%, at a one-sided alpha level of 0.1. The estimated sample size would allow for a rate of nonadherence to the trial regimen of 10%. For the immunogenicity endpoints, we calculated that a minimum of 282 participants would be required to reject the null hypothesis of no difference in the percentage of participants with vaccine response, at a two-sided alpha level of 0.05, assuming that 25% of participants in the placebo group would be exposed to rotavirus and that 50% of the participants in each of the two vaccine groups would have vaccine response, and allowing for a rate of nonadherence of 10%.

The Malawi RV3-BB trial was a phase II, randomised, double-blind, placebo-masked, dose-ranging study involving 711 participants[31]. It was conducted from September 2018 through January 2020 in three health centers in the Blantyre district (Ndirande, Bangwe, and Limbe) and Queen Elizabeth Central Hospital, Blantyre. Healthy infants less than 6 days of age with a birth weight of 2.5 to 4.0 kg, irrespective of in-utero exposure to human immunedeficiency virus, were eligible for enrolment. The infants who met the eligibility criteria were randomly assigned to one of four treatment arms. Three of the arms involved the neonatal schedule, with RV3-BB administered at different vaccine titre while the fourth arm was the infant schedule group administered RV3-BB at a vaccine titer of $1 \times 10^7$ FFU/mL (Fig. 1a). Each participant received three oral doses of RV3-BB vaccine and one dose of placebo. The clinical trial lots of the RV3-BB vaccine were developed from the human neonatal strain RV3-BB at different vaccine titre (high [$1.0 \times 10^7$ FFU/ml], mid [$3.0 \times 10^6$ FFU/mL], and low [$1.0 \times 10^6$ FFU/mL]) in a serum-free medium that was supplemented with 10% sucrose. Placebo consisted of the same medium with 10% sucrose and was visually indistinguishable from RV3-BB. The primary aim of this study was to determine if the RV3-BB vaccine was safe and immunogenic in infants in Malawi and to compare vaccine responses between participants receiving a lower titer of RV3-BB compared with the higher titer in the neonatal schedule[31]. Participants from the Indonesia R3-BB trial and Malawi RV3-BB trial were eligible for inclusion into the RV3-BB Microbiome study cohort if they had adequate stool volume available for microbiome analysis at key study time points (baseline, week 1, week 6–8, week 14–16 and week 18–20) in the per-protocol population (Indonesia $n = 193$; Malawi $n = 186$). For the Indonesia RV3-BB trial, microbiome data is presented for each of the three study arms: the neonatal schedule group, the infant schedule group and the placebo group. For the Malawi RV3-BB trial, microbiome data is presented for the combined neonatal vaccine group (all three vaccine titer groups) and the infant schedule group. Table 1 presents the characteristics of participants from the Indonesian and Malawi studies that were included in the RV3-BB Microbiome study. As reported in the primary publication (http://www.ncbi.nlm.nih.gov/pmc/articles/pmc9021029/; https://doi.org/10.1016/S1473-3099(21)00473-4), the sample size was calculated to demonstrate non-inferiority of the lower titre ($3.0 \times 10^6$ and $1.0 \times 10^6$ FFU per mL) vaccine groups with respect to the proportion of participants who have an IgA seroconversion 4 weeks after three doses of RV3-BB (active vaccine). It was estimated that 30% of participants would be excluded from the per-protocol population due to death, study

withdrawal, loss to follow-up, or study non-compliance and a 50% IgA seroconversion probability was assumed for the active controls. Based on a one-sided 0·025 level score test with 90% power under the alternative of no difference in response probabilities, 172 participants per group were required for a total sample size of 688 participants.

No formal sample size calculations for the microbiome analysis were completed. This study used available samples from other clinical studies, and the sample size was determined by the number of participant samples identified as available for use in future ethically approved research and the number of samples that were not exhausted during the analysis of the clinical trials.

## Sample collection, transport and storage

Stool samples were collected after administration of a dose of vaccine or placebo (Malawi cohort: baseline at 3 to 5 days, week 1, week 6, week 14; Indonesia cohort: 1 to 7 days at week 1, week 14–16, week 18–20) (Fig. 1). A pre-dose sample was collected prior to the administration of IP in the Malawi study. The samples were collected from the participants' homes or healthcare centres by field workers or sample transport workers and stored in a refrigerated environment at a temperature between 2 °C and 10 °C until transfer to the laboratory. The samples were frozen at − 80 °C within 24 h of their collection at the central study laboratory. The frozen stool samples were shipped on dry ice to the Enteric Diseases Laboratory at Murdoch Children's Research Institute (MCRI), Australia, for nucleic acid extraction, 16S amplification, sequencing and microbiome analysis. Microbiome analysis was conducted using samples collected 3 to 5 days after administration of a dose of vaccine or placebo. All laboratory data were collated and stored in a REDCap database. Blood samples were collected from the umbilical cord (serving as the baseline for neonatal schedule group comparisons) and from the infant immediately prior to IP dose 2 (serving as the baseline for infant schedule group comparisons). Subsequent blood samples were collected 28 days after IP dose 3 and 28 days after IP dose 4. The samples were spun to obtain serum which was then frozen at − 80 °C within 24 h of their collection at the central study laboratory. The frozen serum samples were shipped on dry ice to the Enteric Diseases Laboratory at MCRI.

## Rotavirus IgA seroconversion

Serum rotavirus IgA antibody titre were measured by ELISA using rabbit anti-RV3 polyclonal sera as the coating antibody and RV3-BB virus or vero cell lysate as the capture antigen[47]. The antigen-antibody complexes were detected with biotinylated anti-human IgA and streptavidin-horseradish peroxidase[47]. Anti-rotavirus IgA seroconversion was defined as a threefold or greater increase from baseline in blood collected four weeks after any dose of vaccine or placebo (neonatal schedule group: following IP dose 1 (vaccine dose 1), IP dose 2 (vaccine dose 2), IP dose 3, or IP dose 4 (vaccine dose 3 and one dose of placebo); infant schedule group: following the IP dose 1 (placebo dose), IP dose 2 (vaccine dose 1), IP dose 3 (vaccine does 2), or IP dose 4 (vaccine dose 3 and one dose of placebo).

## Stool vaccine virus shedding

RV3-BB vaccine-like shedding in stool was assessed in stools collected at key study time points (Fig. 1) (Malawi cohort: baseline, week 1, week 6, week 14; Indonesia cohort: week 1, week 14–16, week 18–20). Stool samples were analysed using a rotavirus VP6-specific reverse transcription-polymerase chain reaction (RT-PCR) assay. PCR products were then analysed by electrophoresis with the Invitrogen one-step RT-PCR key (Invitrogen, Carlsbad, CA, USA) and Rot3 and Rot5 oligonucleotide primers[48]. Sequence analysis was used to confirm the presence of RV3-BB vaccine (Sequencher Software programme version 4.1, Gene Codes Corp Inc., Ann Arbour, MI, USA) and identity

determined by the GenBank database[48]. Positive stool shedding was defined as positive detection of RV3-BB vaccine virus in any stool after administration a dose of vaccine or placebo (Malawi: day 3 to 5; Indonesia: day 3 to 7)[31,32].

## Definition of vaccine take

Vaccine take was defined as IgA seroconversion or stool shedding following administration of any dose of vaccine or placebo. Positive vaccine take was defined as evidence of vaccine take following IP dose 1 (vaccine dose 1), IP dose 2 (vaccine dose 2) or IP dose 4 (vaccine dose 3 and one dose of placebo) for the neonatal schedule group, and following the IP dose 1 (placebo dose) or IP dose 4 (vaccine dose 3 and on dose of placebo) for the infant schedule group (Fig. 1)[31,32].

## Definition of vaccine response

Positive vaccine response was defined as IgA seroconversion 28 days after administration of vaccine or placebo, stool shedding of RV3-BB detected after a dose of vaccine or placebo (Malawi: 3 to 5 days; Indonesia 1 to 7 days) and/or vaccine take at any study assessment timepoint (Fig. 1)[31,32].

## Bacterial 16S analysis pipeline

A total of 856 stool samples underwent bacterial 16S rRNA gene amplification and sequencing. Total DNA was extracted from 200 mg of stool using the Dneasy® PowerSoil® Pro Kit (QIAGEN, Hilden, Germany) in accordance with the manufacturer's instructions, with the following modifications. The samples were homogenised at 2000 rpm for 30 s, after which they were allowed to rest for 30 s. This process was then repeated, with the samples being homogenised at 2000 rpm for a further 30 s, using the Mini-Beadbeater (Biospec Products, Bartlesville, OK). The DNA was subsequently eluted in a solution of 55 μL of C6 medium. The eluted DNA was stored at a temperature of − 30 °C. Polymerase chain reaction (PCR), library preparation and sequencing were conducted at the Ramaciotti Centre for Genomics, University of New South Wales. Polymerase chain reaction (PCR) of the V3-V4 bacterial 16S region (314F-805R) was conducted in triplicate for each sample, with the pooled PCR products subsequently sequenced on an Illumina MiSeq using the MiSeq Reagent Kit v3 for 2 × 300 bp paired-end sequencing. The samples analysed included 478 from the RV3-BB Indonesia cohort and 355 from the RV3-BB Malawi cohort. Furthermore, five negative controls were incorporated into the Indonesian library preparation, while a single negative control was included in the Malawian library preparation.

The forward and reverse fastq files of each sample from the RV3-BB microbiome cohort were processed in accordance with the MOTHUR MiSeq standard operating procedure, with some modifications (MOTHUR wiki at http://www.mothur.org/wiki/MiSeq_SOP). MOTHUR version 4, build 1.45.2, was utilised[49]. The "make.contigs" command was executed with no additional parameters. In the second command step, "screen.seqs," any ambiguous sequences (maxambig = 0) and sequences containing homopolymers longer than 8 bp (maxhomop = 8) were removed. The quality-screened sequences were aligned using the Silva bacterial database (Silva.nr_v138.align), with the flip parameter set to true. The Silva databases are generated by the German Network for Bioinformatics Infrastructure (https://www.arb-silva.de). Any sequences that did not align within the expected coordinates were subsequently removed using the "screen.seqs" command. The alignment coordinates were set using the "optimise = start-end, criteria = 90" parameters. The correctly aligned sequences were filtered using the "filter.seqs" command with the parameters "vertical = T" and "trump = ." The subsequent filtered sequences were subjected to de-noising, allowing for three mismatches in the "pre-clustering" step. In addition, chimeras were removed using vsearch (v2.16.0) with the dereplicate option set to "true". The chimera-free sequences were subsequently classified using the Silva reference

database (silva.nr_v138.align) and the Silva taxonomy database (silva.nr_v138.tax), with a cut-off value of 80%. Sequences derived from chloroplasts, mitochondria, unidentified sources, Archaea, and Eukaryota were excluded from further analysis. From the 65 million Malawi and 55 million Indonesia bacterial 16S reads, 66% and 67%, respectively, were assessed as high-quality reads using the MOTHUR pipeline. The oligotyping pipeline generated 4600 and 5563 oligotypes (OTPs), respectively for the Malawi and Indonesia cohorts and yielded 108 and 96 bacterial taxa, respectively (Supplementary Table 2). Oligotyping was employed for the purpose of clustering the high-quality filtered FASTAs sequences derived from the MOTHUR pipeline. Oligotyping is a computational method used to investigate the diversity of closely related but distinct bacterial organisms in final operational taxonomic units identified in environmental datasets through 16S ribosomal RNA gene data using the canonical approaches. The "Minimum Entropy Decomposition" (MED) option was employed for the sensitive partitioning of high-throughput marker gene sequences from the Oligotyping pipeline[50m]. The normalised (sub-sampled) high-quality FASTAs and name files from MOTHUR were renamed by appending the group name to the sequence name, using the "rename.seqs" command. Subsequently, a redundant renamed-FASTA file was generated using the "deunique.seqs" command, which creates a redundant FASTA file from a FASTA and name file. Subsequently, the redundant Fasta file was employed for oligotyping via the unsupervised MED. The command line was "decompose <fasta.file> −g −t --M 100 -V 2". The −t character was set to a dash ("-") character. In the MOTHUR "rename.seqs" command, the dash character was employed to differentiate the sample name from the unique information displayed in the sequence name's delimiter. The -M integer defines the minimum substantive abundance of an oligotype, while the -V integer defines the maximum variation allowed in each node. The precise number of samples obtained and analysed at each stage of the oligotyping pipeline is detailed in Supplementary Table 2. Subsequently, the high-quality node sequences of each OTP were employed for taxonomic profiling using MOTHUR and the SILVA database, as detailed in section 3.1. The redundant oligotype node taxa were then consolidated into non-redundant phyla and genera. The consolidation was conducted using the "Consolidate" function in Microsoft Excel for Mac version 16.16.14. One negative control taxon, "*Pseudomonas*," was removed from the Indonesia RV3-BB cohort, and two negative control taxa, "*Roseibacillus*" and "*Vibrio*," were removed from the Malawi RV3-BB cohort. Furthermore, low-abundance taxa that were only detected in ≤ 3 samples and below 0.01% abundance were excluded from the dataset. This resulted in a total of 108 taxa from the MWI RV3-BB cohort, 96 taxa from the Indonesia RV3-BB cohort, 137 taxa from the Malawi Rotarix® cohort, and 140 taxa from the India Rotarix® cohort being available for analysis. Many of the taxa could be classified at the genus level. The complete list of taxa, along with their absolute and relative abundances across the four studies, is provided in Supplementary Table 2.

### Alpha diversity and vaccine response

This analysis was conducted at each microbiome stool collection timepoint and each dose of vaccine or placebo in each of the treatment allocation groups (neonatal schedule, infant schedule, and placebo groups) for the Malawi and Indonesia microbiome cohorts (Fig. 1) with an analysis according to vaccine response (positive or negative). The alpha diversity indexes were presented for Fischer's alpha index, Simpson index, and the observed richness measure calculated using the online web portal Calypso version 8.84 using Total Sum Scaling (TSS) transformation followed by Cumulative Sum Scaling (CSS) + log 2 normalisation, a widely used method for normalising microbial community compositional data[50–55]. The small number of abundant species and the considerable proportion of 'rare' species (the class containing one individual is consistently the largest) predicted by the log series model indicate that akin to the geometric series, it will be most pertinent in scenarios where one or a few factors exert a dominant influence on the ecology of a community. Therefore, we propose that Fisher's alpha index can be considered an ideal supplementary alpha diversity index to the commonly reported Simpson index for any bacterial 16S datasets, which are typically characterised by a few dominant taxa and a multitude of lower-abundant taxa. Two-group data were analysed using either a two-tailed parametric *t* test if they followed a Gaussian distribution or a two-tailed parametric Mann-Whitney test if they were not normally distributed. The non-parametric Kruskal-Wallis test with Dunn's multiple comparisons test was employed to analyse three groups of non-normally distributed data. In the case of matched timepoint analysis with three or more groups, the Skillings-Mack test for incompletely matched data with Dunn's multiple comparisons test was employed for analysis. The data were evaluated for Gaussian distribution using the Anderson-Darling and the Shapiro-Wilk tests, which are available in GraphPad Prism 9. If both tests yielded normally distributed data at the alpha level of 0.05 for all datasets within a given comparison, the parametric test was employed; otherwise, the non-parametric test was utilised. The data were represented on a box-and-whisker plot. The box plot always extends from the 25th to the 75th percentile, with the line in the middle of the box plotted at the median. The whiskers indicate the 10th to 90th percentile. All data points that fall outside the 10th to 90th percentile are displayed. The statistical analysis and plotting were conducted using Prism 9 for macOS (version 9.3.1) except for the Skillings-Mack test, which was analysed in R using the "skillingMackTest" function from the PMCMRplus package (v1.9.10).

### Beta diversity and vaccine response

Stool samples and analysis time points and groups are presented for the alpha diversity analysis.

A multivariate beta diversity analysis was conducted using the PERmutational Multivariate ANalysis Of VAriance (PERMANOVA) test in R, employing the "adonis" function from the R package VEGAN (v2.5-7). The Bray-Curtis similarity index was employed in conjunction with 9999 permutations. A statistically significant PERMANOVA test is reported, accompanied by an F-value, an R² value, and a P-value. The beta diversity analysis was conducted on data that had undergone a TSS transformation. All PERMANOVA tests were conducted with the appropriate single "vaccine variable," and then with all other available clinical and participant variables to test for confounding effects. To visualise the compositional differences between microbial communities associated with environmental and clinical variables, a multivariate method, Principal Coordinate Analysis (PCoA) and Principal Component analysis (PCA), was employed at the taxon level (genus). Beta diversity analysis was conducted on the TSS-transformed data. The PCoA and PCA was conducted using the PAST 4.11 software using the Bray-Curtis similarity index[45]. The concentrated data points in the figures are highlighted by a 50% or 60% concentration ellipse. The PERMANOVA test was initially conducted on all samples within the infant and neonatal schedule groups to identify any differences related to age or at key study time points (baseline, week 1, 6–8, and 14–18, Fig. 1). Secondly, the PERMANOVA test was conducted on all samples within a specific time point to identify any differences between the infant and neonatal schedule groups. Of particular interest was the question of whether a different beta diversity was present in participants with a positive or negative vaccine response (vaccine take or shedding). Vaccine take and stool shedding was also analysed in association with relevant clinical variables, including mode of delivery, birth weight, breastfeeding, an episode of gastroenteritis or exposure to antibiotics during the 18-week study period. These additional variables were included in the PERMANOVA test to assess for confounding factors. The initial PCA analysis was conducted on all samples to identify any differences between the various time points, and a biplot was included to identify the taxa that

were predominantly responsible for the separation of the time point clusters. An additional PCoA analysis was conducted to identify differences between modes of delivery, gender, and antibiotic use, with data separated by time points. Further PCoA was conducted to examine the differences between infant and neonatal schedules, with the data separated by time points. Lastly, PCoA was conducted on week 6 neonatal samples to identify differences between positive and negative vaccine take and stool shedding for doses 2, 3, or 4. The week 6 neonatal samples were selected for analysis because they yielded the majority of the significant PERMANOVA results.

### Determination of the bacterial taxa associated with vaccine response

To determine whether bacterial taxa were associated with vaccine response and other relevant clinical factors, we used "a multivariable statistical framework for finding associations between clinical metadata and potentially high-dimensional microbial multi-omics data" through the R package MaAsLin2 v 1.5.1 (https://huttenhower.sph. harvard.edu/tools). We used the MaAsLin2 default Q-value cutoff of 0.25, and relative abundances of bacterial taxa were used for MaAsLin2 analysis in a mixed effect modelling. Two examples of how the "Maasline2" function was used are below.

The first example analysed if bacterial taxa were associated with the Malawi neonatal schedule sample at week 1 and vaccine take dose 1 in a univariate analysis by not including any other factors.

*fit_data = Maaslin2(input_data = count, input_metadata = meta, output = "MWI Infant Week1 cumVT_dose1 minab 0 minpre 0", fixed_effects = c("VT_dose1"), min_abundance = 0, min_prevalence = 0, normalisation = "none", transform = "none")*

The second example analysed if bacterial taxa were associated with Malawi neonatal schedule samples at week 1 and vaccine take dose 1 in multivariate analysis by including all available and applicable other clinical factors.

*fit_data = Maaslin2(input_data = count, input_metadata = meta, output = "MWI Infant Week1 cumVT_dose1 minab 0 minpre 0 gender, gest_wks, weight, gastro18wks, abx18wks, MofD, breastfeeding_18wks", fixed_effects = c("VT_dose1","gender","gest_wks","birth_weight","gastro_18wks","abx_18wks","MofD","breastfeeding_18wks"), min_abundance = 0, min_prevalence = 0, normalisation = "none", transform = "none")*

The results of the MaAsLin2 analysis with a minimum abundance of 0.1%, for the RV3-BB Malawi and Indonesia cohort are presented in the Supplementary Data 2.

### Comparison between RV3-BB and Rotarix® Microbiome study cohorts

The outcome from the RV3-BB Microbiome study was then compared with data re-analysed from a Rotarix® vaccine clinical trial conducted in India ($n = 1154$ samples) and in Malawi ($n = 461$ samples)[22]. The Rotarix® vaccine study used a similar methodology as our current study including Illumina MiSeq technology sequencing the bacterial 16S V3V4 region[22]. For the comparison study, bacterial 16S reads from the Rotarix® vaccine studies[22] were downloaded from the European Nucleotide Archive (Accession code PRJEB38948) and analysed using the same bioinformatics pipeline as was used in our RV3-BB study (Supplementary Table 2). This enabled us to compare results from two different rotavirus vaccines (RV3-BB and Rotarix®) studied in three countries and to compare results from the RV3-BB and Rotarix® microbiome studies conducted in similar sites in Malawi. The key differences between the two studies included: the method of detection of vaccine stool shedding, the number of vaccine doses and the age of administration of the first dose of vaccine. The Rotarix® study was measured by an NSP2 (Non-Structural Protein-2) assay with a Ct cut-off of < 40[22]. In contrast, in the RV3-BB studies, VP6 was confirmed by conventional PCR and all positive samples were sequenced to confirm RV3-BB vaccine virus shedding as distinct from shedding of a

wildtype rotavirus infection. To minimise the impact of these differences, we used the extended dataset from the Rotarix® study and re-analysed all samples collected at weeks 1, 4, 6, and 10 samples for shedding response ("yes" or "no") by vaccine dose, NSP2 shedding CT value and by combined NSP2 and VP6 shedding CT value (< 35) (from data publicly available but not published[22]). Infants in the Rotarix® study received two doses of vaccine (6 and 10 weeks of age) with the first dose at 6 to 8 weeks of age compared to three doses in the RV3-BB study with the first dose either soon after birth (0 to 5 days) in the neonatal schedule, or the first dose at 6 to 8 weeks of age in the infant schedule group. Also, there was no placebo group in the Rotarix® study. The proportion of high-quality reads from the Rotarix® Malawi study ($n = 283$ samples) and Rotarix® India study ($n = 1145$ samples) was 86% and 77%, respectively, and these were analysed through the same MOTHUR pipeline as used for our RV3-BB cohorts. The oligotyping pipeline generated 6370 and 2922 OTPs, respectively, for the Rotarix® India and Malawi study. From the Rotarix® Malawi and India studies, we classified 137 and 140 taxa, respectively, with the same inclusion criteria as above. This included 99.88% and 99.98% of all bacterial 16S reads for both studies (Supplementary Data 1).

## Data availability

All raw fastq sequence files have been deposited in the NCBI short read archive with basic metadata. Detailed metadata information is available on request to the study sponsor. The trial protocol and statistical analysis plan are publicly available for the Indonesia RV3-BB Trial (http://www.anzctr.org.au/Trial/Registration/TrialReview.aspx? ACTRN=12612001282875)[32] and for the Malawi RV3-BB Trial (ClinicalTrials.gov: NCT 03483116)[31]. All raw fastq sequence data generated in this study have been deposited in the NCBI short-read archive database under the following BioProject ID PRJNA1039271. https://url.au.m. mimecastprotect.com/s/IH3uCWLVLMFPx7Oyu6dyT3?domain=ncbi. nlm.nih.gov. The study metadata are available for ethically approved projects on application to the corresponding author. The informed consent provided by participants requires that their metadata data is shared for ethically approved research studies only. Applications for sharing of metadata can be made to Professor Julie Bines (corresponding author), providing ethical approval and information for the proposed use of the data. Requests for data will be responded to within a 4-week period to allow review of the documentation provided.

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

## Acknowledgements

We would like to thank the infants and their families for participating in this study. We acknowledge the hard work of the RV3 study teams at the Bangwe, Limbe and Ndirande Health Centers and Queen Elizabeth Central Hospital in Malawi and the Soeradji Tirtonegoro Hospital Klaten and District Hospital in Sleman, Indonesia. We are indebted to the members of the DSMB: Dr Beatrice de Vos (Chair), Professors Stephen Graham, Kristine Macartney, Peter Richmond and Matthew Law; and the RV3 Rotavirus Vaccine Scientific Advisory Committee: chaired by Professor Sir Gustav Nossal A.O., including Karen Kotloff, Duncan Steele, Tilman Ruff, John Mathews, Kim Mulholland, and Don Roberton. We acknowledge the administration support from Ms Christine Storey. We are extremely grateful for the guidance provided by consultants to this study, including Nicole Kruger (NMK Consulting), Wasima Rida (independent biostatistics consultant) and Mark Sullivan (Medicines Development for Global Health). We are grateful to the Bill and Melinda Gates Foundation (OPP1164384, OPP1111055, OPP1058454), an Australian Tropical Medicine Commercialisation Grant (APP 50343) and the Australian National Health and Medical Research Council (APP 1012425), PT BioFarma Indonesian and the Victorian Government's Operational Infrastructure Support Programme who provided funding for this study. The funders of this study had a limited role in the study design but no role in protocol development, sample collection or analysis, data interpretation or the writing of this report.

## Author contributions

J.E.B. conceived the study, and J.W. designed and carried out the data analysis with input from J.E.B., A.H., and C.M.D. D.S.O., R.B., D.P., C.M.D., E.L. and N.B.D. processed the samples for analysis. J.A.T., C.S.W., H.N., Y.S., D.W., K.C.J., A.T., A.M., J.M., M.I.G., N.B.Z., N.C., E.W., F.J., N.C., K.B., and J.P.B. were involved in the acquisition of clinical samples and sample handling. J.W. developed the first draft of the report with input from J.E.B. and A.H. All authors were involved in the review and editing of the manuscript.

## Competing interests

MCRI holds the patent for the RV3-BB vaccine; J.W., J.E.B., A.H., C.M.D., E.L., D.S.O., R.B., D.P., N.B.D., K.B., J.P.B., E.W., and F.J. are employees of MCRI. C.M.D. has served on advisory boards for GSK (2019, 2021), with all payments directed to an administrative fund held by MCRI. N.C. is a National Institute for Health and Care Research (NIHR) Senior Investigator (NIHR203756). N.C., D.W., A.T., and K.C.J. are affiliated to the NIHR Global Health Research Group on Gastrointestinal Infections at the University of Liverpool; and to the NIHR Health Protection Research Unit in Gastrointestinal Infections at the University of Liverpool, a partnership with the UK Health Security Agency in collaboration with the University of Warwick. The views expressed are those of the author(s) and not necessarily those of the NIHR, the Department of Health and Social Care, the UK government or the UK Health Security Agency. All the remaining authors do not have competing interests.

## Additional information

Josef Wagner  ✉, Amanda Handley[1,4], Celeste M. Donato[1,3], Eleanor A. Lyons , Daniel Pavlic[1], Darren Suryawijaya Ong , Rhian Bonnici[1], Nada Bogdanovic-Sakran[1], Edward P. K. Parker , Christina Bronowski , Jarir At Thobari[7,8], Cahya Dewi Satria[8], Hera Nirwati[9], Desiree Witte[6,10], Khuzwayo C. Jere , Ashley Mpakiza[10], Emma Watts[1], Ann Turner[10], Karen Boniface[1], Jonathan Mandolo[10,11], Frances Justice[1], Naor Bar-Zeev[6], Miren Iturriza-Gomara[6,12], Jim P. Buttery[1,3,13], Nigel A. Cunliffe[6], Yati Soenarto[8] & Julie E. Bines  ✉

[1]Enteric Diseases, Murdoch Children's Research Institute, Parkville, Victoria, Australia. [2]Respiratory Virus and Microbiome Initiative, Wellcome Sanger Institute, Hinxton, UK. [3]Department of Paediatrics, The University of Melbourne, Parkville, Victoria, Australia. [4]Medicines Development for Global Health,

Melbourne, Victoria, Australia. [5]Department of Infectious Disease Epidemiology and International Health, London School of Hygiene and Tropical Medicine, London, UK. [6]Institute of Infection, Veterinary and Ecological Sciences, University of Liverpool, Liverpool, UK. [7]Department of Pharmacology and Therapy, Faculty of Medicine, Nursing and Universitas Gadjah Mada, Yogyakarta, Indonesia. [8]Pediatric Research Office, Department of Pediatrics, Faculty of Medicine, Nursing and Universitas Gadjah Mada, Yogyakarta, Indonesia. [9]Department of Microbiology, Faculty of Medicine, Nursing and Universitas Gadjah Mada, Yogyakarta, Indonesia, Faculty of Medicine, Nursing and Universitas Gadjah Mada, Yogyakarta, Indonesia. [10]Malawi Liverpool Wellcome Programme, Blantyre, P.O. Box 30096 Chichiri, Malawi. [11]Department of Clinical Science, Liverpool School of Tropical Medicine, Liverpool, UK. [12]GSK Vaccines for Global Health Institute, Sienna, Italy. [13]Department of Infectious Diseases, Royal Children's Hospital, Parkville, Australia. [14]Department of Gastroenterology and Clinical Nutrition, Royal Children's Hospital, Parkville, Australia. ✉e-mail: josef.wagner@mcri.edu.au; jebines@unimelb.edu.au

