## [Transparent Peer Review file · Nature Communications]

Early-life gut microbiome associates with positive vaccine take and shedding in neonatal schedule of the human neonatal rotavirus vaccine RV3-BB

Corresponding Author: Dr Josef Wagner

Version 0:

Reviewer comments:

Reviewer #1

(Remarks to the Author)

The article “RV3-BB Rotavirus vaccine promotes microbiome diversity and a healthy microbiome in infants in Indonesia and Malawi” by Wagner et al is an analysis of associations between microbiome composition and RV3-BB rotavirus vaccine immunogenicity in a cohort of infants in Indonesia and Malawi who received a RV3-BB rotavirus vaccine at differing vaccination schemes.

The study contributes to a growing body of literature hypothesizing that the bacterial microbiome may interfere with rotavirus vaccine performance in low- and middle-income countries, where live attenuated rotavirus vaccines have shown demonstrably lower protection than high income studies. Associations between microbiome composition and rotavirus vaccines in the literature have been varied and inconsistent across geographic settings, with heterogeneity in the characterization of the microbiome and measurement of vaccine immunogenicity. This study is potentially important because the RV3-BB vaccine can be given at birth and the improved performance of the birth-dose may be mediated by the more sparsely colonized bacterial microbiome at that time.

General comments

This is a large and important data set derived from two key rotavirus vaccine trials, with microbiome characterization of a significant number of infant fecal samples, taken from under-represented and under-described geographic areas, with an enormous number of statistical analyses performed. However, this manuscript, in its current form, is extremely difficult to understand and therefore not possible to sufficiently review and interpret.

The major current shortcomings of the manuscript are that the statistical analyses are not shaped by a clearly articulated central hypothesis and the study lacks an underpinning rationale for which specific comparisons and outcomes matter and why. Rather, in the results everything seems to be compared with everything, with confusing and inconsistent labels for study arms, time points, and dosing. This makes interpretation of this substantial body of work extremely challenging. Specific examples include the use of multiple and poorly defined vaccine outcomes, (cumulative and per dose fold change and shedding) despite the availability of a gold standard for immunogenicity (3-fold titer increase 28 days post last vaccination) or even available clinical outcomes (prevention of severe rotavirus gastroenteritis). There is also lack of clarity about why it matters that certain timepoints are being compared and which comparisons have priority. The reviewer, for example, is interested in whether the microbiome composition at the time of vaccine administration associates with vaccine immunogenicity, specifically seroconversion following an entire vaccine series. The appropriate analysis is then testing fecal samples taken at birth from infants given a RV3BB neonatal dose, and comparing between those that did and did not seroconvert. However, despite multiple readings, the reviewer cannot derive this information from figures 2 and 3. This is a shame, because the data is clearly embedded in the manuscript and present somewhere in the analyses. Finally, the multiplicity of testing the authors employ increases risk of false positive findings across the manuscript.

The second major comment for the study is that the authors make central claims that they did not evaluate. The title, abstract, and text claim their vaccine “promotes” a “healthy microbiome”. This claim is problematic as (1) microbiome development, as measured from alpha and beta diversity from birth to 18 weeks was not different for RV3-BB vaccinated and unvaccinated infants (2) there is no shared or consensus definition of what a ‘healthy microbiome’ (title) or “negative” or “beneficial” taxa (line 55, abstract) may be at different timepoints in infants in Indonesian and Malawi settings or LMIC settings in general (3)

the authors did not study any functional outcomes besides vaccine response (metabolomics, metagenomic analysis, inflammatory parameters) to support a claim of difference in functional capacity in infants' microbiota over time.

Specific comments

Title, abstract, Discussion (lines 314-316) Strongly suggest the authors remove claims of promotion of a 'healthy microbiome' and "positive" or "negative" taxa across title, abstract, and text. Suggest that the authors articulate what their pre-existing definition of 'microbiome development' was. Should the authors wish to maintain their health claims, then suggest they add testing of functional or inflammatory markers to support their claims.

Lines 102-108 –Outline of analyses is confusing due to use of vague terminology. For example, lines 102 and 108 the authors use "differences" unclear what is meant in both cases; similar issue with line 103 and the use of "results", and line 106 and the use of "RV3-BB outcomes". In each case it is unclear if the authors refer to differences in microbiome composition, vaccine immunogenicity or efficacy.

Results

General comments. Sufficient review of the manuscript is challenging as it is difficult to understand which study arms, time points, and outcomes are being used. For example, Fig 2 extended data, IP dose 1 is actually days 0-5 at birth, where by one group has been vaccinated and one group has not, but the label implies vaccination for all groups. Suggest the authors harmonize all labels, use age to describe the time point of sampling, and significantly reduce the number of vaccine outcomes they are using. This will simplify comparison across studies and interpretation of the study results.

Line 122-124: how were participants for the microbiome sub-study selected from the main study?

Table 1: Suggest to provide a more expanded table of characteristics of study participants from main study vs sub study participants which include potential modifiers of microbiome composition such as mode and location of delivery, breastfeeding.

Line 124 – 'the participant characteristics were similar across both studies' do the authors refer to Indonesia vs Malawi or main study vs substudy.

Line 127 – "a higher proportion of participants" do the authors mean of the main study or the microbiome substudy, please clarify

Line 130 – GE episodes, the authors refer to Table one, but GE episodes are not included in Table 1, please include.

Line 131 – antibiotic exposure, again the authors refer to table 1, but this data is not included, please include.

Line 133 – rather abrupt transition to bacterial reads. Please add a brief description of when fecal samples were collected from infants, and the number of fecal samples collected to contextualize this data or move all to results.

Line 140, add that Simpson's was not significantly increased over time

Line 149, if the reviewer is interpreting the data correctly, the authors are now comparing the microbiome alpha diversity between study participants at each time point. Suggest to make this clearer in the text and figure 2 extended (e.g. suggest to change to "for most of the diversity indexes and time points)

Extended data figure 2:

- for Indonesia, microbiome samples were not taken at baseline? And if IP Dose 1 samples are at 0-5 days of age, how are baselines samples defined for Malawi?

- What is PBO - placebo? It is not defined in the paper or in the legend. Also, suggest the authors consider combining the INF and placebo infants for this time point? As these infants should have similar exposures and the major question is whether they differ from those children who have been vaccinated.

- Line 155 Because it appears that fecal samples were obtained 3-5 days following RV3-BB administration, the major question here is whether vaccine administration altered alpha diversity. These results suggest that it has, with increases in Fisher and Richness in RV3-BB administered children but not in children without vaccine administration in Indonesia. This significance of this result is not clear from the current text and figures.

Figure 2: see also comments for Methods. The reviewer is unfortunately not understanding what is being compared. For example, in Figure 2a, the reviewer assumes that Fisher's index is being measured for fecal samples taken at baseline and compared between infants with and without a 'cumulative vaccine take'. However, what is IPd1, 2 and 4? Is this presence of vaccine take at these time points despite the label 'cumulative vaccine shedding'. Or are these the different arms of the vaccine study? Same questions for cumulative shedding and figure 3 – here the x-axis is labeled without time but with IP dose. What is IP dose referring to?

Supplementary Fig 1 – label by age instead of by dose. Use systematic labels across both study locations.

Line 173 – suggest again to use time and not dose labels and to make the more simple conclusion here that beta diversity differs by age but not vaccine arms in both geographic settings.

Line 199 – why were 47 comparative analyses done? Authors should define what the major hypothesis of the study is and limit their analysis to testing those hypotheses. The reviewer strongly suggests selecting one primary vaccine outcome per infant (IgA seroconversion 4 weeks post last vaccine dose) and moving all other comparisons to secondary/supplemental analyses.

Line 200 – again, what do the authors mean by "at IP dose 1 and 2" time point or vaccine arm?

Line 201 "two comparative analyses were significantly different" – which comparative analyses?

Supplementary tables 5/6: directionality of associations not evident

Extended data figure 3 and 4 – it is not clear to the reviewer what is being compared in these figures. The legend reads "performed in the neonatal IP dose 2 (week 6 group)" - what does this mean? Fecal samples taken from which time point? And why dose 2 when the individual plots are reading dose 1, 2, and 3? The lack of clarity makes it impossible to assess the

validity of the comparisons.

Line 253, please include the rationale for reanalyzing already analyzed and published data.

Discussion

Line 321-323, the authors say there was no increase in alpha diversity irrespective of RV3-BB administration, however in extended data figure 2D and 2F there is a significant increase in alpha diversity in RV3-BB vaccinated vs non-vaccinated infants, suggesting the opposite of this claim. Please clarify.

Paragraph 336, please address that more effective RV3-BB replication may be altering the beta diversity at early time points and driving the association between composition and response. Please also address that fecal samples were collected following vaccine administration rather than prior to vaccine administration.

Line 341, please define "normal bacterial taxa profile" and how the authors arrived (statistically) at this definition. The opposite argument could also be made that RV3-BB administration altered the neonatal microbiome?

Discussion, the authors make comparisons between their own study and other published literature. Suggest that they include a discussion of limitation of comparisons given age and geography are predominant drivers of microbiome composition.

Discussion, suggest that the authors discuss use of 16S vs metagenomic sequencing in the characterization of the microbiome and concomitant advantages/limitations.

Line 375 to 378, the authors tested associations between taxa and rotavirus vaccine response within vaccine arms, not whether RV3-BB administration vs no RV3-BB administration altered composition towards particular taxa. Suggest to either show this analysis or remove these claims (across the manuscript).

Line 395, the reanalysis of the Rotarix Malawi study and lack of similar associations with vaccine shedding and serum IgA response to the RV3-BB is a fascinating finding and deserves to be highlighted further in the discussion.

Line 399, suggest to better highlight how age is a significant confounder of microbiome composition and complicates study analysis and conclusions. (as shown dramatically in authors own PCoA plots – supplementary figure 1).

Methods

Line 494, please confirm that ethical approval was obtained specifically to evaluate microbiome composition in fecal samples (not only cord blood) from both studies

suggest to contextualize by either expand or move to are referred to

The study is worthwhile as rotavirus vaccination given at birth rather than 6 weeks of age may have increase immunogenicity

Line 522 – suggest to add Figure 1 to text

Line 525 – sample collection, what was the average duration of time between sample collection and storage to -80C?

Line 533 – open access to sequences?

Line 542 – shedding definition – please specify, presence of any shedding on any day or shedding on all days

Line 550 – cumulative vaccine take, please define "positive vaccine take"

Line 593 – were Indonesian infants given Rotarix?

Lines 627 – Rotarix cohort – The acronym IND and IDN are too close, suggest another acronym if these are Rotarix samples from India

Lines 640-645 – all these vaccine outcomes remain unclear despite this text, please clearly define these classifications.

- For example for "cumulative vaccine" after dose 1,2, and 3, do the authors mean a serum IgA value measured 4 weeks post dose of > 3 fold increase from baseline, measured at any of the three time points?

- And what is the difference between A and B?

- What is the difference between cumulative shedding and cumulative shedding? Shedding that was not present at dose 1? What was the physiologic rationale for all of these outcomes?

Line 805, confirm that false discovery rate was used for all reported p-values

Reviewer #2

(Remarks to the Author)

This is a well-presented report detailing the impact of rotavirus vaccination on the gut microbiome composition in children from Malawi and Indonesia. They first describe the association with different diversity metrics then show correlations between microbial composition and key indicators of rotavirus vaccine response. A key result is the association between positive oral rotavirus response and high abundance of Bacteroides when administered in neonates. This is important as it suggested that early administration of rotavirus vaccination has an impact on the development of the gut microbiome.

Comments:

1. The Introduction does not mention strain replacement as one potential explanation for the failure to eliminate RV. Perhaps it merits a mention.
2. There is significant anxiety over the sheer number of statistical tests performed. These are not always appropriate: the Kruskal-Wallis test assumes independence, but this is not true of samples collected sequentially from the same children to study maturation. In Figure 4 are we only shown the significant taxa? Again there is anxiety over multiple testing, but was this particular analysis corrected, and how? This should be made clear in the legend.
3. While the authors state that the number of children delivered by Caesarean section is low, in the Malawi cohort this was 14% and could be analysed as a cofactor.
4. The authors touch on the microbial differences over time in the section starting at line 172 by describing differences in beta diversity but it would be interesting to also highlight the taxa responsible for these PCoA separations seen. Is there an expected maturation of the gut microbiome overtime and is this influenced by vaccination strategies? In the differential abundance sections, they only describe timepoint specific associations.
5. Online Methods: In sample collection, they describe that "Frozen neat stool aliquots were shipped on dry". What does this mean?
6. Online Methods: Alpha diversity/ Richness was calculated on normalized counts as opposed to raw counts. Is there a justification for doing so as normalised values tends to skew/ inflate diversity estimates? This should be corrected.
7. The Discussion should clearly state that a complete understanding of the relationship between microbiome and vaccine effects must ultimately rely on careful measures of protection, not diversity or vaccine take.

Minor points:

1. The meaning of lines 106-108 is not at all clear.
2. Line 125: 'protption' should be 'proportion'
3. Line 333: 'greater significant' should be 'greater significance'
4. 354: 'i9s' should be 'is'
5. Fisher and Fischer are both used; this should be corrected
6. In the legend to Ext Fig 2, it says "plotting were done". Please correct.

Reviewer #3

(Remarks to the Author)

Version 1:

Reviewer comments:

Reviewer #1

(Remarks to the Author)

Many thanks to the authors for their extensive review of their manuscript, "Early bacterial microbiome is associated with positive vaccine take and shedding in neonatal schedule of the human neonatal rotavirus vaccine, RV3-BB." The reviewer appreciates the extensive responses to each of the raised comments, the manuscript is much improved due to the narrowed selection of relevant vaccine endpoints as well as time points.

Major comments

The reviewer (apologies) continues to have misgivings about the number of statistical tests in the manuscript and concomitant risk of multiple testing alongside a sometimes selective presentation of positive associations over negative associations.

The authors have kept three major vaccine outcomes – vaccine seroconversion, vaccine shedding, and a very large cumulative outcome of "vaccine take". However, they also have included sub outcomes in which every dose of vaccine shedding and take are included in the analysis. There does seem to be a grading in the quality of these associations, where the likelihood of false statistical associations increases significantly as the authors move from seroconversion through shedding to take. The reviewer suggests that the authors consider restricting these outcomes to binary outcomes to mitigate this risk. For example, (assuming the same mechanism would be at play for each association) to not assess shedding separately by every time point, but make an outcome that is yes or no for shedding at any time point. The timepoint specific outcomes can be presented in supplementary materials. See specific suggestions in text.

Next, the reviewer suggests that the authors better build their rationale for their refined hypotheses in the introduction. What the literature is that supports this stated hypothesis. Why do the authors (mechanistically) think that microbiome alpha diversity and composition is associated with vaccine shedding and take for neonatal arms but not infants study arms. What are possible virologic or immunologic explanations that have led the authors to make this hypothesis?

Finally, the reviewer suggests that the authors address the possibility that better vaccine replication alters microbiome

composition (infants' microbiome are more easily perturbed) and that associations between shedding and microbiome alpha diversity are a result of vaccine administration and not facilitating the vaccine's performance.

Minor comments (and expansions on major comments) by line item below

Abstract (line 44) suggest association instead of impact

Abstract (line 49) "compared to the placebo group, participants receiving three doses of RV3-bb in the neonatal schedule have higher alpha diversity" is unclear. Higher alpha diversity in baseline samples? Higher alpha diversity following vaccination?

Abstract (line 51) Suggest to define shedding, as not defined within the abstract

Abstract (line 53) suggest to modify to 'may present'

Include anti-RV IgA seroconversion results in the abstract

Introduction

Line 77, suggest to minimally add mode of delivery, breast milk, solid foods, antibiotic insults to development of the microbiome

Line 81, add "preterm neonates"

Line 95, remove "positive characteristics" or clearly cite literature that suggests that the outcomes listed (high alpha diversity, differences in beta diversity, and bacterial taxa profile) are "positive."

Line 100 "one of dose", typo?

Line 107 Suggest to add: and placebo recipients over time.

Line 112, suggest to alter "difference in beta diversity" to "beta diversity"

Line 114 "these data suggest that the early gut microbiome provides a gut environment that optimizes the potential for a positive vaccine response at that time point" this is quite speculative for an introduction, suggest to move to conclusion? Additionally, one could make the opposite argument – that only with the birthdose RV3-BB administration is there a significant association with microbiome composition, therefore other vaccines and timepoints are less dependent on microbiome background characteristics for vaccine protection. Could the authors comment?

Line 118, do the authors mean over time and following vaccination?

Line 119 "who did not shed the vaccine virus" please specify which schedule and dose this is relevant to.

Line 120 - A higher abundance of these bacteria might also suggest that these infants have had other insults (antibiotics/GE/hospital birth/relative immune compromise) that alter their microbiota and therefore also their immune response. Suggest to include this in considerations in conclusion. Further, decreased shedding at dose 2 might be reflective of induction of immunity at dose 1, therefore specify which dose these findings are relevant to.

Note that the authors are not presenting their anti-RV IgA seroconversion results in their introduction or abstract. The lack of association between IgA seroconversion and microbiome composition seems to me a very important finding in this study given that anti-RV IgA is the best available correlate of protection for rotavirus vaccines. The authors should lead with this information, include it clearly in their abstract, and discuss its relative importance clearly in their conclusion.

Authors' rebuttal states:

Fig 1 describes the timepoints when the baseline stool was taken. In the Malawi study, a baseline stool was taken soon after birth prior to administration of the first dose of vaccine or placebo. In the Indonesia study the first stool was collected in the first week of life – 3-7 days after administration of the first dose of vaccine or placebo.

Suggest to include this before result reporting to facilitate interpretation of country-specific results.

Results

Line 170, clarify how the stool samples for these time points were selected – was just any stool sample taken, or the closest to vaccination? For these timepoints, the microbiome is likely altered by rotavirus vaccine strain replication. Therefore associations between vaccine shedding and microbiome are logical, given that the highest vaccine replication will likely

perturb the microbiome the most. Please rebut or include this consideration in your discussion.

Vaccine response in association with alpha and beta diversity

Very strong suggestion to include the IgA seroconversion response in the main text alongside the stool shedding and positive vaccine take, given that these are endpoints defined by the authors and that the IgA seroconversion endpoint has the best evidence that it correlates with vaccine protection.

Table 2/Fig2

I appreciate that the authors took the time to collate their study results into this table and figure, but it remains complex to the reviewer, and therefore will likely be complex to the reader. It appears that there are separate analyses for whether or not there was vaccine take or shedding following 1, 2, or 4 doses for both schedule groups. See General comments. Could the authors further simplify by simplifying their endpoints. This reduces the number of statistical analyses, increases the understandability and improves comparability across country and schedule groups.

Suggest that the authors follow the approach they describe in their rebuttal: divide children in two 3 groups per study arm and simplify: ever had vaccine take (Y/N), ever had stool shedding (Y/N), seroconversion (Y/N), presented by study arm. the understandability and improves the comparability across study arms of the study results. This table and fig2 could be moved to the supplementary.

Line 251 – suggest to not only describe the positive findings and equally note the lack of separation at other doses (and timepoints)? This framing suggests that there are important microbiome differences, but the preponderance of the evidence is actually not showing differences by vaccine endpoints for microbiome composition, which seems just as valuable an outcome to report.

Same suggestion for simplification in Figures 3 and 4, with per dose outcomes in supplementary

Figure 4, if vaccine take definition did not include shedding, was the difference in PCoA at 6 weeks maintained?

Line 357, consider replacing PCA directionality with age directionality for taxa (like Bacteroides, Escherichia and Streptococcus) that distinguish separation across the axes

Bacterial taxa influence RV3-BB vaccine response

Line 379, suggest to change subtitle to “Bacterial taxa associate with RV3-BB vaccine response”

Line 385, confirm no differences by seroconversion

Line 460 – were there differences in breastfeeding rates between the two study cohorts given Bifidobacterium difference

Paragraph 453, is the reviewer correct that there was no re-analysis of associations between microbiome and Rotarix study vaccine endpoints in Malawi, only a comparison across ages for all included infants, regardless of vaccine take, shedding and seroconversion? This is confusing given data prior that evaluates associations with vaccine performance. Suggest to clarify in text.

Discussion

Line 520, what is meant by “shedding following”, this sentence is somewhat confusing, suggest to rephrase.

Line 523, please discuss that increased microbiome alpha diversity associating with shedding may be confounded by effective shedding increasing alpha diversity.

Paragraph 546, suggest to name the vaccine studied for the cited literature for clarity.

Reviewer #2

(Remarks to the Author)

The revised manuscript is definitely improved, but still needs more work. There are three areas which need clarification.

1 There is a major outstanding area of confusion: definitions of endpoints. In Methods, these are defined as follows: “A positive vaccine response was defined as IgA seroconversion 28 days after administration of vaccine or placebo, stool shedding of RV3-BB detected after a dose of vaccine or placebo (Malawi: 3 to 5 days; Indonesia 1 to 7 days) and/or vaccine take at any study assessment timepoint (Fig. 1) 28,29. Vaccine take was defined as IgA seroconversion or stool shedding following administration of any dose of vaccine or 702 placebo. Positive vaccine take was defined as...” Yet in Results (line 177ff) and in Table 2 results are presented as “positive vaccine take and/or vaccine virus stool shedding”, even though the definition of vaccine take includes stool shedding. This does make it difficult to follow. Would it not be simpler to present IgA seroconversion and stool shedding separately and to avoid the terms “take” and “response”? Or

make the definitions clearer?

2 In Table 2, how are the P values derived? The authors have decided, wisely, to focus on the hypothesis now set out at the end of the Introduction. Table 2 should now reflect that. I would have expected a statistical test of the difference in (for example) alpha diversity between those with IgA seroconversion and those without. Is that what was done? If so, what do the numbers 0.5299 or 0.0373 mean? If these are P values as the legend suggests, why is it being given to 4 decimal places? If these are alpha diversities, both responder and non-responder values should be given. The legend needs to be more explicit as in its current form I find it unclear.

My previous points have been addressed, though the issue of multiple testing is not fully resolved and should be addressed in the Discussion.

Reviewer #3

(Remarks to the Author)

Version 2:

Reviewer comments:

Reviewer #1

(Remarks to the Author)

Many thanks to the authors for their careful revision of their manuscript, and particularly the simplification of Table 2. All reviewer concerns have been addressed

Reviewer #2

(Remarks to the Author)

I am happy to say that, from my perspective, the manuscript is greatly improved by the clarifications introduced into text and figure legends. I have two suggestions. The more important is to clarify causation. In line 547 it is stated that "higher alpha diversity at early time points (baseline and week 1) was associated with positive vaccine take..." but the key table 2 and figures 2 and 3 and supplementary figure 4 do not show baseline data. Baseline data are shown in Supp Table 5, but no P values (if I have understood correctly). BL data are shown in Figure 5, but this is about time trends, not the difference between responders and non-responders. These key data are important to establish causation, so need to be shown in Table 2 and Figure 2. With this, it will be possible to strengthen the inference in the Abstract and the Discussion, either to claim that vaccine modulates the microbiota, or that the microbiota determines vaccine responsiveness.

A rather minor point is that in line 544 the results referred to are really the results set out in the previous publications from this trial.

Reviewer #3

(Remarks to the Author)

Early bacterial microbiome is associated with positive vaccine take and shedding in neonatal schedule of the human neonatal rotavirus vaccine, RV3-BB

Josef Wagner^{1,2,3*}, Amanda Handley^{1,4}, Celeste Donato^{1,3}, Eleanor A. Lyons¹, Daniel Pavlic¹, Darren Suryawijaya Ong¹, Rhian Bonnici¹, Nada Bogdanovic-Sakran¹, Edward P.K. Parker⁵, Christina Bronowski⁶, Jarir At Thobari^{7,8}, Cahya Dewi Satria⁸, Desiree Witte^{6,10}, Hera Nirwati⁹, Khuzwayo C. Jere^{6,10}, Ashley Mpakiza^{10,11}, Emma Watts¹, Ann Turner¹⁰, Karen Boniface¹, Jonathan Mandolo^{10,11}, Frances Justice¹, Naor Bar-Zeev⁶, Miren Iturriza-Gomara^{6,12}, James P. Buttery^{1,3,13}, Nigel A. Cunliffe⁶, Yati Soenarto⁸, Julie E. Bines^{1,3,14*}

REVIEWER COMMENTS

Reviewer #1 (Remarks to the Author):

The article "RV3-BB Rotavirus vaccine promotes microbiome diversity and a healthy microbiome in infants in Indonesia and Malawi" by Wagner et al is an analysis of associations between microbiome composition and RV3-BB rotavirus vaccine immunogenicity in a cohort of infants in Indonesia and Malawi who received a RV3-BB rotavirus vaccine at differing vaccination schemes.

The study contributes to a growing body of literature hypothesizing that the bacterial microbiome may interfere with rotavirus vaccine performance in low- and middle-income countries, where live attenuated rotavirus vaccines have shown demonstrably lower protection than high income studies. Associations between microbiome composition and rotavirus vaccines in the literature have been varied and inconsistent across geographic settings, with heterogeneity in the characterization of the microbiome and measurement of vaccine immunogenicity. This study is potentially important because the RV3-BB vaccine can be given at birth and the improved performance of the birth-dose may be mediated by the more sparsely colonized bacterial microbiome at that time.

General comments

This is a large and important data set derived from two key rotavirus vaccine trials, with microbiome characterization of a significant number of infant fecal samples, taken from under-represented and under-described geographic areas, with an enormous number of statistical analyses performed. However, this manuscript, in its current form, is extremely difficult to understand and therefore not possible to sufficiently review and interpret.

The major current shortcomings of the manuscript are that the statistical analyses are not shaped by a clearly articulated central hypothesis and the study lacks an underpinning rationale for which specific comparisons and outcomes matter and why.

We are grateful that the Reviewer appreciated that this manuscript presents a large and an important dataset originating from under-represented and two under-described geographical regions. We sincerely apologize that the Reviewer found that the manuscript difficult to understand. To address this concern we have comprehensively revised the manuscript with a primary focus to clearly articulate the central hypotheses, and primary and secondary aims. We also have revised the results section to more logically follow the results that address these specific aims.

The manuscript now reads:

Introduction: Page 4; line 93-97: Central hypothesis:

- "We hypothesized that vaccine response (vaccine take, IgA seroconversion and/or stool vaccine virus shedding) following administration of RV3-BB in a neonatal vaccine schedule (first dose administered at 0 to 5 days of age) would be associated with positive characteristics of the early gut microbiome (high alpha diversity, differences in beta diversity and bacterial taxa profile), but that these changes would not be

observed when vaccine was administered in the infant vaccine schedule (first dose administered at 6 to 8 weeks of age).”

Introduction: Page 4; lines 97-111: Primary and Secondary Aims:

- “To explore this association we analyzed the gut microbiome in stool samples collected during clinical trials of the RV3-BB vaccine in Indonesia²⁹ and Malawi²⁸ focusing on three key study timepoints; week 1 when the neonatal vaccine schedule group had received one of vaccine compared to a placebo dose in the infant vaccine schedule; week 6 when the neonatal vaccine schedule group had received two doses of vaccine compared one vaccine dose in the infant vaccine schedule and after three doses of vaccine in either the neonatal or infant vaccine schedule (week 14-18) (**Fig. 1**) and analysed this in relation to measures of vaccine response (vaccine take, IgA seroconversion, vaccine virus stool shedding). “
- “To examine if the administration of a live rotavirus vaccine (RV3-BB) administered soon after influenced the development of the gut bacterial microbiome we compared the gut microbiome (alpha diversity, beta diversity and bacterial taxa profile) in vaccine recipients and placebo recipients.”
- “To determine if our findings were vaccine specific (RV3-BB:G3P[6]) or the administration schedule specific (neonatal vaccine schedule versus infant vaccine schedule), we then compared the abundant bacterial taxa identified at key study timepoints in the RV3-BB clinical trials conducted in Indonesia and Malawi with data obtained from clinical trials of the Rotarix vaccine (G1P[8]) administered in the infant vaccine schedule conducted in India and Malawi ²².”

Rather, in the results everything seems to be compared with everything, with confusing and inconsistent labels for study arms, time points, and dosing.

This makes interpretation of this substantial body of work extremely challenging.

We sincerely apologize if the results were not presented in a way that was clear to the Reviewer. To address this concern, we have fully revised all data tables and figures and the text describing the results within the entire manuscript, to simplify and clarify the key study groups, variables and timepoints. We have also focused on ensuring that the variables are clear and consistent across both clinical trials despite some variations in exact timepoints inherent within the clinical trial design (for example – vaccines were administered with EPI vaccines but the EPI schedules varied in Indonesia and Malawi so there is a slight difference in study week definition between the 2 studies). Figure 1 has been modified to be specific which dose was a vaccine vs placebo dose and we have deleted the use of the term “IP”. The text has been expanded to provide further clarify.

Specific examples include the use of multiple and poorly defined vaccine outcomes, (cumulative and per dose fold change and shedding) despite the availability of a gold standard for immunogenicity (3-fold titer increase 28 days post last vaccination) or even available clinical outcomes (prevention of severe rotavirus gastroenteritis).

We have reviewed the entire manuscript to limit the comparisons presented to only those that specifically address the stated hypothesis and aims. We have used the measures vaccine immune response that we were defined (and published) as the immunogenicity endpoints of the RV3-BB clinical trials in both Indonesia and Malawi (vaccine take, IgA seroconversion and vaccine-virus stool shedding). The Malawi clinical trial was not an efficacy study so we have no data on the prevention of severe rotavirus gastroenteritis. Although the Indonesia clinical trial was a Phase 2b efficacy study, the Microbiome Study was conducted in the Immunogenicity Sub-study, a smaller cohort of the main efficacy trial. Due to the much smaller sample in this Immunogenicity Sub-study it was not powered to assess vaccine efficacy against severe rotavirus gastroenteritis. A comprehensive description of the main clinical trial outcomes have now been provided in the revised Materials and Methods section (Page 27-28, lines 622-661). Just to confirm that in the RV3-BB clinical trials we defined IgA response as IgA seroconversion, a three-fold or greater titer increase from baseline 28 days post-receipt of a dose of vaccine or placebo (Fig. 1). In the RV3-BB Microbiome Study in both Malawi and Indonesia we found no significant differences in alpha or beta diversity or bacterial taxa in participants who had IgA seroconversion (Supplementary Table 4). Therefore, the Results section focused on data on vaccine take and stool shedding where differences between groups were observed. This has now been described in the Results and Methods Sections.

Materials and Methods Section:

Page 28, lines 678-686;

- “Serum rotavirus IgA antibody titers were measured by ELISA using rabbit anti-RV3 polyclonal sera as the

coating antibody and RV3-BB virus or vero cell lysate as the capture antigen⁴⁴. The antigen-antibody complexes were detected with biotinylated anti-human IgA and streptavidin-horseradish peroxidase as previously described⁴⁴. Anti-rotavirus IgA seroconversion was defined as a threefold or greater increase from baseline in blood collected four weeks after any dose of vaccine or placebo (neonatal vaccine schedule group: following IP dose 1 (vaccine dose 1), IP dose 2 (vaccine dose 2), IP dose 3, or IP dose 4 (vaccine dose 3 and one dose of placebo); infant vaccine schedule group: following the IP dose 1 (placebo dose), IP dose 2 (vaccine dose 1), IP dose 3 (vaccine dose 2), or IP dose 4 (vaccine dose 3 and one dose of placebo)^{28, 29}.”

Page 29, lines 697-705

- “A positive vaccine response was defined as IgA seroconversion 28 days after administration of vaccine or placebo, stool shedding of RV3-BB detected after a dose of vaccine or placebo (Malawi: 3 to 7 days; Indonesia 1 to 7 days) and/or vaccine take at any study assessment timepoint (**Fig. 1**)^{28,29}. Vaccine take was defined as IgA seroconversion or stool shedding following administration of any dose of vaccine or placebo. Positive vaccine take was defined as evidence of vaccine take following IP dose 1 (vaccine dose 1), IP dose 2 (vaccine dose 2) or IP dose 4 (vaccine dose 3 and one dose of placebo) for the neonatal schedule group, and following the IP dose 1 (placebo dose) or IP dose 4 (vaccine dose 3 and one dose of placebo) for the infant schedule group (**Fig. 1**)^{28, 29}.”

Results Section: page 7, lines 180-184

- “No significant differences in alpha and beta diversity were observed in association with IgA seroconversion in the neonatal or infant vaccine schedule groups in the Indonesia or Malawi study (**Supplementary Table 4**). Therefore, further description below focus on analyses with significant diversity differences observed in association with vaccine take and stool shedding.”

Discussion: page 23, lines 525-533

The justification for the use of vaccine take and stool shedding as markers of vaccine response has been included.

- “We measured vaccine response using a combination of vaccine take, IgA seroconversion and vaccine virus shedding in the stool^{28,29}. This acknowledges that serum IgA is not an optimal serological correlate of protection for rotavirus²². As maternal serum IgA is not transferred across the placenta, newborns are relatively IgA deficient at birth. For this reason, IgA seroconversion or serum IgA titers may not be a reliable marker to assess vaccine response of a rotavirus vaccine administered at birth. Although not validated as a correlate of protection, vaccine-virus shedding in the stool provides a valuable insight, at the gut level, of the physiological response to a live virus, orally administered vaccine and the potential impact of the gut microbiome on the rotavirus vaccine response.”

Discussion: page 25, line 583-592

We have included the lack of a clinical efficacy endpoint in this study in the limitations.

- “We addressed this by re-analyzing the data from the Rotarix® Study from Malawi and India noting that some aspects of the methodology were similar, although not the definition of vaccine virus shedding, and there was overlap in study location²². This comparison revealed striking differences in the association between highly abundant taxa with vaccine virus shedding²². Whether this reflected the intrinsic differences between the vaccines (RV3-BB: G3P[6] vs. Rotarix® : G1P[8]), age at administration of the first dose (day 0-5 vs week 6), geography or the definition of vaccine shedding could not be determined in this analysis. However ultimately the success of a rotavirus vaccine is measured by its ability to protect individuals from severe rotavirus disease, and serum immune response and stool shedding of vaccine virus are markers of the vaccine response and not always an accurate predictor of the level of protection provided by the vaccine.”

There is also lack of clarity about why it matters that certain timepoints are being compared and which comparisons have priority. The reviewer, for example, is interested in whether the microbiome composition at the time of vaccine administration associates with vaccine immunogenicity, specifically seroconversion following an entire vaccine series.

The justification of the timepoints for assessment of the stool microbiome directly relates to the presence or absence of evidence of vaccine virus shedding in the stool at that same timepoint – which is day 3 to 7 after a dose of vaccine or placebo (Figure 1). The optimal timing of stool collection to detect vaccine virus shedding is based on kinetic studies of shedding of RV3BB in the stool of infants (Reference: Cowley et al: Rotavirus shedding

following administration of RV3-BB human neonatal rotavirus vaccine. Hum Vacc Immunother 13: 1908-1915. 2017).

We have added to the Discussion page 23, line 515-533:

- “Our results suggest the timing of the first dose of RV3-BB is important to optimize vaccine response. Although we found no differences in alpha diversity observed at baseline (pre-vaccine) and vaccine response (vaccine take, IgA seroconversion or stool shedding 3 to 5 days) after a birth dose of RV3-BB, higher alpha diversity observed at early timepoints (baseline and week 1) was associated with positive vaccine take and/or shedding at later timepoints (post-vaccine dose 2 and post-vaccine dose 3/placebo dose [IP dose 4*]) in Malawi. Positive vaccine take and shedding following was also associated with higher alpha diversity in stool collected 3 to 5 days after the time-matched dose of RV3-BB (vaccine dose 1 and microbiome week 1; and vaccine dose 2 and microbiome week 6). However, this association was only observed in the neonatal schedule group. Our results when RV3-BB was administered in the infant schedule group are consistent with studies using the Rotarix® and RotaTeq® vaccines, also administered in the routine infant schedule¹⁶⁻²¹ (**Supplementary Table 10**). We measured vaccine response using a combination of vaccine take, IgA seroconversion and vaccine virus shedding in the stool^{28,29}. This acknowledges that serum IgA is not an optimal serological correlate of protection for rotavirus²². As maternal serum IgA is not transferred across the placenta, newborns are relatively IgA deficient at birth. For this reason, IgA seroconversion or serum IgA titers may not be a reliable marker to assess vaccine response of a rotavirus vaccine administered at birth. Although not validated as a correlate of protection, vaccine-virus shedding in the stool provides a valuable insight, at the gut level, of the physiological response to a live virus, orally administered vaccine and the potential impact of the gut microbiome on the rotavirus vaccine response.”

The justification of the IgA seroconversion timepoint is based on current understanding of the peak level for serum IgA response after an oral dose of a rotavirus vaccine or exposure to wildtype rotavirus infection and has been the timepoint for almost all other previous rotavirus vaccine clinical trials.

Our stated central hypothesis is that vaccine response (vaccine take, IgA seroconversion and/or stool vaccine virus shedding) following administration of RV3-BB in a neonatal schedule (first dose < 5 days of birth) would be positively associated with the early gut microbiome (higher alpha, different beta diversity and specific bacterial taxa), but not observed when vaccine was administered in the infant schedule (first dose at 6 to 8 weeks of age). To address this hypothesis we focused our study on three key study timepoints;

- week 1 when the neonatal schedule group had received one of vaccine compared to a placebo dose in the infant schedule;
- week 6 when the neonatal schedule group had received 2 doses of vaccine compared one vaccine dose in the infant schedule
- after 3 doses of vaccine in either the neonatal or infant schedule (week 14-18)

Our results showed that the characteristics of the gut microbiome (alpha and beta diversity, and bacterial taxa) at the time of vaccine administration was positively associated with vaccine virus shedding and vaccine take. However, there was no significant association with IgA seroconversion in blood taken 28 days after vaccination. (Table 2, Fig.2, Supplementary Fig. 1).

Table 2 has been added to further explore not only if the microbiome assessed in stool taken at a specific timepoint (ie post dose 1, 2 or 4) influence the vaccine response at that timepoint, but also if the administration of an oral rotavirus vaccine impact on the characteristics of the microbiome at a later timepoint (for example – a positive vaccine take after one dose of vaccine in the neonatal schedule in Malawi was associated with increased alpha diversity at week 14). The Discussion has been expanded to include:

Discussion: page 23, line 511-514:

- “Here we report that vaccine response (vaccine take and/or stool shedding) was associated with a high alpha diversity, differences in beta diversity and bacterial taxa in the early gut microbiome when the human neonatal rotavirus vaccine (RV3-BB) was administered in the neonatal schedule in Malawi and Indonesia, but not in the infant schedule (**Table 2, Fig. 1a and 1b**).

Discussion: page 23, line 516-523:

- “Although we found no differences in alpha diversity observed at baseline (pre-vaccine) and vaccine response (vaccine take, IgA seroconversion or stool shedding 3 to 5 days) after a birth dose of RV3-BB, higher alpha diversity observed at early timepoints (baseline and week 1) was associated with positive vaccine take and/or shedding at later timepoints (post-vaccine dose 2 and post-vaccine dose 3/placebo dose [IP dose 4*]) in Malawi. Positive vaccine take and shedding following was also associated with higher alpha diversity in stool collected 3 to 5 days after the time-matched dose of RV3-BB (vaccine dose 1 and microbiome week 1; and vaccine dose 2 and microbiome week 6). However, this association was only observed in the neonatal schedule group.”

The appropriate analysis is then testing fecal samples taken at birth from infants given a RV3BB neonatal dose, and comparing between those that did and did not seroconvert. However, despite multiple readings, the reviewer cannot derive this information from figures 2 and 3. This is a shame, because the data is clearly embedded in the manuscript and present somewhere in the analyses.

We agree with the Reviewer that this was important and apologize if this information was not clearly presented.

The text and Figures and Tables have been revised to improve clarity:

- *Fig. 1:* describes the timepoints when the baseline stool was taken. In the Malawi study, a baseline stool was taken soon after birth prior to administration of the first dose of vaccine or placebo. In the Indonesia study the first stool was collected in the first week of life – 3-7 days after administration of the first dose of vaccine or placebo.
- *Methods page 28, line 663- 665 and 671-675* has been expanded to include the timepoints for stool and blood sample collection.
- *Table 2:* Describes the data on the stool microbiome for the Malawi and Indonesia study with the Baseline data (pre-IP) for Malawi and Week 1 for Indonesia; as well as other study timepoints.
- *Fig. 2:* presents the data from the Malawi neonatal schedule group with the top panel displaying the data on alpha diversity at baseline according to positive (Y) or negative (N) vaccine take and stool shedding.
- *Supplementary Fig. 3:* presents the data from the Indonesia vaccine schedule group with the top panel presenting the data on alpha diversity at week 1 according to positive (Y) or negative (N) vaccine take and stool shedding.
- *Results section: page 9, line 196-200*
 “There were no significant differences in alpha diversity observed in stool collected at baseline (pre-dose) and vaccine response (vaccine take or stool shedding) after the first dose of vaccine (**Fig. 2a**). A significant association was observed between alpha diversity in stool collected at baseline, with vaccine take at later timepoints (dose 2, dose 4* [following 3 doses of vaccine and one dose of placebo]) in the neonatal vaccine schedule group but not for stool shedding (**Fig. 2a**). “

Finally, the multiplicity of testing the authors employ increases risk of false positive findings across the manuscript.

We acknowledge that we used a number of tests to describe the alpha diversity (Fisher, Simpson’s index and Observed Richness) and beta diversity. We investigated characteristics of the high abundance bacterial taxa through MaAsLin2 analysis which is the next generation of MaAsLin (Microbiome Multivariable Association with Linear Models) statistic. We also included PCoA analysis to visualise microbial clusters associated with different metadata including vaccine variables. We believe that this detailed approach has provided robustness to our analysis and conclusions.

During the revision of the manuscript we have limited the presentation of the data to the results that address the stated primary and secondary aims. The data presented with respect to these aims are flagged below with the relevant figure and table as it appears in the revised manuscript.

- *Page 4, line 97-111*
 - “To explore this association we analyzed the gut microbiome in stool samples collected during clinical trials of the RV3-BB vaccine in Indonesia²⁹ and Malawi²⁸ focusing on three key study timepoints; week 1 when the neonatal vaccine schedule group had received one of vaccine compared to a placebo dose in the infant vaccine schedule; week 6 when the neonatal vaccine schedule group had received two doses of vaccine compared one vaccine dose in the infant vaccine schedule and after three doses of vaccine in either the neonatal or infant vaccine schedule (week 14-18) and analyzed this in relation to measures of vaccine response (vaccine take, IgA seroconversion, vaccine virus stool shedding).”

The data used to assess this aim included data:

- Describing alpha diversity and beta diversity, and high abundance bacterial taxa in stool samples at week 1 when the neonatal schedule group had received 1 dose of vaccine compared one dose of placebo in the infant schedule; and compared data in those participants who had positive (Y) vaccine take/stool shedding compared to negative (N) vaccine take/stool shedding (Table 2, Alpha diversity data Fig. 2 and Supplementary Fig.1); Beta diversity data Figure 4, Supplementary Fig. 2, 3 and 4); Bacterial taxa Fig.7)
 - Describing alpha diversity and beta diversity, and high abundance bacterial taxa in stool samples at week 6 when the neonatal schedule group had received 2 doses of vaccine compared one vaccine in the infant schedule; and compared data in those participants who had positive (Y) vaccine take/stool shedding compared to negative (N) vaccine take/stool shedding (Table 2, Alpha diversity data Fig. 2 and Supplementary Fig.1); Beta diversity data Figure 4, Supplementary Fig. 2, 3 and 4); Bacterial taxa Fig.7)
 - Describing alpha diversity and beta diversity, and high abundance bacterial taxa in stool samples after 3 doses of vaccine in either the neonatal or infant schedule (week 14-20); and compared data in those participants who had positive (Y) vaccine take/stool shedding compared to negative (N) vaccine take/stool shedding (Table 2, Alpha diversity data Fig. 2 and Supplementary Fig.1); Beta diversity data Figure 4, Supplementary Fig. 2, 3 and 4; Bacterial taxa Fig.7, Supplementary Fig. 4 and 5).
- “To examine if the administration of a live rotavirus vaccine (RV3-BB) administered soon after birth influenced the development of the gut bacterial microbiome we compared the gut microbiome (alpha diversity, beta diversity and bacterial taxa) in vaccine recipients and placebo recipients in Indonesia.”

The data used to assess this aim included data:

- Describing alpha diversity and beta diversity, and high abundance bacterial taxa in stool samples at week 1 when the neonatal schedule group had received 1 dose of vaccine compared one dose of placebo in the placebo group; and compared data in those participants who had positive (Y) vaccine take/stool shedding compared to negative (N) vaccine take/stool shedding (Table 2, Fig. 3, Supplementary Table 5 and 7)
 - Describing alpha diversity and beta diversity, and high abundance bacterial taxa in stool samples at week 6 when the neonatal schedule group had received 2 doses of vaccine compared to three doses of placebo in the placebo group; and compared data in those participants who had positive (Y) vaccine take/stool shedding compared to negative (N) vaccine take/stool shedding (Table 2, Supplementary Fig.1, Fig. 3, Supplementary Table 5 and 7)
 - Describing alpha diversity and beta diversity, and high abundance bacterial taxa in stool samples after 3 doses of vaccine and one dose of placebo in the neonatal vaccine schedule and four doses of placebo in the placebo group (week 14-18); and compared data in those participants who had positive (Y) vaccine take/stool shedding compared to negative (N) vaccine take/stool shedding (Table 2, Supplementary Fig.1, Fig. 3, Supplementary Table 5 and 7).
- “To determine if our findings were vaccine specific (RV3-BB:G3P[6]) or the administration schedule specific, we then compared our results with publicly available raw sequence data obtained from clinical trials of the Rotarix vaccine (G1P[8]) administered in the infant schedule conducted in India and Malawi.”

The data used to assess this aim included data:

- Describing high abundance bacterial taxa in stool samples (baseline, week 6 and week 14) in the combined RV3-BB neonatal and infant schedule group after 1 or 3 dose of vaccine compared to the Rotarix group at week 1, 6 and 10; and compared data in those participants who had positive (Y) vaccine take/stool shedding compared to negative (N) vaccine take/stool shedding (Fig 8 Supplementary Table 3, 9 ,10)

The second major comment for the study is that the authors make central claims that they did not evaluate. The title, abstract, and text claim their vaccine “promotes” a “healthy microbiome”. This claim is problematic as (1) microbiome development, as measured from alpha and beta diversity from birth to 18 weeks was not different for RV3-BB vaccinated and unvaccinated infants (2) there is no shared or consensus definition of what a ‘healthy microbiome’ (title) or “negative” or “beneficial” taxa (line 55, abstract) may be at different timepoints in infants in Indonesian and

Malawi settings or LMIC settings in general (3) the authors did not study any functional outcomes besides vaccine response (metabolomics, metagenomic analysis, inflammatory parameters) to support a claim of difference in functional capacity in infants' microbiota over time.

We acknowledge the Reviewer's comment and in response have:

Changed the title to "Early bacterial microbiome is associated with positive vaccine take and shedding in neonatal schedule of the human neonatal rotavirus vaccine RV3-BB". We believe that this title accurately reflects the results of this study. We acknowledge that this manuscript does not study any functional measures of the gut microbiome and have therefore deleted all reference to a "healthy microbiome" throughout the manuscript.

Specific comments

Title, abstract, Discussion (lines 314-316) Strongly suggest the authors remove claims of promotion of a 'healthy microbiome' and "positive" or "negative" taxa across title, abstract, and text. Suggest that the authors articulate what their pre-existing definition of 'microbiome development' was. Should the authors wish to maintain their health claims, then suggest they add testing of functional or inflammatory markers to support their claims.

- We acknowledge that this manuscript does not study any functional measures of the gut microbiome and have therefore deleted all reference to a "healthy microbiome" throughout the manuscript, including the abstract and title.

Lines 102-108 –Outline of analyses is confusing due to use of vague terminology. For example, lines 102 and 108 the authors use "differences" unclear what is meant in both cases; similar issue with line 103 and the use of "results", and line 106 and the use of "RV3-BB outcomes". In each case it is unclear if the authors refer to differences in microbiome composition, vaccine immunogenicity or efficacy.

We apologize that this was confusing to the Reviewer and have revised the manuscript throughout to improve clarity of meaning. For the sections specifically referred to above we have reworded this section as outlined below:

Introduction: Page 4, line 93-111 now reads:

- "We hypothesized that vaccine response (vaccine take, IgA seroconversion and/or stool vaccine virus shedding) following administration of RV3-BB in a neonatal vaccine schedule (first dose administered at 0 to 5 days of age) would be associated with positive characteristics of the early gut microbiome (high alpha diversity, differences in beta diversity and bacterial taxa profile), but that these changes would not be observed when vaccine was administered in the infant vaccine schedule (first dose administered at 6 to 8 weeks of age). To explore this association we analyzed the gut microbiome in stool samples collected during clinical trials of the RV3-BB vaccine in Indonesia²⁹ and Malawi²⁸ focusing on three key study timepoints; week 1 when the neonatal vaccine schedule group had received one of vaccine compared to a placebo dose in the infant vaccine schedule; week 6 when the neonatal vaccine schedule group had received two doses of vaccine compared one vaccine dose in the infant vaccine schedule and after three doses of vaccine in either the neonatal or infant vaccine schedule (week 14-18) and analysed this in relation to measures of vaccine response (vaccine take, IgA seroconversion, vaccine virus stool shedding). To examine if the administration of a live rotavirus vaccine (RV3-BB) administered soon after influenced the development of the gut bacterial microbiome we compared the gut microbiome (alpha diversity, beta diversity and bacterial taxa profile) in vaccine recipients and placebo recipients. To determine if our findings were vaccine specific (RV3-BB:G3P[6]) or the administration schedule specific (neonatal vaccine schedule versus infant vaccine schedule), we then compared the abundant bacterial taxa identified at key study timepoints in the RV3-BB clinical trials conducted in Indonesia and Malawi with data obtained from clinical trials of the Rotarix vaccine (G1P[8]) administered in the infant vaccine schedule conducted in India and Malawi²²."

Results

General comments. Sufficient review of the manuscript is challenging as it is difficult to understand which study arms, time points, and outcomes are being used. For example, Fig 2 extended data, IP dose 1 is actually days 0-5 at birth, where by one group has been vaccinated and one group has not, but the label implies vaccination for all groups. Suggest the authors harmonize all labels, use age to describe the time point of sampling, and significantly reduce the

number of vaccine outcomes they are using. This will simplify comparison across studies and interpretation of the study results.

We apologize for any confusion and we have completely revised the Results section and all figures and tables to improve clarity. In doing this we confirmed that the labelling has been harmonized and is consistent throughout and referred to Figure 1 which presents the key vaccine variables and timepoints.

Line 122-124: how were participants for the microbiome sub-study selected from the main study?

We had included the method for participant selection in the Material and Methods section (page 28, line 653-660) but in response to the Reviewer's comment we have added this to the main text (*Page 5, line 127-130*); which now reads:

- "Participants from the Indonesia R3-BB trial and Malawi RV3-BB trial were eligible for inclusion into the RV3-BB Microbiome Study if they had adequate stool volume available for microbiome analysis at key study time points (baseline, week 1, week 6-8, week 14-16 and week 18-20) in the per-protocol population (Indonesia n=193; Malawi n=186)."

Also to the *Methods section page 28, line 653-660* which now reads:

- "Participants from the Indonesia R3-BB trial and Malawi RV3-BB trial were eligible for inclusion into the RV3-BB Microbiome Study if they had adequate stool volume available for microbiome analysis at key study time points (baseline, week 1, week 6-8, week 14-16 and week 18-20) in the per-protocol population (Indonesia n=193; Malawi n=186)."

Table 1: Suggest to provide a more expanded table of characteristics of study participants from main study vs sub study participants which include potential modifiers of microbiome composition such as mode and location of delivery, breastfeeding.

For clarity and simplicity, we have limited *Table 1* to a comparison of participant characteristics in the Indonesia and Malawi Microbiome Studies – with the primary aim to focus on the comparability between the two study cohorts across a number of clinically relevant domains including mode of delivery and breast-feeding duration. An expanded table including data of study participants from the main study and Microbiome study has now been included as a *Supplementary Table 1*.

Line 124 – ‘the participant characteristics were similar across both studies’ do the authors refer to Indonesia vs Malawi or main study vs substudy.

We have addressed this potential for confusion across cohorts by dividing *Table 1* into the direct comparison between the Microbiome study cohorts in Indonesia and Malawi. We have now moved the participant characteristics between the main study and the Microbiome substudy into *Supplementary Table 1*. We anticipate that this will address any potential for confusion.

Line 127 – “a higher proportion of participants” do the authors mean of the main study or the microbiome substudy, please clarify

This is a comparison between the Microbiome study cohorts in Indonesia and Malawi. It is anticipated that the new and revised *Table 1* should limit any potential for confusion.

Line 130 – GE episodes, the authors refer to *Table one*, but GE episodes are not included in *Table 1*, please include. This data has been included into revised *Table 1*.

Line 131 – antibiotic exposure, again the authors refer to *table 1*, but this data is not included, please include. This data has been included into revised *Table 1*.

Line 133 – rather abrupt transition to bacterial reads. Please add a brief description of when fecal samples were collected from infants, and the number of fecal samples collected to contextualize this data or move all to results.

Thank you for this recommendation. The following sentences have been added to provide a smoother transition to the bacterial taxonomic analysis:

Page 7, Line 169-171 now reads:

- “After administration of the investigational product (vaccine or placebo dose), stool samples were collected on days 1 to 7 in Indonesia and days 3 to 5 in Malawi. The extracted bacterial 16s RNA was subjected to bacterial 16s variable region 3 and 4 sequencing.”

Line 140, add that Simpson’s was not significantly increased over time.

We have added reference to the Simpson index in describing changes over time.

Page 14, line 293-294 now reads:

- “Alpha diversity increased with increasing age in both the Malawi and Indonesia studies using the Fisher’s alpha index and Observed richness measure but not the Simpson index.”

Line 149, if the reviewer is interpreting the data correctly, the authors are now comparing the microbiome alpha diversity between study participants at each time point. Suggest to make this clearer in the text and figure 2 extended (e.g. suggest to change to “for most of the diversity indexes and time points)

In response to the Reviewer’s comment we have provided further analysis that focuses on the changes in the alpha diversity between time points and is now shown in a revised *Figure 5*. In *Figure 5* we present the Fisher’s alpha index, Simpson index and observed richness measure at each of the key study timepoints in the Malawi and Indonesia study cohorts, independent of vaccine response. The text has been adapted to reflect this change.

Page 14, line 293-311 which now reads:

- “Alpha diversity increased with increasing age in both the Malawi and Indonesia studies using the Fisher’s alpha index and Observed richness measure, but not the Simpson index. In Malawi, the Fisher’s alpha index increased from baseline to week 1 ($p=0.0239$), baseline to week 6 ($p<0.0001$), and baseline to week 14 ($p=0.0002$), and in Indonesia, from week 1 to week 14 ($p<0.0001$) and week 1 to week 18 ($p<0.0001$) (**Fig. 5a, 5d**). The Observed richness measure also increased with increasing age (Malawi: baseline to week 6 ($p=0.0002$) and baseline to week 14 ($p=0.0002$); Indonesia: week 1 to week 14 [$p<0.0001$] and week 1 to week 18 ($p<0.0001$) (**Fig. 5c, 5f**). In Malawi, no significant difference in the change in alpha diversity with increasing age was observed between the infant and neonatal vaccine schedule group when analysed using the Fisher’s alpha index or Observed Richness measure by timepoints, independent of vaccine response (**Fig. 5g, 5i**). The Simpson index identified a difference at week 6 with a higher alpha diversity ($p=0.003$) in the Malawi neonatal vaccine schedule group, when participants had received 2 doses of vaccine compared to one vaccine dose in the infant vaccine schedule group (**Fig. 5h**). As the Indonesia study design included a placebo group, we were able to compare the diversity in participants who received the RV3-BB with those who received placebo. We found no significant difference in alpha diversity between the neonatal schedule group and the placebo group, at the key study timepoints (week 1, 14 and 18) (**Fig. 5j, 5k, 5l**). In the infant vaccine schedule we observed a higher alpha diversity independent of vaccine response when compared to the placebo group at week 1 (Fisher’s alpha index: $p= 0.0007$; Observed richness measure $p=0.0108$) and at week 18 (Fisher’s alpha index: $p= 0.0194$) **Fig. 5j, 5l**.”

Extended data figure 2:

For Indonesia, microbiome samples were not taken at baseline? And if IP Dose 1 samples are at 0-5 days of age, how are baselines samples defined for Malawi?

- Stool samples were taken at baseline in Malawi, prior to administration of the first dose of vaccine or placebo and the key study timepoints in this revised manuscript refer to the timepoints in relation to the collection of stool samples for the microbiome analysis. This has been clarified in *Fig. 1*: describes the timepoints when the baseline stool was taken and in the *Methods page 28, line 663-665*.

- What is PBO - placebo? It is not defined in the paper or in the legend. Also, suggest the authors consider combining the INF and placebo infants for this time point? As these infants should have similar exposures and the major question is whether they differ from those children who have been vaccinated.

In revising the manuscript, tables and figures we have focused on simplifying, removing any unnecessary abbreviations and making all labels consistent and clear. We have deleted the use of “PBO”

We respectfully acknowledge the Reviewer’s suggestion to combine the infant vaccine schedule first dose (placebo) and the placebo group data at week 6 but decided to maintain these groups presented separately to avoid the potential any further confusion by adding another group to what a complex set of results is already.

- Line 155 Because it appears that fecal samples were obtained 3-5 days following RV3-BB administration, the major

question here is whether vaccine administration altered alpha diversity. These results suggest that it has, with increases in Fisher and Richness in RV3-BB administered children but not in children without vaccine administration in Indonesia. This significance of this result is not clear from the current text and figures.

We hope that in the revision of this manuscript this association is clearer. Our results show:

- *Page 7, line 176-196:* “High alpha diversity and differences in beta diversity of the early gut microbiome was observed in participants who had positive vaccine take and/or vaccine virus stool shedding after receiving RV3-BB in the neonatal vaccine schedule, with greater differences observed in Malawi compared to Indonesia. These differences were not observed in the infant vaccine schedule group in Indonesia or Malawi (except for a measure of beta diversity at week 14 in Malawi) or in the placebo group in Indonesia (**Table 2**).”
- *Page 9, line 196-198:* “In the Malawi cohort, Fisher’s alpha index and the Observed Richness measure were consistently higher in participants in the neonatal vaccine schedule group with a positive vaccine response (vaccine take or stool shedding) (**Table 2, Fig. 2**).”
- *Page 9, line 198-205:* “There were no significant differences in alpha diversity observed in stool collected at baseline (pre-dose) and vaccine response (vaccine take or stool shedding) after the first dose of vaccine (**Fig. 2a**).”
- *Page 9, line 200-206:* “A significant association was observed between alpha diversity in stool collected at baseline, with vaccine take at later timepoints (dose 2, dose 4* [following 3 doses of vaccine and one dose of placebo]) in the neonatal vaccine schedule group but not for stool shedding (**Fig. 2a**). At week 1, high alpha diversity was associated with a positive vaccine take and stool shedding after administration of vaccine doses 1 and 2 in the neonatal vaccine schedule group (**Table 2 and Fig. 2b**). This association was even stronger at week 6, when high alpha diversity was also associated with positive vaccine take and stool shedding after vaccine dose 2 and dose 4* (following three doses of vaccine and one dose of placebo), (**Table 2 and Fig. 2c**)”
- *In the cross-analysis of alpha diversity in the neonatal schedule compared to the placebo group in Indonesia:* “Compared to the placebo group, participants receiving three doses of RV3-BB in the neonatal schedule have higher alpha diversity, although beta diversity and abundant bacterial taxa are similar. Therefore, administration of the live virus RV3-BB at birth did not have a negative impact on the development of the gut microbiome in the first 18 weeks of life.” As shown in *Fig.3, Supplementary Fig.1* and in *Discussion page 24, line 546-547*.

Figure 2: see also comments for Methods. The reviewer is unfortunately not understanding what is being compared. For example, in Figure 2a, the reviewer assumes that Fisher’s index is being measured for fecal samples taken at baseline and compared between infants with and without a ‘cumulative vaccine take’. However, what is IPd1, 2 and 4? Is this presence of vaccine take at these time points despite the label ‘cumulative vaccine shedding’. Or are these the different arms of the vaccine study? Same questions for cumulative shedding and figure 3 – here the x-axis is labeled without time but with IP dose. What is IP dose referring to?

We apologize that Figure 4 and 5 have been confusing to understand and we have significantly revised these figures to improve the clarity of presentation.

Supplementary Fig 1 – label by age instead of by dose. Use systematic labels across both study locations.

We have revised the main text, all figures and tables with a focus on consistency of labelling.

Line 173 – suggest again to use time and not dose labels and to make the more simple conclusion here that beta diversity differs by age but not vaccine arms in both geographic settings.

Thank you and we have revised the labelling to focus on microbiome sample timepoints.

Line 199 – why were 47 comparative analyses done? Authors should define what the major hypothesis of the study is and limit their analysis to testing those hypotheses. The reviewer strongly suggests selecting one primary vaccine outcome per infant (IgA seroconversion 4 weeks post last vaccine dose) and moving all other comparisons to secondary/supplemental analyses.

As highlighted above we have limited the data presentation in the main text, Figures and Tables to the data that address the primary and secondary aims. We have focused the data presentation on the vaccine response variables – vaccine take and stool shedding – as these are the variables with the significant differences observed. We have also focused on addressing our primary and secondary aims. We believe that limiting the analysis to IgA seroconversion to

a single timepoint at 14-18 weeks misses the opportunity to identify the potential impact of a birth dose/neonatal vaccine schedule on the microbiome. Consistent with the data from Parker et al observed in the Rotarix studies, stool vaccine virus shedding, not surprisingly has the most consistent link with gut microbiome diversity. This raises important questions about the interaction between the gut microbiome and vaccine virus uptake and replication and gut mucosal immunity that is totally missed using an imperfect serological surrogate of protection (sIgA). We have added to the Discussion the following on page 24, line 526-533:

- “We measured vaccine response using a combination of vaccine take, IgA seroconversion and vaccine virus shedding in the stool^{28,29}. This acknowledges that serum IgA is not an optimal serological correlate of protection for rotavirus²². As maternal serum IgA is not transferred across the placenta, newborns are relatively IgA deficient at birth. For this reason, IgA seroconversion or serum IgA titers may not be a reliable marker to assess vaccine response of a rotavirus vaccine administered at birth. Although not validated as a correlate of protection, vaccine-virus shedding in the stool provides a valuable insight, at the gut level, of the physiological response to a live virus, orally administered vaccine and the potential impact of the gut microbiome on the rotavirus vaccine response.”.

Line 200 – again, what do the authors mean by “at IP dose 1 and 2” time point or vaccine arm?

We have revised the manuscript to improve clarity of data presentation and have consistently defined the intervention product as either vaccine or placebo dose and referred the reader back to Fig. 1 which describes the study design were relevant.

Line 201 “two comparative analyses were significantly different” – which comparative analyses?

This has been amended

Supplementary tables 5/6: directionality of associations not evident.

All results from MaAsLin2 analysis are now shown in Supplementary Tables 7 for the RV3-BB Malawi and Indonesia cohort, respectively, and in Supplementary Table 9 for the Rotarix Malawi and India cohort, respectively.

A yes value in the “value” column indicates that a particular taxa (feature column) was associated with a particular vaccine variable (metadata column). The direction of association is indicated by a positive or negative coefficient value in the “coef” column. The association of the most abundant taxa is now shown in a simplified form in new Figure 7.

Extended data figure 3 and 4 – it is not clear to the reviewer what is being compared in these figures. The legend reads “performed in the neonatal IP dose 2 (week 6 group)” - what does this mean? Fecal samples taken from which time point? And why dose 2 when the individual plots are reading dose 1, 2, and 3? The lack of clarity makes it impossible to assess the validity of the comparisons.

The extended data figures have been deleted. Key data from these figures have now been incorporated into the main and Supplementary figures. The legends have been revised for clarity.

Line 253, please include the rationale for reanalyzing already analyzed and published data.

The rationale for re-analysis of previously analyzed and published data is to limit any bias caused by different methods of sequence quality filtering and analysis methods. To compare any publicly available data with a newly analyzed dataset, both or all of the different datasets should be subjected to the same quality control and analysis methods as closely as possible, so that the most accurate comparison between studies can be made. We have included justification for this and how these studies differ in:

Methods: page 34, line 857-871.

- “For the comparison study, bacterial 16S reads from the Rotarix[®] vaccine studies²² were downloaded from the European Nucleotide Archive (Accession code PRJEB38948) and analysed using the same bioinformatics pipeline as was used in our RV3-BB study (**Supplementary Table 2**). This enabled us to compare results from two different rotavirus vaccines (RV3-BB and Rotarix[®]) studied in three countries and to compare results from the RV3-BB and Rotarix[®] microbiome studies conducted in similar sites in Malawi. The key differences between the two studies included: the method of detection of vaccine stool shedding, the number of vaccine doses and the age of administration of the first dose of vaccine. In the Rotarix[®] study was measured by an NSP2 (Non-Structural Protein-2) assay with an NSP2 Ct cut-off of <40²². In contrast, the RV3-BB studies

used a stricter shedding by combining a VP6 Ct <35 in addition to an NSP2 Ct <40 value cut-off value and then all positive samples were sequenced to confirm RV3-BB vaccine virus shedding as distinct from shedding of a wildtype rotavirus infection. To minimize the impact of these differences, we used the extended dataset from the Rotarix® study and re-analyzed all samples collected at week 1, 4, 6, and 10 samples for shedding response (“yes” or “no”) by vaccine dose, NSP2 shedding CT value and by combined NSP2 and VP6 shedding CT value (<35)(from data publicly available but not published²²).

Discussion page 25, line 576-589:

- “A shared limitation of studies is that the gut microbiome is influenced by multiple factors including environmental, social and genetic factors ^{22,43}. Study methodology, age at sample collection, limitation in sample numbers within subgroups, intrinsic differences between vaccines and vaccine schedules, and variability in definitions of vaccine response and vaccine virus shedding make it also challenging to compare data between studies ¹⁶⁻²². We attempted to address this by re-analysing the data from the Rotarix® Study from Malawi and India noting that some aspects of the methodology were similar, although not the definition of vaccine virus shedding, and there was overlap in study location ²². This comparison revealed striking differences in the association between highly abundant taxa with vaccine virus shedding ²². Whether this reflected the intrinsic differences between the vaccines (RV3-BB: G3P[6] vs. Rotarix® : G1P[8]), age at administration of the first dose (day 0-5 vs week 6), geography or the definition of vaccine shedding could not be determined in this analysis. However ultimately the success of a rotavirus vaccine is measured by its ability to protect individuals from severe rotavirus disease, and serum immune response and stool shedding of vaccine virus are markers of the vaccine response and not always an accurate predictor of the level of protection provided by the vaccine.”

:

Discussion

Line 321-323, the authors say there was no increase in alpha diversity irrespective of RV3-BB administration, however in extended data figure 2D and 2F there is a significant increase in alpha diversity in RV3-BB vaccinated vs non-vaccinated infants, suggesting the opposite of this claim. Please clarify.

For clarification the cross-analysis of alpha diversity in the neonatal schedule compared to the placebo group in Indonesia. This has now been shown in new *Fig.3, Supplementary Fig. 1* and in *Discussion page 24, line 546-547*.

Paragraph 336, please address that more effective RV3-BB replication may be altering the beta diversity at early time points and driving the association between composition and response. Please also address that fecal samples were collected following vaccine administration rather than prior to vaccine administration.

The Discussion throughout has been revised to limit any confusion in terminology and interpretation.

Line 341, please define “normal bacterial taxa profile” and how the authors arrived (statistically) at this definition. The opposite argument could also be made that RV3-BB administration altered the neonatal microbiome?

We have reworded this to avoid any misinterpretation:

Discussion, the authors make comparisons between their own study and other published literature. Suggest that they include a discussion of limitation of comparisons given age and geography are predominant drivers of microbiome composition.”

To address this recommendation, we have expanded the Discussion:

Page 25, line 576-589:

- “A shared limitation of all studies is that the gut microbiome is influenced by multiple factors including environmental, social and genetic factors ^{22,43}. Study methodology, age at sample collection, limitation in sample numbers within subgroups, intrinsic differences between vaccines and vaccine schedules, and variability in definitions of vaccine response and vaccine virus shedding make it also challenging to compare data between studies ¹⁶⁻²². We attempted to address this by re-analysing the data from the Rotarix® Study from Malawi and India noting that some aspects of the methodology were similar, although not the definition of vaccine virus shedding, and there was overlap in study location ²². This comparison revealed striking differences in the association between highly abundant taxa with vaccine virus shedding ²². Whether this reflected the intrinsic differences between the vaccines (RV3-BB: G3P[6] vs. Rotarix® : G1P[8]), age at

administration of the first dose (day 0-5 vs week 6), geography or the definition of vaccine shedding could not be determined in this analysis.”

- We have also added Supplementary Table 10 which provides a summary of previous microbiome studies and the region they were conducted, methodology and outcomes.

Discussion, suggest that the authors discuss use of 16S vs metagenomic sequencing in the characterization of the microbiome and concomitant advantages/limitations.

To address this suggestion, we have added the following to the Discussion.

Page 26, line 592-594 which now reads:

- “It is hoped that future advances will enable a more comprehensive investigation of the interactions between the complete gut microbiome and oral vaccine responses. These advances should also overcome the intrinsic limitations using the 16S marker gene analysis compared to metagenomic sequencing from which functional gene differences between vaccine responders and non-vaccine responders could be explored.”

Line 375 to 378, the authors tested associations between taxa and rotavirus vaccine response within vaccine arms, not whether RV3-BB administration vs no RV3-BB administration altered composition towards particular taxa. Suggest to either show this analysis or remove these claims (across the manuscript).

In this revised manuscript we present the results of a comparative analyses between the RV3-BB vaccinated group and a placebo group in the Indonesian cohort at week 14 and week 18. For this cross-analysis, we compared the alpha and beta diversity between the vaccinated and unvaccinated groups. This is now shown in *Fig.3 and Supplementary Table 5*. We have also expanded the presentation of bacterial taxa data across treatment allocation groups including vaccinated and placebo groups in *Fig.7 and Supplementary Table 7 and Supplementary Fig. 4 and 5*.

Line 395, the reanalysis of the Rotarix Malawi study and lack of similar associations with vaccine shedding and serum IgA response to the RV3-BB is a fascinating finding and deserves to be highlighted further in the discussion.

We have expanded to include this in the Discussion:

Page 23, line 523-533

- “Our results in the RV3-BB infant schedule group are consistent with published studies of the Rotarix and RotaTeq vaccines, also administered in an infant schedule, albeit these studies reported alpha diversity in association to IgA seroconversion or anti-rotavirus IgA titers¹⁶⁻²¹ (**Supplementary Table 10**). In our study we assessed vaccine response using a combination of vaccine take, IgA seroconversion and vaccine virus shedding in the stool, particularly relevant to assess response to a vaccine administered from birth. This acknowledges that serum IgA is not an optimal serological correlate of protection for rotavirus and maternal serum IgA is not transferred across the placenta, so newborns are relatively IgA deficient at birth. The gut microbiome reflects an intraluminal environment that could limit or facilitate vaccine virus uptake and replication. Although not validated as a correlate of protection, vaccine-virus shedding in the stool provides a valuable measure of vaccine response at the gut level and offers further insights into the potential impact of the gut microbiome on the response to an oral rotavirus vaccine²².

Page 25, line 580-589.

- “We addressed this by re-analyzing the data from the Rotarix® Study from Malawi and India noting that some aspects of the methodology were similar, although not the definition of vaccine virus shedding, and there was overlap in study location²². This comparison revealed striking differences in the association between highly abundant taxa with vaccine virus shedding²². Whether this reflected the intrinsic differences between the vaccines (RV3-BB: G3P[6] vs. Rotarix® : G1P[8]), age at administration of the first dose (day 0-5 vs week 6), geography or the definition of vaccine shedding could not be determined in this analysis.”

Line 399, suggest to better highlight how age is a significant confounder of microbiome composition and complicates study analysis and conclusions. (as shown dramatically in authors own PCoA plots – supplementary figure 1).

We have now added a more detailed analysis which taxa was the driving taxa behind age cluster separation as shown in PCoA plots in the result section (*Fig. 5 and Supplementary Fig. 3 and 4*).

Methods

Line 494, please confirm that ethical approval was obtained specifically to evaluate microbiome composition in fecal samples (not only cord blood) from both studies suggest to contextualize by either expand or move to are referred to

We have clarified this in the Methods Section, page 26, line 619-622 which now reads:

- “A two-stage consent process was followed. Pregnant women provided consent for the collection of a pre-birth maternal blood and stool sample and an after-birth infant cord blood and stool sample. Written informed study consent was obtained after birth from parents or guardians, prior to confirming eligibility for enrolment.”

The study is worthwhile as rotavirus vaccination given at birth rather than 6 weeks of age may have increase immunogenicity

Line 522 – suggest to add Figure 1 to text

We have added the description of the vaccine schedules, the timing of vaccine/placebo administration as requested – also referring to Figure 1:

Page 27, lines 625-632 now reads for the Indonesia study:

- “The infants were randomly assigned in a 1:1:1 ratio to one of three groups: the neonatal-schedule group, the infant schedule group, and a placebo group. The participants received four oral one-millilitre doses of the investigational product (IP), with doses administered at 0 to 5 days of age (IP dose 1), 8 to 10 weeks of age (IP dose 2), 14 to 16 weeks of age (IP dose 3) and 18 to 20 weeks of age (IP dose 4) (**Figure 1**). Each participant received three oral doses of RV3-BB and one dose of placebo. In the neonatal schedule vaccine group, doses 1, 2 and 3 were RV3-BB vaccine, while dose 4 was a placebo dose. In the infant schedule vaccine group, the first dose was a placebo dose, and the second, third, and fourth doses were the RV3-BB vaccine.”

Page 27, lines 643-647 now reads for the Malawi study:

- “The infants who met the eligibility criteria were randomly assigned to one of four treatment arms. Three of the arms involved the neonatal vaccine schedule, with RV3-BB (G3P[6]) administered at different vaccine titers while the fourth arm was the infant vaccine schedule group administered RV3-BB at a vaccine titer of 1×10^7 FFU/mL (Fig. 1a). Each participant received three oral doses of RV3-BB vaccine and one dose of placebo.”

Line 525 – sample collection, what was the average duration of time between sample collection and storage to -80C? This has been included in the Methods section, page 28, line 665-669.

- “The samples were collected from the participants' homes or healthcare centers by field workers or sample transport workers and stored in a refrigerated environment at a temperature between 2 and 10 degrees Celsius until transfer to the laboratory³. The samples were frozen at -80°C within 24 hours of their collection at the central study laboratory. “

Line 533 – open access to sequences?

The following information is available in the data availability statement, Page 36, lines 906-912:

- “All raw fastq sequence files have been deposited in the NCBI short read archive with basic metadata. Detailed metadata information is available on request to the Study Sponsor. The trial protocol and statistical analysis plan are publicly available for the Indonesia RV3-BB Trial (<http://www.anzctr.org.au/Trial/Registration/TrialReview.aspx?ACTRN=12612001282875>)²⁹ and for the Malawi RV3-BB Trial (ClinicalTrials.gov: NCT 03483116). The sequence data are available under the following BioProject ID PRJNA1039271

<https://url.au.m.mimecastprotect.com/s/IH3uCWLVLMPx7Oyu6dyT3?domain=ncbi.nlm.nih.gov>”.

Line 542 – shedding definition – please specify, presence of any shedding on any day or shedding on all days We have provided in the Methods Section

Page 29, line 696-697 now reads:

- “Positive stool shedding was defined as positive detection of RV3-BB vaccine virus in any stool after administration of vaccine (Malawi: day 3 to 5; Indonesia: day 3-7).”

Line 550 – cumulative vaccine take, please define “positive vaccine take”

The term “cumulative vaccine take” has been deleted. Additional information defining vaccine take has been included in Methods:

Page 29, lines 700-705 which now reads:

“Vaccine take was defined as IgA seroconversion 28 days after administration of a dose of vaccine or placebo or RV3-BB vaccine virus-like shedding detected in the stool taken, on days 3-7 in the Indonesia study and days 3 to 5 in the Malawi study, following administration after administration of vaccine or placebo. A positive vaccine take was defined as evidence of vaccine take following IP dose 1 (vaccine dose 1), IP dose 2 (vaccine dose 2) or IP dose 4 (vaccine dose 3 and one dose of placebo) for the neonatal schedule group, and following the IP dose 1 (placebo dose) or IP dose 4 (vaccine dose 3 and on dose of placebo) for the infant schedule group (see Fig 1).”

Line 593 – were Indonesian infants given Rotarix?

I think there maybe some confusion here. During the RV3-BB clinical studies in Indonesia and Malawi the participants were administered the RV3-BB vaccine or placebo. After completion of all study activities (including after the final stool and blood collection), a single dose of Rotarix was administered. This was to ensure that all study participants were protected by a licensed vaccine in the case of poor response to RV3-BB study vaccine. However in this revised manuscript we have focused on ensuring that the study design is clear by improving Fig 1, the text of the main manuscript and in the Methods section.

Lines 627 – Rotarix cohort – The acronym IND and IDN are too close, suggest another acronym if these are Rotarix samples from India

We agree that this could be confusing and removed all abbreviations for IND and IDN throughout.

Lines 640-645 – all these vaccine outcomes remain unclear despite this text, please clearly define these classifications.

- For example for “cumulative vaccine” after dose 1,2, and 3, do the authors mean a serum IgA value measured 4 weeks post dose of > 3 fold increase from baseline, measured at any of the three time points?

- And what is the difference between A and B?

- What is the difference between cumulative shedding and cumulative shedding? Shedding that was not present at dose 1?

In response to the Reviewer’s comments we have simplified the terminology in the text and figures. We also we have removed reference and data related to individual timepoint shedding or IgA seroconversion, so all data in this revised version relates to “cumulative” shedding to simplify data presentation. The definition of vaccine take, IgA seroconversion and stool shedding is described on page 28-29, line 680-706:

- “In the RV3-BB study in Indonesia and Malawi the data used in this analysis included:
 - Definition of Vaccine take (page 29, line 700-705)
 - “A positive vaccine take was defined as evidence of vaccine take (section 1.2) following IP dose 1 (vaccine dose 1), IP dose 2 (vaccine dose 2) or IP dose 4 (vaccine dose 3 and one dose of placebo) for the neonatal vaccine schedule group, and following the IP dose 1 (placebo dose) or IP dose 4 (vaccine dose 3 and on dose of placebo) for the infant vaccine schedule group (Fig 1).”
 - Definition IgA seroconversion (page 29 line 682-686)
 - “Anti-rotavirus IgA seroconversion was defined as a threefold or greater increase from baseline in blood collected four weeks after any dose of vaccine or placebo (Neonatal vaccine schedule group: following IP dose 1 (vaccine dose 1), IP dose 2 (vaccine dose 2), IP dose 3, or IP dose 4 (vaccine dose 3 and one dose of placebo); infant vaccine schedule group: following the IP dose 1 (placebo dose), IP dose 2 (vaccine dose 1), IP dose 3 (vaccine does 2), or IP dose 4 (vaccine dose 3 and one dose of placebo).”
 - Definition of RV3-BB vaccine virus stool shedding (page 29, line 695-696):
 - “Positive stool shedding was defined as positive detection of RV3-BB vaccine virus in any stool after administration a dose of vaccine or placebo (Malawi: day 3 to 5; Indonesia: day 3-7)^{28, 29}.”

What was the physiologic rationale for all of these outcomes?

The Microbiome study measured gut microbiome collected within a larger Phase 2b RV3-BB vaccine safety, immunogenicity and efficacy study in Indonesia and a Phase 2 dose-ranging vaccine safety and immunogenicity in Malawi. The Microbiome uses the same main vaccine response outcome variables as the primary study: that is vaccine take, IgA seroconversion and stool shedding.

We have included a statement on the physiological basis of the vaccine response outcomes in the discussion: Page 23, line 525-533:

- “We measured vaccine response using a combination of vaccine take, IgA seroconversion and vaccine virus shedding in the stool^{28,29}. This acknowledges that serum IgA is not an optimal serological correlate of protection for rotavirus²². Maternal serum IgA is not transferred across the placenta, so newborns are

relatively IgA deficient at birth, so IgA seroconversion or serum IgA titers may not be a reliable marker of vaccine response for a rotavirus vaccine administered at birth. Although not validated as a correlate of protection, vaccine-virus shedding in the stool provides a valuable insight of the response at the gut level and the potential impact of the gut microbiome on the response to an oral rotavirus vaccine²². “

Line 805, confirm that false discovery rate was used for all reported p-values

GraphPad Prism uses the original FDR method of Benjamin and Hochberg.

Therefore, in the Method section we confirm this:

Page 36, line 896-898:

- “For three or more group comparisons we used the Prism default FDR method of Benjamin and Hochberg”.

Reviewer #2 (Remarks to the Author):

This is a well-presented report detailing the impact of rotavirus vaccination on the gut microbiome composition in children from Malawi and Indonesia. They first describe the association with different diversity metrics then show correlations between microbial composition and key indicators of rotavirus vaccine response. A key result is the association between positive oral rotavirus response and high abundance of Bacteroides when administered in neonates. This is important as it suggested that early administration of rotavirus vaccination has an impact on the development of the gut microbiome.

Comments:

1. The Introduction does not mention strain replacement as one potential explanation for the failure to eliminate RV. Perhaps it merits a mention.

Reference to rotavirus strain replacement has been added to the introduction as suggested.

Page 3, line 62-65 now reads:

- “Challenges ensuring timely administration of a full 2 or 3 dose schedule from 6 weeks of age, cost and ongoing concerns regarding safety and rotavirus strain replacement remain barriers to the success of rotavirus vaccines⁴”

2. There is significant anxiety over the sheer number of statistical tests performed.

We appreciate the Reviewer’s comments and agree that this is a complex dataset and many statistical tests have been used to maximize the interrogation of this data. To address this we have significantly limited the number of test and data presented to those that directly address the primary and secondary outcomes. We have also simplified the presentation of vaccine response outcomes.

During the revision of the manuscript we have limited the presentation of the data to the results that address the stated primary and secondary aims. The data presented with respect to these aims are flagged below with the relevant figure and table as it appears in the revised manuscript.

- *Page 4, line 97-104*
 - “To explore this association we analyzed the gut microbiome in stool samples collected during clinical trials of the RV3-BB vaccine in Indonesia²⁹ and Malawi²⁸ focusing on three key study timepoints; week 1 when the neonatal vaccine schedule group had received one of vaccine compared to a placebo dose in the infant vaccine schedule; week 6 when the neonatal vaccine schedule group had received two doses of vaccine compared one vaccine dose in the infant vaccine schedule and after three doses of vaccine in either the neonatal or infant vaccine schedule (week 14-20) and analyzed this in relation to measures of vaccine response (vaccine take, IgA seroconversion, vaccine virus stool shedding).”

The data used to assess this aim included data:

- Describing alpha diversity and beta diversity, and high abundance bacterial taxa in stool samples at week 1 when the neonatal schedule group had received 1 dose of vaccine compared one dose of placebo in the infant schedule; and compared data in those participants who had positive (Y) vaccine take/stool shedding compared to negative (N) vaccine take/stool shedding (Table 2, Alpha diversity

- data Fig. 2 and Supplementary Fig.1); Beta diversity data Figure 4, Supplementary Fig. 2, 3 and 4); Bacterial taxa Fig.7)
- Describing alpha diversity and beta diversity, and high abundance bacterial taxa in stool samples at week 6 when the neonatal schedule group had received 2 doses of vaccine compared one vaccine in the infant schedule; and compared data in those participants who had positive (Y) vaccine take/stool shedding compared to negative (N) vaccine take/stool shedding (Table 2, Alpha diversity data Fig. 2 and Supplementary Fig.1); Beta diversity data Figure 4, Supplementary Fig. 2, 3 and 4); Bacterial taxa Fig.7)
- Describing alpha diversity and beta diversity, and high abundance bacterial taxa in stool samples after 3 doses of vaccine in either the neonatal or infant schedule (week 14-20); and compared data in those participants who had positive (Y) vaccine take/stool shedding compared to negative (N) vaccine take/stool shedding (Table 2, Alpha diversity data Fig. 2 and Supplementary Fig.1); Beta diversity data Figure 4, Supplementary Fig. 2, 3 and 4; Bacterial taxa Fig.7, Supplementary Fig. 4 and 5).
- “To examine if the administration of a live rotavirus vaccine (RV3-BB) administered soon after birth influenced the development of the gut bacterial microbiome we compared the gut microbiome (alpha diversity, beta diversity and bacterial taxa) in vaccine recipients and placebo recipients in Indonesia.”
The data used to assess this aim included data:
 - Describing alpha diversity and beta diversity, and high abundance bacterial taxa in stool samples at week 1 when the neonatal schedule group had received 1 dose of vaccine compared one dose of placebo in the placebo group; and compared data in those participants who had positive (Y) vaccine take/stool shedding compared to negative (N) vaccine take/stool shedding (Table 2, Fig. 3, Supplementary Table 5 and 7)
 - Describing alpha diversity and beta diversity, and high abundance bacterial taxa in stool samples at week 6 when the neonatal schedule group had received 2 doses of vaccine compared to three doses of placebo in the placebo group; and compared data in those participants who had positive (Y) vaccine take/stool shedding compared to negative (N) vaccine take/stool shedding (Table 2, Supplementary Fig.1, Fig. 3, Supplementary Table 5 and 7)
 - Describing alpha diversity and beta diversity, and high abundance bacterial taxa in stool samples after 3 doses of vaccine and one dose of placebo in the neonatal vaccine schedule and four doses of placebo in the placebo group (week 14-18); and compared data in those participants who had positive (Y) vaccine take/stool shedding compared to negative (N) vaccine take/stool shedding (Table 2, Supplementary Fig.1, Fig. 3, Supplementary Table 5 and 7).
- “To determine if our findings were vaccine specific (RV3-BB:G3P[6]) or the administration schedule specific, we then compared our results with publicly available raw sequence data obtained from clinical trials of the Rotarix vaccine (G1P[8]) administered in the infant schedule conducted in India and Malawi.”
The data used to assess this aim included data:
 - Describing high abundance bacterial taxa in stool samples (baseline, week 6 and week 14) in the combined RV3-BB neonatal and infant schedule group after 1 or 3 dose of vaccine compared to the Rotarix group at week 1, 6 and 10; and compared data in those participants who had positive (Y) vaccine take/stool shedding compared to negative (N) vaccine take/stool shedding (Fig 8 Supplementary Table 3, 9 ,10)

These are not always appropriate: the Kruskal-Wallis test assumes independence, but this is not true of samples collected sequentially from the same children to study maturation.

We acknowledge the Reviewer's comment and have now re-analysed the bacterial abundances over time the Skillings-Mack rank sum test, which is designed for partially balanced incomplete block designs or partially balanced random block designs (R package 'PMCMRplus') (Fig.5).

However, there is still some complexity in the analysis as we have performed alpha and beta diversity testing over multiple time points with each vaccine variable at each time point and with time point matched vaccine/placebo doses and retrospectively and prospectively correlated vaccine/placebo doses (Table 2, Fig. 5).

In Figure 4 are we only shown the significant taxa? Again there is anxiety over multiple testing, but was this particular analysis corrected, and how? This should be made clear in the legend.

We have revised former Figure 4. New Figure 7 shows all bacterial taxa with high abundance from the MaAsLin2 analysis with a minimum frequency of 1% and a coefficient value in the range ± 1.5 . This has been described in the Figure legend. For the identification of MaAsLin2 significant bacterial taxa, the default Q value of 0.25 was used – this has been described in the Methods section (page 34, line 832-852). For completeness we have presented all MaAsLin2 statistically significant taxa with a minimum abundance of 0.01% in Supplementary Table 7.

3. While the authors state that the number of children delivered by Caesarean section is low, in the Malawi cohort this was 14% and could be analyzed as a cofactor.

We agree that cofactors could influence the gut microbiome. We included a number of potentially relevant cofactors (gender, birth weight, mode of delivery, duration of breast feeding, gastroenteritis during the study period, antibiotic administration, gestational age) in the MaAsLin2 analysis and the PERMANOVA beta diversity analysis to explore if there was evidence suggestive of a confounding effect. We have added Supplementary Table 6, Supplementary Fig.3 that provide further data on these important co-founders.

4. The authors touch on the microbial differences over time in the section starting at line 172 by describing differences in beta diversity but it would be interesting to also highlight the taxa responsible for these PCoA separations seen. Is there an expected maturation of the gut microbiome overtime and is this influenced by vaccination strategies? In the differential abundance sections, they only describe timepoint specific associations.

We thank the Reviewer for this suggestion and have now included an additional analysis to the PCA timepoint cluster analysis, as now shown in Figure 6 and have added a biplot to the PCA plot (Supplementary Fig. 4 and 5) which show the main taxa responsible for the cluster separation associated with specific time points.

5. Online Methods: In sample collection, they describe that “Frozen neat stool aliquots were shipped on dry”. What does this mean?

Many thanks this has been corrected and now reads:

Page 28, line 668-671.

- “The samples were collected from the participants' homes or healthcare centres by field workers or sample transport workers and stored in a refrigerated environment at a temperature between 2 and 10 degrees Celsius until transfer to the laboratory⁵. The samples were frozen at -80°C within 24 hours of their collection at the central study laboratory. The frozen stool samples were shipped on dry ice to MCRI for subsequent extraction, sequencing and microbiome analysis.”

6. Online Methods: Alpha diversity/ Richness was calculated on normalized counts as opposed to raw counts. Is there a justification for doing so as normalised values tends to skew/ inflate diversity estimates? This should be corrected.

We acknowledge that there are conflicting opinions on the most appropriate method. We have used the normalisation of raw counts based on the approach recommended by Schloss et al (Ref #46). This method acknowledges that many metrics (including alpha and beta diversity) are affected by different levels of sampling and that the number of artefacts increases with additional sequences (Schloss et al. Reducing the Effects of PCR Amplification and Sequencing Artifacts on 16S rRNA-Based Studies: <https://doi.org/10.1371/journal.pone.0027310>).

7. The Discussion should clearly state that a complete understanding of the relationship between microbiome and vaccine effects must ultimately rely on careful measures of protection, not diversity or vaccine take.

We agree with the Reviewer's comment and have added to the Discussion

Page 25, line 586-589 which now reads:

- “Ultimately the success of a rotavirus vaccine is measured by its ability to protect individuals from severe rotavirus disease, and serum immune response and stool shedding of vaccine virus are markers of the vaccine response and not always an accurate predictor of the level of protection provided by the vaccine.”

Minor points:

1. The meaning of lines 106-108 is not at all clear.

The original sentence:

NCOMMS-24-26295-T

Thanks you – we have revised this sentence to improve clarification:

2. Line 125: 'protption' should be 'proportion'

Thank you. This has been corrected.

3. Line 333: 'greater significant' should be 'greater significance'

Thank you. This has been corrected.

4. 354: 'i9s' should be 'is'

Thank you. This has been corrected.

5. Fisher and Fischer are both used; this should be corrected

Thank you. This has been corrected.

6. In the legend to Ext Fig 2, it says "plotting were done". Please correct.

Thank you. This has been corrected. Extended Figures have now been removed. This analysis is now Figure main text Fig. 5 with an amended figure legend.

Reviewer #3 (Remarks to the Author):

Many thanks

REVIEWER COMMENTS

Reviewer #1 (Remarks to the Author):

Many thanks to the authors for their extensive review of their manuscript, “Early bacterial microbiome is associated with positive vaccine take and shedding in neonatal schedule of the human neonatal rotavirus vaccine, RV3-BB.” The reviewer appreciates the extensive responses to each of the raised comments, the manuscript is much improved due to the narrowed selection of relevant vaccine endpoints as well as time points.

We are grateful that the Reviewer has been satisfied that the extensive revision of the manuscript has addressed many of their concerns.

Major comments

The reviewer (apologies) continues to have misgivings about the number of statistical tests in the manuscript and concomitant risk of multiple testing alongside a sometimes selective presentation of positive associations over negative associations.

As the Reviewer has previously acknowledged this manuscript presents a very large and complex dataset. In an effort to meet the Journal requirements for word count, figures and tables we had to make a selection of the data presented in the main manuscript based on what we considered were the key findings (both positive and negative). However, data of interest not presented in the main text is presented in the extensive group of supplementary figures and tables and referenced within the main text. This was a pragmatic approach and in no way intended to bias the presentation of the outcomes of this study.

The authors have kept three major vaccine outcomes – vaccine seroconversion, vaccine shedding, and a very large cumulative outcome of “vaccine take”. However, they also have included sub outcomes in which every dose of vaccine shedding and take are included in the analysis. There does seem to be a grading into the quality of these associations, where the likelihood of false statistical associations increases significantly as the authors move from seroconversion through shedding to take. The reviewer suggests that the authors consider restricting these outcomes to binary outcomes to mitigate this risk. For example, (assuming the same mechanism would be at play for each association) to not assess shedding separately by every time point, but make an outcome that is yes or no for shedding at any time point. The timepoint specific outcomes can be presented in supplementary materials. See specific suggestions in text.

For clarification, the three major vaccine variables presented – seroconversion, shedding and vaccine take, directly reflect the key outcomes of the clinical trial data presented in the primary clinical trial publications (Witte, D. *et al.*. *Lancet Infect Dis* 22, 668–678 (2022). Bines, J. E. *et al.* *N Engl J Med* 378, 719–730 (2018)). The analysis of the microbiome was conducted at the clinical trial related timepoints associated with vaccine administration. We acknowledge the presentation of data per timepoint and related to each vaccine dose does look complex within Table 2 and have addressed the Reviewer’s concern by simplifying the data to a binary outcome and to data related to the same timepoint as suggested. Although we do wish to point out that in doing so it does exclude presentation of some interesting

associations – for example the characteristics of the microbiome at week 1 and 6 may influence vaccine shedding at week 14 after dose 4 (as is the case for the Malawi neonatal vaccine group). However, we have noted this in the text and referred to the more complex original data table which is now included in the supplementary appendix as recommended.

Next, the reviewer suggests that the authors better build their rationale for their refined hypotheses in the introduction. What the literature is that supports this stated hypothesis. Why do the authors (mechanistically) think that microbiome alpha diversity and composition is associated with vaccine shedding and take for neonatal arms but not infants study arms. What are possible virologic or immunologic explanations that have led the authors to make this hypothesis?

In the Introduction (line 79-85) we have now more clearly introduced the concept of developmental immaturity of the newborn gut, including immaturity of the gut mucus barrier, intestinal barrier function and immune barrier, as well as an immature gut microbiome and have added 3 references to support this concept. This developmental immaturity is thought to explain the increase susceptibility of the newborn to gastrointestinal infection. A live oral rotavirus vaccine acts by infecting the gut, replicating and inducing an immune response that is activated to protect when exposure to rotavirus in the future. RV3-BB is naturally adapted the newborn gut and is able to infect and replicate (as reflected by shedding) and induce an immune response (as reflected by vaccine take) without cause disease.

This now reads (line 80-85):

“It has been postulated that administration of the first dose of a rotavirus vaccine at birth may limit challenges to vaccine uptake and replication observed in older infants. The newborn gut is developmentally immature, characterised by a thin mucus layer, increased epithelial permeability and immaturity of the mucosal system, which may contribute to the increased susceptibility to gastrointestinal infection observed in newborns^{3, 22-24}.”

Line 93-95: now reads:

“Neonatal P[6] strains, such as RV3, are naturally adapted to the newborn gut and replicate well in the newborn gut in the presence of maternal and breast milk antibodies, irrespective of histoblood group antigen status³⁴⁻³⁷”

We have also expanded in the Discussion (line 535-540):

“The newborn gut differs structurally and functionally from the mature gut²³⁻²⁵. The first weeks after birth are also acknowledged as a critical period in the development of the gut microbiome as the newborn responds to the ex-utero environment. Birth may provide a window of opportunity to target the first dose of a live, oral rotavirus vaccine, with the aim to improve vaccine uptake and provide early protection from severe rotavirus disease.”

Finally, the reviewer suggests that the authors address the possibility that better vaccine replication alters microbiome composition (infants' microbiome are more easily perturbed) and that associations between shedding and microbiome alpha diversity are a result of vaccine administration and not facilitating the vaccine's performance.

We would refer the Reviewer to Figure 3 which presents the alpha diversity in the neonatal vaccine group and the placebo group at week 14 and 18. In this Figure, alpha diversity is higher in participants with positive shedding at week 14 after receiving 3 doses of vaccine, and higher than the alpha diversity observed in the

placebo group at this same timepoint. We interpret this data as suggesting that shedding following vaccine administration did not negatively alter the diversity of the microbiome but was associated with a more diverse microbiome. However, for clarification, we do not infer that the vaccine related differences in the diversity have facilitated the vaccine performance, rather vaccine administration in the neonatal schedule is positively associated higher diversity.

Minor comments (and expansions on major comments) by line item below

Abstract (line 44) suggest association instead of impact
This has been amended as recommended

Abstract (line 49) “compared to the placebo group, participants receiving three doses of RV3-bb in the neonatal schedule have higher alpha diversity” is unclear. Higher alpha diversity in baseline samples? Higher alpha diversity following vaccination? The comparison is alpha diversity measure (Fisher and Observed richness measure) at week 14 in the participants who had positive stool shedding in the neonatal vaccine group after three doses of vaccine compared to the alpha diversity measure in the non-shedding group in the neonatal vaccine group after three doses of vaccine and in the placebo group, as presented in Figure 3.

This has been clarified in the Abstract (line 53-55):

“Higher alpha diversity was observed in participants with shedding after three doses of RV3-BB in the neonatal vaccine schedule at week 14, compared to participants with no shedding or in the placebo group.”

Abstract (line 51) Suggest to define shedding, as not defined within the abstract
We have added definitions as recommended however this has increased the word count of the Abstract (line 47-50).

“Vaccine response were assessed using anti-rotavirus IgA seroconversion (three-fold increase from baseline), shedding of vaccine virus in stool (3 to 5/7 days following a dose of vaccine or placebo) and vaccine take (anti-rotavirus IgA seroconversion and/or stool shedding)”.

Abstract (line 53) suggest to modify to ‘may present’
This has been added in line 57.

Include anti-RV IgA seroconversion results in the abstarct

We have added reference to IgA seroconversion in the abstract (line 52-53):

“Here we show high alpha diversity, differences in beta diversity and high abundance of Bacteroides is associated with positive vaccine take and stool shedding following administration of RV3-BB in the neonatal schedule, but no association was observed with IgA seroconversion or in the infant schedule.”

Introduction

Line 77, suggest to minimally add mode of delivery, breast milk, solid foods, antibiotic insults to development of the microbiome

This has been added (line 83-85):

“The newborn gut microbiome is also less complex, initially reflecting maternal flora, but develops over the first months of life in response to feeding, antibiotic and environmental exposure and, in turn is shaped, by the developing infant immune system²⁶.”

Line 81, add “preterm neonates”

Added (line 88-89) the full term : “late onset neonatal sepsis”

Line 95, remove “positive characteristics” or clearly cite literature that suggests that the outcomes listed (high alpha diversity, differences in beta diversity, and bacterial taxa profile) are “positive.”

The word “positive” has been deleted.

Line 100 “one of dose”, typo?

We have corrected this and line 107 now reads: “their first dose”.

Line 107 Suggest to add: and placebo recipients over time.

We have added “over time” to this sentence (line 114) as recommended.

Line 112, suggest to alter “difference in beta diversity” to “beta diversity”

We have amended this sentence as requested (line 113).

Line 114 “these data suggest that the early gut microbiome provides a gut environment that optimizes the potential for a positive vaccine response at that time point” this is quite speculative for an introduction, suggest to move to conclusion? Additionally, one could make the opposite argument – that only with the birthdose RV3-BB administration is there a significant association with microbiome composition, therefore other vaccines and timepoints are less dependent on microbiome background characteristics for vaccine protection. Could the authors comment?

We had formatted this final paragraph in the introduction to include a statement of conclusion in according to the formatting instructions for the Journal. But will be guided by the Editor to their preference.

With respect to the Reviewer’s request for comment regarding the association between vaccine response and the microbiome in the neonatal schedule group but not in the infant schedule group – we postulate that this relates to the administration of the vaccine at birth when the gut microbiome is less complex and therefore is more favourable to vaccine uptake. The clinical trial data of increased RV3-BB vaccine efficacy and immunogenicity when administered in the neonatal vaccine schedule compared to the infant schedule is consistent with this hypotheses. However, this might not be the case for other rotavirus vaccines. RV3-BB is based on a naturally occurring human neonatal vaccine strain (RV3) which is well adapted to infecting the newborn gut without causing disease.

Line 118, do the authors mean over time and following vaccination?

This sentence has been clarified to refer to data reported after the participants in the neonatal schedule group had received 3 doses of RV3-BB. This now reads (line 125-126):

“Compared to the placebo group, participants receiving three dose of RV3-BB in the neonatal schedule have higher alpha diversity at week 14, although beta diversity and abundant bacterial taxa are similar.”

Line 119 “who did not shed the vaccine virus” please specify which schedule and dose this is relevant to.

To reduce the risk of this sentence now too long and less of a summary we have added data on the neonatal schedule group only and itemised the study site, vaccine dose and timepoint where this was observed. The sentence (line 127-131) now reads:

“Neonatal schedule group participants who did not shed the vaccine virus were more likely to have high abundance of Staphylococcus (Malawi: post dose 1 at week 1), Streptococcus (Malawi: post dose 2 at week 6; Indonesia: post dose 3 at week 14) and Escherichia (Indonesia: post dose 3 at week 14), suggesting that opportunistic, facultative anaerobic bacteria colonising the gut can interfere with vaccine response.”

Line 120 - A higher abundance of these bacteria might also suggest that these infants have had other insults (antibiotics/GE/hospital birth/relative immune compromise) that alter their microbiota and therefore also their immune response. Suggest to include this in considerations in conclusion. Further, decreased shedding at dose 2 might be reflective of induction of immunity at dose 1, therefore specify which dose these findings are relevant to.

We explored the potential effect of co-factors that could influence the composition of the microbiome, including mode of delivery, gastroenteritis, breast/formula feeding, birth weight and antibiotic exposure and reported this under the subheading (line 307): “Age, breast feeding, mode of delivery, birth weight and antibiotic exposure” and in Supplementary figures 2 and 3 and Supplementary tables 7 and 8. In brief we observed no differences in association with antibiotic use anytime during the study period (Supplementary Fig. 4). Some differences in beta diversity were noted for some parameters at early weeks (baseline, week 1 +/- 4) in the neonatal schedule group, but these did not persist at later timepoints (line 398-406) (Supplementary table 7 and 8). We further investigated the key taxa responsible for the separation in beta diversity between the early (week 1) versus later (Malawi: 6/14 week or Indonesia: 14/18 week) age clusters. This is presented in the text and Figure 6 and Supplementary Figure 5. Therefore, in this study we could find no evidence to support a significant effect of mode of delivery, gastroenteritis, breast/formula feeding, birth weight and antibiotic exposure on the effect of the microbiome on vaccine response to RV3-BB observed.

We have added the description of the dose related to the bacterial taxa findings reported in lines 424-439. This data is also displayed in Figure 7 a and b.

Note that the authors are not presenting their anti-RV IgA seroconversion results in their introduction or abstract. The lack of association between IgA seroconversion and microbiome composition seems to me a very important finding in this study

given that anti-RV IgA is the best available correlate of protection for rotavirus vaccines. The authors should lead with this information, include it clearly in their abstract, and discuss its relative importance clearly in their conclusion.

We have now added the definition of vaccine response to explicitly include anti-rotavirus IgA seroconversion and key results to the abstract (line 50-53) as requested. We have also included in Table 2 IgA seroconversion data in a similar format as for shedding and vaccine take. The more complete analysis for sIgA seroconversion is presented that had previously been in Supplementary Table 5.

However, we do not agree that the lack of an association between IgA seroconversion and microbiome composition is a “very important” finding. The pathway for the development of rotavirus vaccines have been challenged by the lack of a mechanistic correlate of protection that can be measured within the context of a clinical trial : “Due to the absence of a suitable correlate of protection, all RV vaccine efficacy trials have a clinical endpoint” (Angel J, Steele AD, Franco MA. Correlates of protection for rotavirus vaccines: possible alternative trial endpoints, opportunities and challenges. *Hum Vaccin Immunother* 2014; 1(10): 3659). Some rotavirus vaccine trials have used anti-rotavirus IgA seroconversion and/or response (Rotarix) whereas other vaccine trials have used serum neutralizing antibodies (Rotateq) as their primary endpoint. Serum IgA is a non-mechanistic correlate, it does not predict individual protection and a dose effect is not consistent between vaccines and assays. In addition, as IgA is not transferred via the placenta, newborns are born IgA deficient. Therefore, a newborn may not produce a significant or reliable sIgA response to an oral vaccine administered at birth. This is the justification for assessment of stool vaccine virus shedding as a marker of vaccine virus replication and “vaccine take” reflecting the combination of sIgA seroconversion and shedding, as additional markers of vaccine response in the primary clinical trials of the RV3-BB vaccine.

In the Discussion we have addressed this issue on line 554-562.

“We measured vaccine response using a combination of vaccine take, IgA seroconversion and vaccine virus shedding in the stool^{31,32}. This acknowledges that serum IgA is not an optimal serological correlate of protection for rotavirus²⁵. As maternal serum IgA is not transferred across the placenta, newborns are relatively IgA deficient at birth. For this reason, IgA seroconversion or serum IgA titers may not be a reliable marker to assess vaccine response of a rotavirus vaccine administered at birth. Although not validated as a correlate of protection, vaccine-virus shedding in the stool provides a valuable insight, at the gut level, of the physiological response to an live virus, orally administered vaccine and the potential impact of the gut microbiome on the rotavirus vaccine response. “

Authors’ rebuttal states:

Fig 1 describes the timepoints when the baseline stool was taken. In the Malawi study, a baseline stool was taken soon after birth prior to administration of the first dose of vaccine or placebo. In the Indonesia study the first stool was collected in the first week of life – 3-7 days after administration of the first dose of vaccine or placebo. Suggest to include this before result reporting to facilitate interpretation of country-specific results.

This information is defined in the Methods section, line 693-5:

“Stool samples were collected after administration of a dose of vaccine or placebo (Malawi study: baseline at 3 to 5 days, week 1, week 6, week 14; Indonesia study: 1 to 7 days at week 1, week 14-16, week 18-20) (Fig. 1). A pre-dose sample was collected prior to the administration of IP in the Malawi study ³¹.”

And in line 701-2:

“Microbiome analysis was conducted using samples collected 3 to 5 days after administration of a dose of vaccine or placebo”

Also now been included before results reporting as suggested on line 137-140:

“Participants from the Indonesia R3-BB trial and Malawi RV3-BB trial were eligible for inclusion into the RV3-BB Microbiome Study if they had adequate stool volume available for microbiome analysis at key study time points (Malawi: baseline, week 1, week 6, week 14; Indonesia: week 1, week 6-8, week 14-16, week 18-20) in the per-protocol population (Indonesia n=193; Malawi n=186).”

Results

Line 170, clarify how the stool samples for these time points were selected – was just any stool sample taken, or the closest to vaccination? For these timepoints, the microbiome is likely altered by rotavirus vaccine strain replication. Therefore associations between vaccine shedding and microbiome are logical, given that the highest vaccine replication will likely perturb the microbiome the most. Please rebut or include this consideration in your discussion.

The stool samples were collected 3 to 5 days after administration of a dose of vaccine or placebo with timepoints defined in the Methods section line 701-2. This has been clarified:

“Microbiome analysis was conducted using samples collected 3 to 5 days after administration of a dose of vaccine or placebo.”

Vaccine response in association with alpha and beta diversity

Very strong suggestion to include the IgA seroconversion response in the main text alongside the stool shedding and positive vaccine take, given that these are endpoints defined by the authors and that the IgA seroconversion endpoint has the best evidence that it correlates with vaccine protection.

Data on IgA seroconversion has been summarised in the revised Table 2 as recommended. Full data is available in Supplementary Table 5.

Table 2/Fig2

I appreciate that the authors took the time to collate their study results into this table and figure, but it remains complex to the reviewer, and therefore will likely be complex to the reader. It appears that there are separate analyses for whether or not there was vaccine take or shedding following 1, 2, or 4 doses for both schedule groups. See General comments. Could the authors further simplify by simplifying their endpoints. This reduces the number of statistical analyses, increases the understandability and improves comparability across country and schedule groups. Table 2 has been significantly simplified as recommended with binary outcomes and an additional data table with the complete analysis has now been moved to the Supplementary Appendix (Supplementary Table 4). In Figure 2, we have presented

binary outcomes (Y or N) for vaccine take and shedding for the Neonatal schedule group in Malawi, with respect to key study timepoints and according to vaccine dose. We believe this data provides important additional findings. We report that alpha diversity of the microbiome detected at an early timepoint is not only associated with positive vaccine take and shedding observed at that timepoint, but also for later vaccine doses. We also report that positive vaccine take after a birth dose of RV3-BB vaccine is associated with higher alpha diversity at week 14 suggesting that it is possible that birth dose administration may influence microbiome alpha diversity weeks later.

Suggest that the authors follow the approach they describe in their rebuttal: divide children in two 3 groups per study arm and simplify: ever had vaccine take (Y/N), ever had stool shedding (Y/N), seroconversion (Y/N), presented by study arm. This improves the understandability and improves the comparability across study arms of the study results. This table and fig2 could be moved to the supplementary. **Table 2** has been significantly simplified, with binary outcomes as recommended. The original data table with the more extensive analysis has now been moved to the Supplementary Appendix as recommended above (Supplementary Table 4).

Line 251 – suggest to not only describe the positive findings and equally note the lack of separation at other doses (and timepoints)? This framing suggests that there are important microbiome differences, but the preponderance of the evidence is actually not showing differences by vaccine endpoints for microbiome composition, which seems just as valuable an outcome to report.

We had selected the neonatal schedule group at week 6 as this group, and this timepoint, that is of greatest scientific interest. This is the critical timepoint to compare the administration of a birth dose of RV3-BB vaccine in the neonatal schedule with the placebo dose 1 at birth in the infant schedule. We have presented data on vaccine take and shedding as these were the only outcomes with a significant difference but to address the comment of the Reviewer we have documented in the text the lack of an association with IgA sero-conversion in line 186-187 and this is referenced in revised Table 2 and Supplementary Table 5.

Same suggestion for simplification in Figures 3 and 4, with per dose outcomes in supplementary

Figure 3 provides the comparison between the neonatal schedule group and the placebo group at key study timepoints and using a binary presentation of “y” or “n” for stool shedding as had been suggested by the Reviewer for Figure 2 above, so this is already in this format. The aim of this Figure is to compare alpha diversity in the non-shedders compared to the shedders and to that observed in the placebo group. In Figure 3, the alpha diversity of participants that shed was higher than that observed in the placebo group (and the non-shedding group) at week 14 (after 3 doses of vaccine administered in the neonatal schedule). We would argue that this is presented clearly in Figure 3, but to address the Reviewer’s request to simplify the Figure we have removed the data reporting results for vaccine take as there is no significant findings in these panels, the full Figure has now been included in the supplementary data (Supplementary Table 4).

Figure 4, if vaccine take definition did not include shedding, was the difference in PCoA at 6 weeks maintained?

Please refer to the definition of vaccine take (Methods; line 761-766) which defines vaccine take as the combination of IgA sero-conversion and stool shedding. So to confirm, vaccine take includes stool shedding in the definition.

Line 357, consider replacing PCA directionality with age directionality for taxa (like *Bacteroides*, *Escherichia* and *Streptococcus*) that distinguish separation across the axes

To address the Reviewer's comment we have:

1. Amended the text to clarify the bacteria that were identified as the main drivers of the age-related clusters: The text (line 368-374) now reads: "In the Malawi study, the primary taxa responsible for driving age-related cluster separation along PC axis 1 was *Bifidobacterium* (loading value 0.852), while *Escherichia* was a driver of age-related cluster separation along the PC axis 2 (loading value (loading value 0.815). In the Indonesia study PCA analysis, *Bifidobacterium* was the primary taxa responsible for the age-related separation along PC axis 1 (loading value -0.88287). The PC loading values are shown in Fig. 6 and additional PCA plots with overlaid Biplots visualising species driving age-related separations are shown in Supplementary Fig. 5.
2. Amended the Figure 6a to remove the reference to PC axis 3, which is not related to age, to reduce complexity for the reader.

Bacterial taxa influence RV3-BB vaccine response

Line 379, suggest to change subtitle to "Bacterial taxa associate with RV3-BB vaccine response"

We have changed the title as recommended to:

"Bacterial taxa associated with RV3-BB vaccine response"

Line 385, confirm no differences by seroconversion

This has been confirmed and added to the text (line 416-421) as recommended:

"At baseline, we observed no significant differences in bacterial taxa between vaccine response groups (positive or negative vaccine take; or positive or negative stool shedding; positive or negative IgA seroconversion) in the Indonesia and in Malawi study cohort, except for *Enterococcus* which was positively associated with vaccine virus shedding at week 14 in the neonatal schedule group in Malawi (coefficient = 15.6)."

Please note, in the Supplementary Table 7 and 8 shows MaAsLin2 analysis for taxa associated with vaccine variables. We have only considered taxa with a minimum abundance of 1% and, more importantly, with a MaAsLin2 coefficient value of ≥ 1.5 or ≥ -1.5 as relevant for the main study findings.

Line 460 – were there differences in breastfeeding rates between the two study cohorts given *Bifidobacterium* difference

The exclusive breast feeding rates for the Malawi RV3-BB study and the Malawi Rotarix study at study end were both $\geq 90\%$ of participants albeit that the period of reporting was slightly different in the 2 studies (RV3BB to 18 weeks and Rotarix to 11

weeks of age). However, the Reviewer is correct that the increased duration of exclusive breast feeding observed in the RV3-BB study cohort in Malawi could possibly have played a role on the difference in abundance of Bifidobacteria observed at week 6 and 14 between the 2 study cohorts. We have amended the sentence to include this:

“A higher abundance of *Bifidobacterium* in the RV3-BB study cohort observed at week 6 and week 14 compared to the Rotarix® study cohort ($p= 0.0046$ and $p=0.0006$ respectively) however there was a longer duration of exclusive breast feeding in the RV3-BB study cohort (**Fig. 8b**).”

Paragraph 453, is the reviewer correct that there was no re-analysis of associations between microbiome and Rotarix study vaccine endpoints in Malawi, only a comparison across ages for all included infants, regardless of vaccine take, shedding and seroconversion? This is confusing given data prior that evaluates associations with vaccine performance. Suggest to clarify in text.

For the Rotarix study, we re-analysed the microbiome data at key study defined timepoints (week 1 [pre-vaccine], week 6 and week 10). The microbiome at these timepoints was then assessed and reported for highly abundant bacteria after each dose of vaccine and in association to stool shedding measured in stool collected at the same timepoint. So our re-analysis is restricted to exploring the association between participant age and vaccine dose, with the abundant bacteria taxa identified in the Rotarix study, which is shown in the Supplementary Table 3 and 11 with the major outcomes are now included in Figure 7c and 7d.

Discussion

Line 520, what is meant by “shedding following”, this sentence is somewhat confusing, suggest to rephrase.

This has been amended and now reads line 549-551:

“Positive vaccine take and shedding was also associated with higher alpha diversity in stool collected 3 to 5 days after the time-matched dose of RV3-BB (vaccine dose 1 and microbiome week 1; and vaccine dose 2 and microbiome week 6). “

Line 523, please discuss that increased microbiome alpha diversity associating with shedding may be confounded by effective shedding increasing alpha diversity. We are not aware of any evidence that supports an association between viral shedding and differences in alpha diversity of the bacterial microbiome. This is a complex question and we don't think it can be adequately addressed within the scope of this Discussion.

Paragraph 546, suggest to name the vaccine studied for the cited literature for clarity.

We have added the vaccine names throughout this paragraph as recommended:

“Compared to the placebo group, participants receiving three doses of RV3-BB vaccine in the neonatal schedule have higher alpha diversity, although beta diversity and abundant bacterial taxa are similar. But we observed a significant association, positive and negative, between high abundance bacterial taxa and vaccine response to the RV3-BB vaccine. Bacteroides was consistently associated with a positive vaccine response to the RV3-BB vaccine. Bacteroides species are considered

beneficial taxa contributing to the metabolism of polysaccharides and oligosaccharides and facilitating the provision of nutrients to the host and luminal bacteria⁴⁰. In Ghana, the abundance of Bacteroides was reported in participants with no evidence of a vaccine response following Rotarix® vaccine administration whereas, in Zimbabwe, Bacteroides thetaiomicron was associated with serum IgA titer but not with IgA seroconversion^{16, 18}. In Nicaragua, a difference in the abundance of Bacteroides was observed between RotaTeq® vaccine sero-responders and non-responders¹⁹. A high abundance of the facultative anaerobes, Streptococcus, Staphylococcus, Klebsiella and Enterbacteriaceae, were negatively associated with vaccine response in our RV3-BB vaccine study in Malawi and Indonesia. This contrasts with the study in Ghana, where a positive association between the abundance of Streptococcus bovis and Rotarix® vaccine IgA seroconversion was reported¹⁶. The decrease in Staphylococcus abundance observed from week 1 to week 14 in our RV3-BB vaccine study in Malawi is consistent with patterns of colonisation in early life⁴⁴. E. coli abundance was similar in the Malawi and Indonesia RV3-BB vaccine study cohorts, although an association with the lack of a vaccine response was only observed in the Indonesian cohort. This finding conflicts with a study from Pakistan where increased E. coli abundance was associated with a positive vaccine response to Rotarix® vaccine¹⁷. “

Reviewer #2 (Remarks to the Author):

The revised manuscript is definitely improved, but still needs more work. There are three areas which need clarification.

We are grateful that the Reviewer has been satisfied that the extensive revision of the manuscript has addressed many of their concerns. We will address the additional comments in sequence below:

1 There is a major outstanding area of confusion: definitions of endpoints. In Methods, these are defined as follows:

“A positive vaccine response was defined as IgA seroconversion 28 days after administration of vaccine or placebo, stool shedding of RV3-BB detected after a dose of vaccine or placebo (Malawi: 3 to 5 days; Indonesia 1 to 7 days) and/or vaccine take at any study assessment timepoint (Fig. 1) 28,29. Vaccine take was defined as IgA seroconversion or stool shedding following administration of any dose of vaccine or 702 placebo. Positive vaccine take was defined as...”

Yet in Results (line 177ff) and in Table 2 results are presented as “positive vaccine take and/or vaccine virus stool shedding”, even though the definition of vaccine take includes stool shedding. This does make it difficult to follow. Would it not be simpler to present IgA seroconversion and stool shedding separately and to avoid the terms “take” and “response”? Or make the definitions clearer?

Thank you for this comment. This study is a substudy of the primary clinical trials of the RV3-BB vaccine conducted in Indonesia and Malawi and the immunogenicity endpoints in the current study directly reflect the endpoints reported for the previous clinical trials (Witte, D. et al. Neonatal rotavirus vaccine (RV3-BB) immunogenicity and safety in a neonatal and infant administration schedule in Malawi: a randomised, double-blind, four-arm parallel group dose-ranging study. *Lancet Infect Dis* **22**, 668–678 (2022).; Bines, J. E. et al. Human Neonatal Rotavirus Vaccine (RV3-BB) to Target Rotavirus from Birth. *N Engl J Med* **378**, 719–730 (2018). We are reluctant to

alter the previously published immunogenicity endpoints as this is likely to cause confusion across the various publications reporting results of these trials. However we have reviewed the manuscript for clarity and restricted our reporting to the three key terms: IgA seroconversion, stool shedding and the combined assessment of vaccine take. We have expanded the definition of these primary outcomes to avoid confusion in the Methods line 709-737.

One of the key challenges in the development of rotavirus vaccines has been the lack of accurate correlate of protection so that rotavirus vaccine efficacy is still dependent on a clinical endpoint (Angel J, Steele AD, Franco MA. Correlates of protection for rotavirus vaccines: possible alternative trial endpoints, opportunities and challenges. *Hum Vaccin Immunother* 2014; 1(10): 3659). Serum IgA is a non-mechanistic correlate, it does not predict individual protection and a dose effect is not consistent between vaccines and assays. In addition, as IgA is not transferred via the placenta, newborns are born IgA deficient. Therefore, a newborn may not produce a significant or reliable sIgA response to an oral vaccine administered at birth. Although not a correlate of protection, stool vaccine virus shedding has been increasingly used in oral rotavirus vaccine trials as a marker of vaccine virus response by way of gut level replication of the vaccine virus. In attempt to try and capture both markers which measure different aspects of the vaccine response, the term "vaccine take" has been used. We appreciate that, particularly for readers unfamiliar with oral vaccines, neonatal vaccines and rotavirus vaccines, that this terminology may seem unnecessary but it has been useful in providing an indicator of vaccine response in an "imperfect" situation.

In recognition of the potential for this to be clearer we have:

1. Defined the vaccine response endpoints in the abstract
2. Replaced the broader term "vaccine response" wherever it might be more clearly described specifically with respect to IgA seroconversion, shedding and/or vaccine take
3. Amended Table 2 to include all measures of the vaccine response in the Table, including IgA seroconversion, even no significant associations were observed
4. Removed vaccine take from Figure 3 so this now only presents shedding data
5. In the Discussion we have addressed this issue in lines 554-562:

"We measured vaccine response using a combination of vaccine take, IgA seroconversion and vaccine virus shedding in the stool^{31,32}. This acknowledges that serum IgA is not an optimal serological correlate of protection for rotavirus²². As maternal serum IgA is not transferred across the placenta, newborns are relatively IgA deficient at birth. For this reason, IgA seroconversion or serum IgA titers may not be a reliable marker to assess vaccine response of a rotavirus vaccine administered at birth. Although not validated as a correlate of protection, vaccine-virus shedding in the stool provides a valuable insight, at the gut level, of the physiological response to a live virus, orally administered vaccine and the potential impact of the gut microbiome on the rotavirus vaccine response. "

2 In Table 2, how are the P values derived?

The method for determining the difference between the alpha diversity and beta diversity indices in the participants that had positive vs negative IgA seroconversion, negative or positive shedding and positive or negative vaccine take with any

differences presented as the p value is included in the Figure 2 legend. The Figure 2 legend has been expanded so the derivation of the p values are now presented.

The authors have decided, wisely, to focus on the hypothesis now set out at the end of the Introduction. Table 2 should now reflect that.

We have amended Table 2 to reflect the central hypothesis as it relates to alpha and beta diversity data presented according to study group and markers of vaccine response (sIgA seroconversion, shedding and vaccine take) as a binary outcome at key study timepoints.

I would have expected a statistical test of the difference in (for example) alpha diversity between those with IgA seroconversion and those without. Is that what was done?

Yes this was done and had been reported in the original Supplementary Table 4. However in response to this Reviewer comment this data has now been inserted into the revised Table 2. Supplementary Table 5 provides a more in depth presentation of the IgA seroconversion data.

If so, what do the numbers 0.5299 or 0.0373 mean?

These are p values – please refer to the revised Table legend as noted above. Just to summarise all P values shown in Table 2 and Figure 2 have been now corrected across the different timepoint for a given vaccine variables using the Benjamini-Hochberg method. The uncorrected P values are still presented in Supplementary Table 4 for comparison.

If these are P values as the legend suggests, why is it being given to 4 decimal places?

We understand that this is consistent with the Journal guidelines.

If these are alpha diversities, both responder and non-responder values should be given. The legend needs to be more explicit as in its current form I find it unclear. The legend has been revised as recommended – see above.

My previous points have been addressed, though the issue of multiple testing is not fully resolved and should be addressed in the Discussion.

We have added into the Discussion (line 607-610) which now reads:

“Study methodology, age at sample collection, limitation in sample numbers within subgroups, intrinsic differences between vaccines and vaccine schedules, variability in definitions of vaccine response and vaccine virus shedding make it also challenging to compare data between studies and avoid the need for multiple testing.”

Reviewer #3 (Remarks to the Author):

REVIEWERS' COMMENTS

Reviewer #1 (Remarks to the Author):

Many thanks to the authors for their careful revision of their manuscript, and particularly the simplification of Table 2. All reviewer concerns have been addressed.

Reviewer #2 (Remarks to the Author):

I am happy to say that, from my perspective, the manuscript is greatly improved by the clarifications introduced into text and figure legends. I have two suggestions. The more important is to clarify causation. In line 547 it is stated that "higher α -diversity at early time points (baseline and week 1) was associated with positive vaccine take..." but the key table 2 and figures 2 and 3 and supplementary figure 4 do not show baseline data. Baseline data are shown in Supp Table 5, but no P values (if I have understood correctly). BL data are shown in Figure 5, but this is about time trends, not the difference between responders and non-responders. These key data are important to establish causation, so need to be shown in Table 2 and Figure 2. With this, it will be possible to strengthen the inference in the Abstract and the Discussion, either to claim that vaccine modulates the microbiota, or that the microbiota determines vaccine responsiveness.

Response:

- As recommended, we have inserted baseline data into a revised Figure 2. However, adding baseline data into Table 2 proved more difficult as this table presents data on the alpha and beta diversity in days 3-5/7 AFTER each dose of vaccine or placebo. As the baseline data reports the pre-vaccine microbiome, we were unable to reconfigure the table such that the baseline (pre-vaccine) data and the subsequent key timepoints week 1, week 6 and week 12 could be included into the same table format. As this data is now presented in Figure 2 for the Malawi study, we have included a revised Supplementary Table 4 and 5 where the baseline data is now presented with respect to vaccine response variables after all vaccine/placebo doses.
- Reviewer 2 also asked for baseline data to be inserted into Figure 3 (Indonesia study). But there were no baseline microbiome data available at baseline for the Indonesia study cohort as shown in Figure 1.
- All significant FDR-corrected P values are shown in Table 2 and in all supplementary tables where applicable.

We have amended the abstract and discussion as requested.

A rather minor point is that in line 544 the results referred to are really the results set out in the previous publications from this trial.

The sentence in the previous line 544 (now line 332) indicates that our results refer to microbiome differences between neonatal and infant vaccine schedule groups. And that a neonatal vaccine dose changes towards higher alpha diversity and a more beneficial microbiome.

Reviewer #3 (Remarks to the Author):
